# Temporally resolved analyses of aperiodic features track neural dynamics during sleep

Mohamed S. Ameen [1,2] ✉, Joshua Jacobs [2], Manuel Schabus [1], Kerstin Hoedlmoser[1] & Thomas Donoghue [2]

The aperiodic (1/f-like) component of electrophysiological data, whereby power systematically decreases with increasing frequency, as quantified by the aperiodic exponent, has been shown to differentiate sleep stages. Previous studies typically measured this exponent over narrow frequency ranges and averaged across sleep stages. A systematic review following PRISMA 2020 guidelines, which identified 16 eligible studies examining aperiodic neural activity during sleep, revealed heterogeneous frequency ranges and methodological approaches across studies. Building on these insights, the present study expands the analysis to include wider frequency ranges and alternative models, such as detecting 'knees' in the aperiodic component, which reflect bends in the power spectrum indicating changes in the exponent. Additionally, we applied time-resolved analyses to examine the dynamic patterns of aperiodic activity during sleep. We analyzed data from two sources: intracranial EEG (iEEG) from 106 epilepsy patients and high-density EEG from 17 healthy individuals and compared different frequency ranges and model forms of aperiodic activity. Results showed that broadband aperiodic models and the inclusion of a 'knee' feature effectively captured sleep stage-dependent differences in aperiodic activity. The knee parameter exhibited stage-specific variations, indicating different processing timescales across sleep stages. Time-resolved analysis of the aperiodic exponent tracked sleep stage transitions and responses to external stimuli, highlighting rapidly varying temporal dynamics during sleep. These findings offer valuable insights into brain dynamics during sleep and reveal novel insights and interpretations for understanding aperiodic neural activity during sleep.

Studies investigating the electrophysiology of sleep have classically relied on a scoring system focused on oscillatory activity (such as alpha oscillations, sleep spindles and slow wave activity) to classify sleep into different stages[1]. However, the electrophysiological signal contains a mixture of oscillatory, periodic components that rise above a non-oscillatory, aperiodic signal[2–4]. The aperiodic signal decays with increasing frequency in a $1/f^x$ relationship, quantifiable by the aperiodic exponent (x), which corresponds to the slope of the log-log power spectrum[3–6]. Building on earlier research that identified differences in aperiodic activity between sleep and wake states[2,3,7–9], recent studies have consistently found that the spectral exponent varies across different sleep stages[10–18], contributing to a growing body of literature focusing on aperiodic brain activity during sleep.

Changes in aperiodic activity during sleep provide potential information about the underlying physiological activity. Computational models, supported by empirical data from animal studies, have identified the spectral exponent as a non-invasive indicator of key neural processes including (i) transitions between up and down states[19,20], or (ii) the cortical excitation-inhibition (E-I) balance[15,21–24], where a steeper exponent suggests stronger inhibitory control, and a flatter exponent indicates higher excitation. This relationship is consistent with observations that the spectral exponent of the electro-encephalography (EEG) signal becomes progressively steeper with deeper sleep[12–14,17], consistent with an increase in inhibitory processes during sleep[25,26].

[1]Laboratory for Sleep, Cognition, and Consciousness research, Center for Cognitive Neuroscience Salzburg, Department of Psychology, University of Salzburg, Salzburg, Austria. [2]Electrophysiology, Memory, and Navigation Laboratory, Department of Biomedical Engineering, Columbia University, New York, NY, USA. ✉e-mail: mohamed.ameen@plus.ac.at

Accurately measuring and interpreting aperiodic activity requires methods that effectively differentiate aperiodic and oscillatory activity, which need to be employed using appropriate settings[27]. A key decision when examining aperiodic neural activity includes selecting the frequency range to analyze, including considering if chosen ranges could be biased by overlapping spectral peaks[28]. Previous sleep studies have thus far examined a wide variety of frequency ranges. Narrow frequency ranges such as the 30–45 Hz range[13,14,29], are often chosen due to their potential relationship to the E-I ratio[22].

In addition to the frequency range, analyzing aperiodic neural activity requires choosing the appropriate model form by selecting the function to fit to the power spectrum. The vast majority of the existing literature in sleep research has applied measures of a single exponent fit. As a result, previous studies have tended to examine a simple model fit to a narrow range, only characterizing and describing a restricted part of the neural power spectrum.

A key potential strength of analyzing aperiodic activity is to be able to characterize broad frequency ranges. To do this, one has to assess the overall shape of the power spectral density (PSD) to choose the most appropriate frequency range and model form. Notably, the neural power spectrum often displays an inflection point, referred to as the "knee"[4,30]. The frequency at which this knee occurs, referred to as the knee frequency, has been proposed to reflect the population timescale, i.e. the characteristic duration over which a neural population integrates or processes information[4,30]. Notably, this knee frequency has also been shown to vary systematically across sleep stages[13,31]. Collectively, this suggests that analyses of aperiodic neural activity during sleep may benefit from formalizing and potentially extending the frequency range under investigation. However, this requires considering different fit functions (e.g., fitting a function with a knee parameter), and selecting appropriate frequency ranges to fit across.

In addition, most studies of aperiodic activity during sleep thus far have predominantly focused on examining aperiodic activity over entire sleep stages. Recent methodological and empirical development have demonstrated the rich temporal dynamics of aperiodic activity and their relation to behavior[32–34]. This suggests that previous analyses may have overlooked nuanced dynamics within and across sleep stages. Novel methods are now available that can now be used to investigate questions such as whether changes in aperiodic activity show sharp transitions or slow drifts between sleep stages, and explore whether aperiodic activity undergoes event-related changes during sleep. Based on these considerations, our objective was to extend previous research by broadening the investigation of aperiodic activity during sleep.

Specifically, in this exploratory, data-driven investigation, we evaluated different frequency ranges and model forms for examining aperiodic activity in sleep recordings. We then applied these measures in a time-resolved manner to examine dynamics across multiple timescales throughout the night. In doing so, we sought to explore the temporal dynamics of aperiodic neural activity during sleep, hypothesizing that changes in such activity would track sleep stage transitions. To this end, we analyzed publicly available intracranial EEG (iEEG) data from across different sleep stages[35], as well as a high-density EEG dataset from healthy human participants through an entire night of sleep[36]. To measure aperiodic neural activity, we used the specparam toolbox -formerly 'FOOOF'-to compare model forms[32], fit spectral models, and compute time-resolved estimates.

In doing so, we highlight the advantages of using a model form for aperiodic activity that examines broader frequency ranges and accounts for the presence of the aperiodic knee. By further expanding the model forms and the temporal resolution of aperiodic activity measures, we aim to capture and characterize more of the variance of sleep data, in a way that highlights systematic relationships to sleep architecture and offers putative interpretations of the underlying neural circuits.

## Methods
We did not preregister the analyses for this study.

## Literature search
We conducted a systematic review of the literature to identify reports examining aperiodic neural activity during sleep and to investigate the frequency ranges used in such studies. The literature search was performed following the PRISMA 2020 guidelines[37] using the LISC Python toolbox (v0.3.0)[38], which finds publications based on specified search terms in the Pubmed database. Literature searches collected reports published between 1929—the year of the first published EEG paper—and the end of 2024.

**Systematic review of aperiodic activity.** To examine the prominence of investigations into aperiodic activity during sleep and its evolution over time, we quantified the number of publications mentioning predefined terms related to sleep and spectral properties. The initial search included descriptors of sleep activity ("sleep") as well as terms related to measures of aperiodic activity. To contextualize these findings, we conducted parallel analyses for other EEG-derived measures, specifically the Lyapunov exponent and measures of chaotic dynamics, which served as controls to compare the relative growth trajectories of different measures. Eligibility Criteria: Studies were included if they i) reported on brain activity or sleep, ii) included aperiodic or 1/f features, and iii) employed electrophysiological techniques (EEG, MEG, iEEG). We excluded reports that used consumer-grade devices. Data Collection and Analysis: Searches were conducted using the co-occurrence of the term "sleep" with: i) aperiodic measures ("aperiodic exponent," "spectral slope," "1/f," and "power-law exponent"), ii) Lyapunov exponent ("Lyapunov exponent," "Largest Lyapunov exponent," "Maximal Lyapunov exponent," "Lyapunov characteristic exponent"), and iii) chaos dynamics ("chaos," "chaotic dynamics," "deterministic chaos," "nonlinear dynamics," "chaotic measures"). We restricted these searches to electrophysiological modalities ("EEG," "MEG," "iEEG") and extracted the results over two-year intervals. For visualization purposes, the time range was set to start at the year with the first non-zero number of publications. We provide a flow diagram illustrating the search in Supplementary Fig. 1A.

**Frequency range for investigating the spectral exponent in sleep.** To examine studies that investigated the aperiodic exponent in the context of sleep, we ran an additional literature search to collect reports and metadata on such studies. The search terms specifically targeted studies examining the spectral exponent in sleep. These included variations of the spectral exponent ('spectral slope', 'spectral exponent', '1/f exponent', '1/f slope', 'aperiodic slope', 'aperiodic exponent', 'power-law exponent', 'power-law slope') combined with the keyword "sleep." Eligibility Criteria: We included studies that, i) focused on aperiodic features of neural activity, and ii) examined these features in the context of sleep. Data Collection and Analysis: The search yielded 21 studies, and metadata (publication year, title, abstract) were reviewed to confirm relevance. Five studies were excluded for the following reasons: i) consumer grade recording device (1 study), ii) absence of neural data (3 studies), or iii) non-human data (1 study). An additional five studies were excluded due to the absence of clear mention of the frequency bands used to estimate the aperiodic exponent. In addition, we manually added five studies meeting the inclusion criteria that were not detected by the literature searches[7,8,15,16,39]. In total, 16 studies were included in our analysis, detailed in Supplementary Table 1. We provide a flow diagram illustrating the search in Supplementary Fig. 1B.

## Datasets
**iEEG data.** We analyzed an openly accessible and previously published dataset of iEEG data from the Montreal Neurological Institute (MNI; https://mni-open-ieegatlas.research.mcgill.ca/). The dataset consists of one-minute recordings from 106 patients (52 females, 33.1 ± 10.8 years) with focal epilepsy during quiet wakefulness with eyes closed, NREM sleep (N2 and N3), and REM sleep. The dataset contains 1772 channels during Wake[40], 1468 channels in NREM sleep and 1012 channels in REM sleep[35].

Preprocessing. Already pre-processed data was accessed from the MNI database, which provides channels of bipolar data collected from neighboring pairs of electrodes. Briefly, raw data were low-pass filtered at 80 Hz and resampled to 200 Hz. Line noise was attenuated using an adaptive filtering approach that estimated the amplitude of the line noise frequency (50 or 60 Hz depending on recording site) and the first two harmonics, which were subsequently removed. The data were then visually inspected by an expert neurophysiologist, and artefactual segments of the signal were removed. Consequently, for some patients, the data used to compile one minute of recording comprised several non-consecutive segments. Each channel in each segment was demeaned, and when continuous 60 s data were unavailable, multiple shorter segments were concatenated. To minimize potential artifacts at segment boundaries, a 2-s buffer of zero amplitude was inserted between concatenated artefact-free segments. Up to five such segments were concatenated per condition, resulting in a maximum of four 2-second buffers and a maximum total duration of 68 s (60 s of EEG data plus 8 s of inter-segment buffers). As the number of clean segments varied across recordings, the number of inserted buffers also differed. Therefore, to ensure uniformity in signal length across all channels, sleep stages, and patients, signals shorter than 68 s were zero-padded at the end to reach a consistent length of 68 s. This standardization ensured consistent data and buffer durations across all recordings, preserving the comparability of PSD estimates across channels, sleep stages, and participants. The data set contains signals organized into 38 brain regions, with each region containing data from five patients. The five patients differed between regions but were always the same across sleep stages per region. Detailed preprocessing steps are described in the associated paper[35].

**EEG data.** We further analyzed a dataset of high-density EEG from 17 healthy human participants (14 females, 22.6 years ±2.3) who spent two nights in the sleep laboratory at the University of Salzburg[36]. We did not collect data on the race or ethnicity of the participants. The first night served as an adaptation night, in which we recorded polysomnography (PSG) data. The second night was an experimental night, during which we recorded PSG data while also presenting sounds throughout the night. For the analyses presented here, we used data only from the experimental night. The sounds were the subject's own name (SON) and two unfamiliar names (UNs) spoken by either a familiar voice (FV) to the subject or an unfamiliar voice (UFV). The stimuli were played via in-ear speakers and started immediately when the subjects went to bed. Throughout the night, auditory stimuli were presented continuously for 90 min (stimulation periods) then followed by a 30-min phase of quiet sleep (no-stimulation periods). This accounts for a 120-min cycle that was repeated four times during the night. We adjusted the sound volume individually for each participant based on their preference, ensuring that it was clearly audible without being excessively loud to avoid disrupting their sleep. We jittered the inter-stimulus interval between 2800 and 7800 ms in 500 ms steps. At the beginning of the experiment, all participants signed written informed consents. Participants received compensation in the form of money or credit hours upon completion of the experiment. The ethics department of the University of Salzburg approved the experiment. Acquisition and preprocessing: We used a high-density EEG 256-channel GSN HydroCel Geodesic Sensor Net (Electrical 478 Geodesics Inc., Eugene, Oregon, USA) and a Net Amps 400 amplifier. Data were acquired at a sampling rate of 250 Hz with Cz as the online reference. For preprocessing, face and neck channels (53 channels) were excluded and the analyses were carried out on 183 EEG channels. We high-pass-filtered the signal from these channels at 0.1 Hz, then removed the 50 Hz line noise using a notch filter. Thereafter, we removed and interpolated bad channels, restored the reference electrode (Cz), and re-referenced the data to an average reference using the PREP pipeline[41]. Finally, we performed independent component analysis (ICA) in EEGLab and visually labeled and removed noise-, heart-, and eye-related components.

**Simulated data.** For demonstration purposes of fitting spectral models, we simulated an example neural power spectrum using the specparam toolbox[32], with a frequency range of 1–45 Hz including two oscillatory peaks at 10 and 30 Hz, a knee frequency at 13.13 Hz, an exponent value of 1.25.

## Sleep staging and K-complex (KC) detection

Sleep EEG data were scored semi-automatically using an algorithm developed by the Siesta group (Somnolyzer 24×7; The SIESTA Group Schlafanalyse GmbH., Vienna, Austria[42,43], based on the criteria of the American Academy of Sleep Medicine (AASM)[44]. Sleep staging was conducted using two frontal (F3, F4), two central (C3, C4), two parietal (P3, P4), and two occipital (O1, O2) EEG electrodes. Two EOG channels were extracted from the high-density EEG cap: one positioned above the right eye and the other below the left eye, both referenced to the right mastoid. Bipolar EMG channels were recorded from facial muscles using EEG electrodes placed above the cheek muscles, with one electrode on each side of the face.

For the detection of K-complexes (KCs), we employed a dedicated algorithm developed by the SIESTA Group, which has been validated for its effectiveness in identifying KCs[45,46]. Briefly, the detection was carried out in a two-step process; first, the algorithm detects possible KCs via an approach that combines a matched-filtering detection method and a slow-wave detection method[47]. First, events that had (a) minimum negative-to-positive peak-to-peak amplitude of 50 mV and (b) a duration between 480 and 1500 ms were detected. Second, all detected events were matched to a prototypical template via wavelet analysis, and we used linear discriminant analysis (LDA) to select only real KCs. For our analysis, we considered real KCs to be events with an LDA score of 0.8 or higher. KCs were detected at C3 and C4, and the events detected at C3 were used for the analysis. Evoked KCs were defined as events that started within the 2 s post-stimulus-onset window during N2 sleep. Further details on the sleep architecture-related processing are described in ref. 48.

## PSD estimation

**PSD estimation across entire sleep stages (EEG and iEEG data).** To estimate power spectra across entire sleep stages, we used Welch's method[49] on both EEG and iEEG data, with a 15 s window length with 50% overlap and tapered with a Hamming window. We created average PSDs per stages by taking the mean over all segments per epoch then averaging over all epochs. To compare the performance of the model at different frequency ranges and time windows of PSD calculation, we employed different time windows and frequency ranges for Welch's PSD calculations, with all other settings left unchanged. We used four different time windows (5 s, 10 s, 15 s, 20 s) and examined different frequency ranges covering either broad (1–30 Hz, 1–45 Hz, 1–60 Hz, 1–75 Hz) or and narrow ranges (30–45 Hz, 25–45 Hz, 1–20 Hz, 1–8 Hz). These ranges were informed by the literature search. Additionally, we varied the locations of the frequency bands while keeping the bandwidth fixed (1-20 Hz, 10-30 Hz, 20-40 Hz, 30-45 Hz) to control for possible effects related to the position of the frequeny range. We report the results averaged over all sleep stages of iEEG (Wake, N2, N3, and REM) or EEG (Wake, N1, N2, N3, and REM) data. For the whole night time-frequency plot, we plotted the average value per epoch.

**PSD estimation across sleep stage transitions (EEG data only).** For analyses across sleep stage transitions, the EEG data were segmented into 20 s intervals with a moving window of 2 s, resulting in a 90% overlap between adjacent segments. We then applied Welch's method to calculate the PSD for each of these segments using the same parameters outlined in section above. We focused on segments falling within a 120 s timeframe around the transitions, extending from 60 s before to 60 s after each transition, to capture the spectral dynamics occurring around these changes in sleep stages.

**Analysis of the temporal dynamics during specific events (EEG data only).** For measuring aperiodic activity during auditory stimulation and KC events, we computed frequency representations using a multi-taper approach within the 1–45 Hz frequency range, with 0.5 Hz steps. We used Discrete Prolate Spheroidal Sequences (DPSS) as tapers and performed FFT-based convolutions with the number of cycles for each frequency set to match the frequency, resulting in a consistent time window of 1 s across all frequencies (e.g., at 1 Hz: 1 cycle → 1 s window, at 45 Hz: 45 cycles → 1 s window). EEG data were segmented into 10-s epochs centered around the event of interest (stimulus onset or KC onset). Bad epochs (amplitude fluctuations >1000 μV) were excluded from analysis. To avoid edge artifacts, the first and last second of each epoch were discarded. We then baseline-corrected the resulting time-resolved aperiodic estimates by subtracting the mean of the estimates in the 500 ms pre-stimulus-onset window. We used a sampling rate of 250 Hz for all time-resolved analyses except for KC vs no-KC analysis, where we downsampled the data to 128 Hz, as KCs were scored on downsampled data. For KC analysis, we used $74.47 \pm 45.04$ trials on average per subject. For the comparison between the exponent during stimulation and no-stimulation periods, one subject was removed from NREM and another from REM analysis due to poor model fit ($R^2$ values lower than 2 standard deviations below the mean). Thus, these analyses were conducted on 16 subjects. Further, for NREM, we restricted the analysis to only N2 and N3 stages, since brain responses to external stimuli are well documented during these stages.

### Spectral parametrization

The specparam toolbox (formerly: 'FOOOF', v1.0.0) was used to parametrize iEEG and EEG power spectra[32]. Unless otherwise specified, power spectra were parameterized across the frequency range 1–45 Hz. This frequency range was used as a standard, except in cases where we specifically focused on investigating the impact of different frequency ranges. Settings for the algorithm were defined as: peak width limits: 1–12 Hz, max number of peaks: 8; minimum peak height: 0; peak threshold: 2. Aperiodic activity was defined according to the formula: $10^b \times 1/(k + f^{1/x})$, where x is the exponent at given frequency range (f), b is the y-axis intercept and k is the knee parameter. The aperiodic mode was set to either 'fixed' for a fixed/single-exponent model, i.e. a model that fits a single exponent value to the whole spectrum was used or 'knee' when adding a knee parameter to the model, which models a bend in the PSD at a specific frequency after which the exponent changes. The knee frequency is calculated from the knee constant using the formula: $knee_{freq} = k^{1/x}$, where k is the knee value, and x is the exponent[30]. We excluded knee frequency values that were more than 2 standard deviations from the mean knee frequency for each subject and each sleep stage. Further, we checked the goodness-of-fit ($R^2$) for all the spectral models. To examine potential changes in aperiodic parameters across the course of the night, we split the total number of epochs, per sleep stage, into quartiles, and examined average exponent values per quartiles across participants.

### Event-related potential (ERP) analysis

To measure the auditory ERPs of all stimuli as well as the difference between the ERPs of the different stimulus categories (FV vs. UFV and SON vs. UNs), the data were segmented into 10 s epochs centered around stimulus onset. We performed baseline correction using the 500 ms prestimulus-onset window and according to the formula: (data – mean baseline values)/mean baseline values, followed by calculating the grand average of all epochs per participant

### Classification analysis

In order to evaluate the discriminability of different sleep stages based on the estimated aperiodic parameters, classification analysis was done using the scikit-learn toolbox[50]. We employed an LDA classifier with K-fold cross-validation (5 splits, 2 repetitions). The number of 30 s epochs per stage was equalized and the total number of trials over all stages used for training and

testing was equalized between conditions. For iEEG data regions with less than 25 trials per condition were discarded leaving data from 16 out of the 38 regions average, the number of epochs used per subject was $41.25 \pm 19.72$ per condition for the iEEG data and $182.7 \pm 0.47$ for the EEG data. We employed a stratified K-fold cross-validation with 5 splits and 2 repetitions, ensuring that in each iteration, the classifier was trained on 80% of the trials and evaluated on the remaining 20%. Chance levels were defined as the reciprocal of the number of alternative outcomes. For instance, in the case of a five-class classification, a chance level of 1/5 or 0.2, denotes the accuracy level that could be achieved only by guessing. To assess the relative importance of the knee frequency and Exponent in the knee Model, we performed a random forest analysis using 100 estimators with 5-fold cross-validation. Subsequently, we conducted a permutation feature importance analysis with 1000 repetitions to quantify the contribution of each feature to the model's predictive performance. To address collinearity, we reduced the number of features to one, ensuring that the analysis focused on a single predictor at a time.

### Statistical analysis

For comparisons of exponent values and knee frequencies across sleep stages as well as decoding accuracies between different parameters, we performed Friedman chi-square tests and reported chi-square ($X^2$) statistics, p-values, as well as Kendall's $W$ as a measure of effect size. $W$ was computed by the equation: $W = X^2/N(K-1)$, where $N$ is the sample size and $K$ is the number of measurements per subject. Kendall's $W$ ranges between 0, indicating no relationship, and 1, indicating a perfect relationship. Interpretation of $W$ values followed commonly used benchmarks, where $W$ values between 0.00 and 0.20 indicate slight agreement, 0.21–0.40 fair agreement, 0.41–0.60 moderate agreement, 0.61–0.80 substantial agreement, and values above 0.80 reflect almost perfect agreement[51]. We performed post-hoc tests, when applicable, via Dunn's test with Bonferroni's correction for multiple comparisons and reported the z-values, p-values and Cliff's delta (cd) as a measure of effect size which is interpreted using the following values: <0.33 (small), from 0.33 to 0.474 (medium), and ≥0.474 (large)[52]. To provide 95% confidence intervals for cd, we used a large-sample normal approximation. Specifically, Cliff's delta was expressed as $cd = 2p - 1$, where p denotes the probability that a randomly chosen observation from one group exceeds an observation from the other group. The standard error of p was approximated as: $SEp = \sqrt{p(1-p)/(n1*n2)}$, with n1 and n2 denoting the group sizes. The 95% CI for p was calculated as $p \pm 1.96 \cdot SE$ and subsequently transformed back to Cliff's delta. This procedure assumes independence of pairwise comparisons and no ties, and should be regarded as an approximation. For comparing $R^2$ values between broadband and narrowband frequency ranges, we performed a Wilcoxon sign-rank test and reported W as test statistic, p-values and rank biserial correlations (r) as a measure of effect size. Correlations between knee frequency and exponent values were performed either using Pearson's or Spearman's according to the results of the normality test and correlation coefficients and p-values were reported.

We measured model performance using the Bayesian Information Criterion (BIC) for Gaussian models[53] for the different models, calculated through the equation: $BIC = N \times \log(mse) + np * \log(N)$, where $N$ is the sample size, mse is the mean squared error calculated as square of the model error parameter, and np is the number of parameters in the model. Since we fit a maximum of 8 peaks, the number of parameters was equal to: n_peaks * 3 + n_ap_params, where n_peaks depends on the model fit (up to a maximum of 8), and n_ap_params was 2 for the fixed model (offset, exponent), and 3 for the knee model (offset, knee, exponent). A difference in BIC between the two models between 0 and 2 constitutes 'weak' evidence in favor of the model with the smaller BIC, a difference between 2 and 6 constitutes 'positive' evidence; and a difference above 6 constitutes 'strong' evidence[54].

We compared classification accuracies against chance levels using a permutation t-test. We created an array of chance level values that is equal to the original array of observations. Then, we generated a distribution by

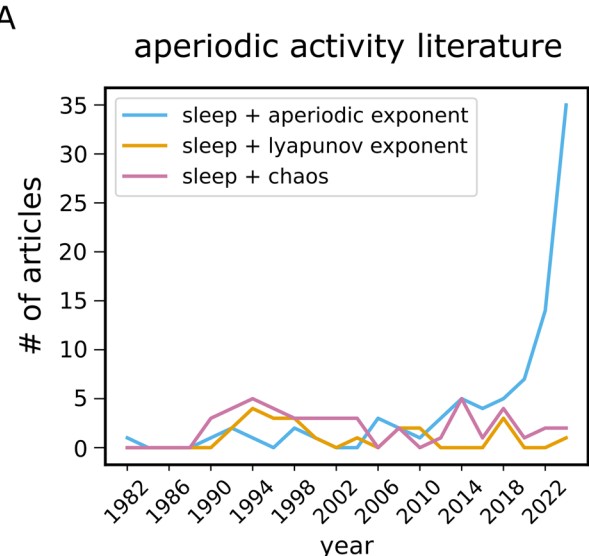

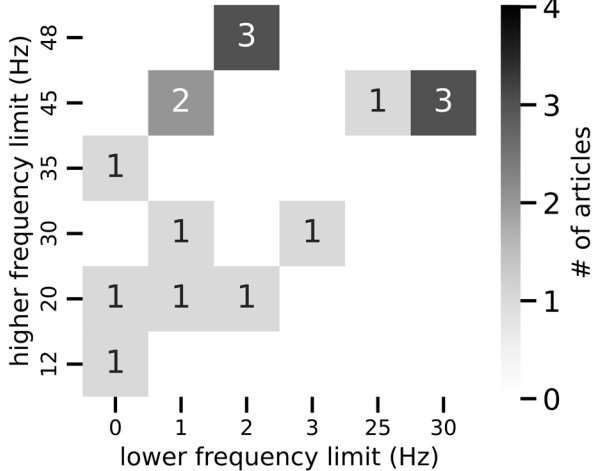

**Fig. 1 | Review of literature on aperiodic activity during sleep. A** The number of peer-reviewed articles retrieved from searches combining "sleep" with predefined electrophysiological measures. Specifically, searches included: i) aperiodic activity ("aperiodic exponent," "spectral slope," "1/f," and "power-law exponent"), ii) Lyapunov exponent ("Lyapunov exponent," "Largest Lyapunov exponent," "Maximal Lyapunov exponent," "Lyapunov characteristic exponent"), and iii) chaotic dynamics ("chaos," "chaotic dynamics," "deterministic chaos," "nonlinear dynamics," "chaotic measures"). Results were restricted to electrophysiological modalities ("EEG," "MEG," "iEEG") and aggregated over two-year intervals. Notably, the number of articles investigating aperiodic activity during sleep increased substantially in recent years, whereas studies on Lyapunov exponents and chaotic dynamics did not witness such growth. This contrast indicates that the observed rise reflects a specific increase in interest in aperiodic activity rather than a general trend across electrophysiologically-derived measures of sleep. **B** A heatmap illustrating the frequency ranges used to measure aperiodic brain activity in the sleep literature. Each number represents the number of studies using the frequency range defined by the lower limit on the x-axis and the upper limit on the y-axis.

shuffling the labels of the two arrays then split the resulting array into two before performing a $t$-test. This process was repeated 10,000 times and we calculated the averages of $t$-value. We report Bonferroni-corrected $p$-values, and Cohen's d effect sizes. We compared feature importance results using a parametric $t$-test and reported Cohen's d effect sizes. To track the change in aperiodic parameters (exponent and knee frequency) across epochs of sleep stages, a regression analysis was done for every sleep stage using the *statsmodels* toolbox in Python[55] and the regression coefficients ($R^2$), the F-statistics, and the $p$-values were reported. Comparison of time-series data was done using a cluster-based permutation analysis implemented in the *Fieldtrip* toolbox[56,57] in Matlab (v. 2019a) using a two-sided $t$-test and 5000 permutations. Alpha level was set at 0.025 and we report the sum of t-statistics ($\sum$t) of the cluster, $p$-values as well as Cohen's d measured over all possible permutations.

### Reporting summary
Further information on research design is available in the Nature Portfolio Reporting Summary linked to this article.

## Results
### Model selection for estimating aperiodic activity
A primary aim for this study was to broaden the scope of analyzing aperiodic activity during sleep. Given the growing interest in this field in recent years (Fig. 1A), previous studies have employed diverse frequency ranges (Fig. 1B) and model forms, which can lead to substantial variability in the resulting fits (Fig. 2). To address this, we first systematically investigated how the selection of frequency ranges and model forms influences the results.

To evaluate the impact of the chosen frequency range on the measured aperiodic parameters, we manipulated two settings of the PSD estimation procedure: a) the frequency range the model is fit to, and b) the time window used in Welch's power estimation, which influences the frequency resolution. For this analysis, we used the iEEG dataset, including all patients, regions, and sleep stages, to fit a fixed exponent model to either broad (Fig. 3A) or narrow frequency ranges (Fig. 3B). This approach allowed us to evaluate the stability of the models across varying conditions. The results demonstrated that although all models had consistently high $R^2$ values, irrespective of frequency range or time window (all >0.86), $R^2$ values were significantly higher for broadband (0.99 ± 0.004) ranges as compared to narrow band (0.95 ± 0.04) ranges (Wilcoxson sign rank test: $W = 136$, $p < 0.001$, $r = 0.92$). Broadband models also showed less variation in $R^2$ values between the different frequency ranges and time windows (Fig. 3C, Wilcoxson sign rank test: $W = 0$, $p < 0.001$, $r = 0.98$), suggesting that models fit on broadband ranges provide more stable model estimates.

Furthermore, the exponent values calculated across various broad and narrow frequency ranges (Supplementary Fig. 3A, B) exhibited substantially reduced variability with broad ranges relative to those derived from narrow frequency bands (Fig. 3D, Wilcoxson sign rank test: $W = 0$, $p < 0.001$, $r = 1$). Additionally, to examine the association between model fit stability and spectral estimates, we compared the variance of spectral exponents with the variance of $R^2$ values using Spearman correlation analyses, conducted separately for broad and narrow frequency ranges while controlling for sleep stage (Fig. 3E). In the broad frequency range, the partial correlation revealed a moderate positive association between exponent variance and $R^2$ variance (rho(13) = 0.37, $p = 0.003$, 95% CI = [0.13, 0.56]). A similar effect was also observed in the narrow frequency range, with a partial correlation of $r = 0.37$ (rho(13) = 0.003, 95% CI = [0.14, 0.57]). These results indicate that greater variability in model fits corresponds to greater variability in exponent estimates, suggesting that instability in model fitting can reduce the replicability of spectral exponent measures.

We observed similar results using the knee model (Supplementary Fig. 3C, D). However, in the EEG data, model performance varied more notably with frequency range: broader ranges yielded lower model fits compared to narrower bands. For instance, the 1–8 Hz range resulted in a higher model fit than 1–65 Hz and 1–75 Hz (Supplementary Fig. 4). Further, we found no correlation between the $R^2$ and exponent values across all models, indicating that the exponent values are not systematically biased by model fit quality (Supplementary Fig. 5). To validate our analysis and disentangle whether differences in model fits were driven by the bandwidth (narrow vs. broad) or the location of the frequency range used (low vs. high frequencies), we conducted additional analyses where we varied either the

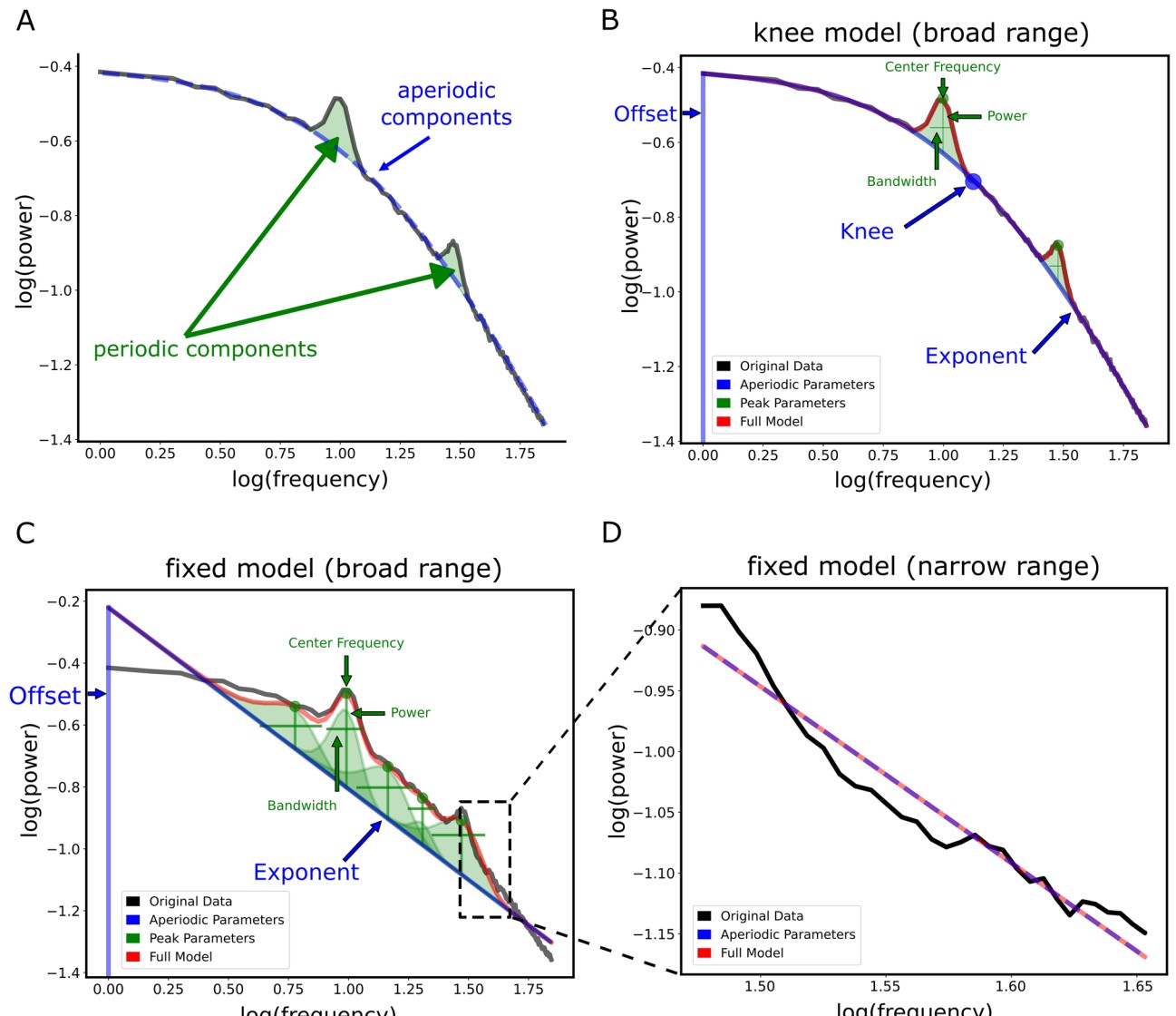

**Fig. 2 | Schematic of measuring aperiodic activity from power spectra, examining different frequency ranges and model forms. A** An example of a simulated power spectrum, with 2 oscillatory peaks at 10 Hz and 30 Hz, a knee at 13.13 Hz and an exponent of 1.25. Periodic components are highlighted in green, and the simulated aperiodic component is shown as dashed blue line. **B** An example spectral model fit using a 'knee' model, i.e. a model that incorporates a knee parameter, with annotated spectral features. Notably, fitting this model over a broad frequency range (1–45 Hz) produced a very high goodness-of-fit measure ($R^2 = 0.99$) and low mean squared error (MSE = 0.004). **C** Example spectral model fit using the same frequency band but fitting a fixed model, i.e. a model that assumes a single exponent value, resulted in

a high $R^2$ (0.99) but increased error (MSE = 0.013) as compared to the knee model. Note the difference in the number of oscillatory peaks (green) between **B** and **C** - while both models can attain high $R^2$, if there is a model mismatch, the fixed model tends to overfit oscillatory peaks. **D** An example model fit of a fixed model fit over a narrow (30–45 Hz) frequency band. This model had the lowest model performance ($R^2 = 0.95$, MSE = 0.016). For an example of fitting a model without simulated a knee component see Supplementary Fig. 2, which also shows that both the knee and fixed models resulted in high and comparable performance levels (both with an $R^2$ of 0.99), in contrast to the narrowband model fit which had a lower fit value ($R^2 = 0.95$).

width of the frequency band (Supplementary Fig. 6A) or its location along the spectrum (Supplementary Fig. 6B). These analyses revealed a decline in model fit ($R^2$) when narrower frequency bands were used or when lower frequencies were excluded. Collectively, while $R^2$ values do not in isolation adjudicate between good and bad models (see e.g. Fig. 2), these results suggest that fitting spectral models over broader frequency ranges, particularly those that include lower frequencies, provides more reliable estimates.

**Stage-specific aperiodic activity in iEEG data**
To examine the results of different models for estimating aperiodic parameters in empirical data, we analyzed publicly accessible iEEG sleep data containing epochs from Wake, N2, N3, and REM sleep[35]. In a first step, we visually inspected the PSDs of the different sleep stages,

averaged over cortical regions (Fig. 4A). Based on the visible appearance of a knee in the PSDs, we fit a knee model, as illustrated in the annotated model-fit example (Fig. 4B). The knee model better captured the data as compared to using the fixed model (Supplementary Fig. 7). Moreover, the single-region PSDs per sleep stage (Fig. 4C) demonstrated that when the knees were detected in the average PSD, they were consistently present across the recorded brain areas but varied in their frequencies across stages.

Previous findings have reported that the spectral exponent differs between sleep stages[12–14,18]. We sought to replicate these findings using both the knee model and the fixed model. Applying the knee model, the difference in the exponent values between the stages was significant (Fig. 4D left; Friedman chi-square test: $X^2 = 46.11$, $p < 0.001$, $W = 0.4$). Dunn's post-hoc test with Bonferroni's correction revealed that the exponent decreased

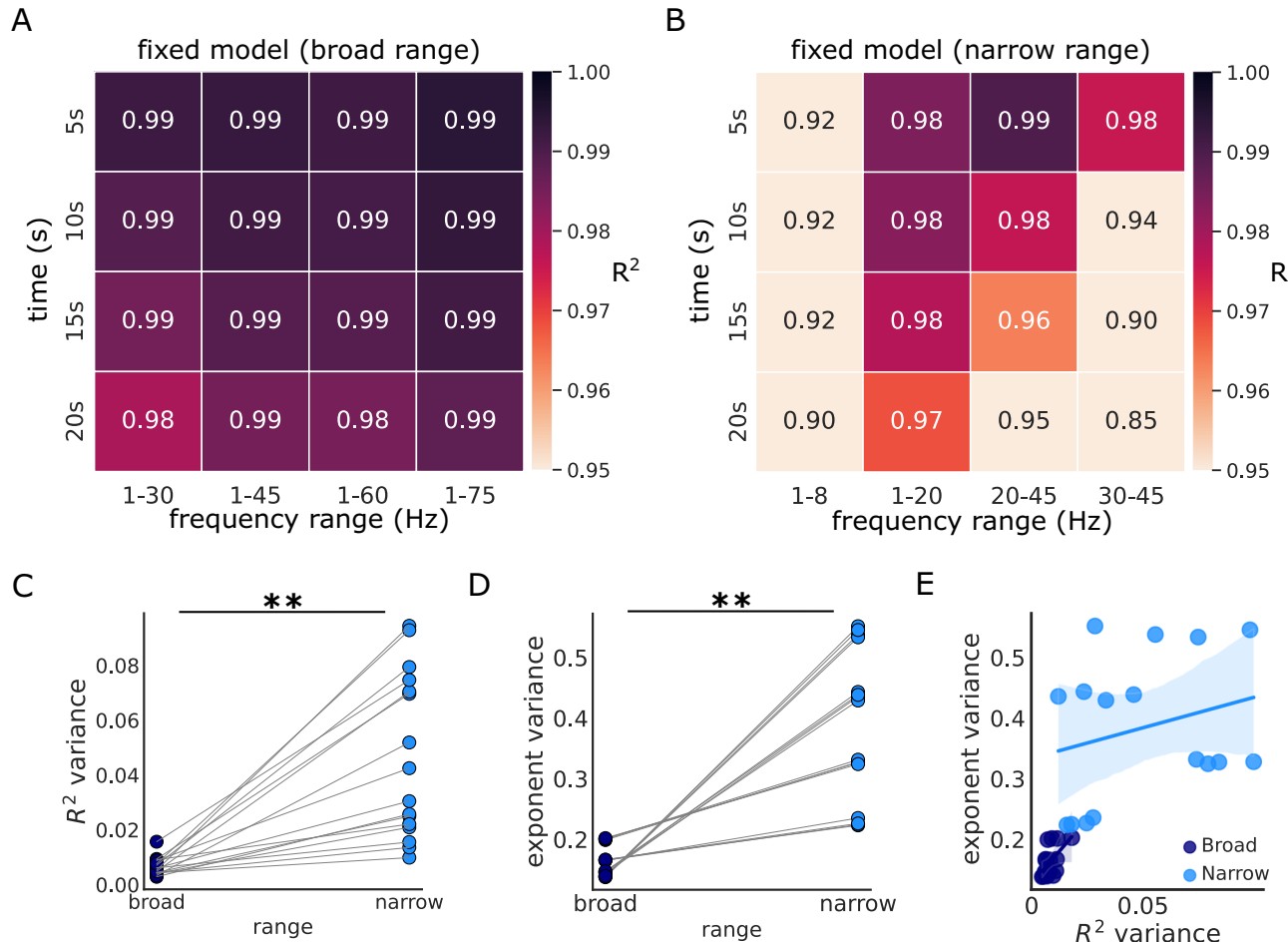

**Fig. 3 | Effects of frequency range and time window on model performance in iEEG data. A** The $R^2$ values, indicating goodness-of-fit, for models applied to different broad frequency ranges (x-axis) and with different time windows (y-axis). **B** The $R^2$ values for models using narrow frequency ranges. Note the variability of $R^2$ values in comparison to (**A**). Models incorporating broadband frequencies demonstrated a marked reduction in variance for **C** $R^2$ values and **D** estimated exponents. **E** The relationship between $R^2$ variability and exponent variability.

A significant positive correlation was observed between the variances in $R^2$ and exponent values of the narrow and broad frequency ranges. iEEG data were obtained from 38 brain regions across 106 subjects, with each region including a different subset of subjects. Each point on the graphs symbolizes the value for a particular combination of time window and frequency range, averaged across data from all sleep stages. **\*\***$p < 0.001$.

significantly from wakefulness to N2 ($p_{bonf} < 0.001$), to N3 ($p_{bonf} < 0.001$), and to REM ($p_{bonf} < 0.001$). There was no statistically significant difference, however, between the exponents of the different sleep stages (N2-N3, N2-REM, and N3-REM: $p_{bonf} = 1$), see Table 1 for the detailed statistical results. Importantly, the knee frequency exhibited significant stage-dependent differences (Fig. 4D right; Friedman chi-square test: $X^2 = 95.34$, $p < 0.001$, $W = 0.84$). Specifically, the knee frequency decreased from Wake to N2 ($p_{bonf} < 0.001$), to N3 ($p_{bonf} < 0.001$), and to REM ($p_{bonf} = 0.001$). Moreover, the knee frequency was lower for N3 than N2 ($p_{bonf} = 0.001$) and REM ($p_{bonf} < 0.001$), but did not differ between N2 and REM ($p_{bonf} = 0.13$) The detailed results of the post-hoc tests are reported in Table 2. We also observed significant stage-specific differences in the exponent of the fixed model (Fig. 4E, Friedman chi-square test: $X^2 = 99.69$, $p < 0.001$, $W = 0.87$), replicating previous work. Specifically, the exponent decreased significantly from Wake to N2 ($p\_bonf < 0.001$), Wake to N3 ($p_{bonf} < 0.001$), from N2 to N3 ($p_{bonf} = 0.001$), from N2 to REM ($p_{bonf} < 0.001$), and from N3 to REM ($p_{bonf} < 0.001$). However, there was no significant difference between Wake and REM ($p_{bonf} = 1$). The detailed results are reported in Table 3. $R^2$ values of the models are depicted in Supplementary Fig. 8. It is also important to note that the knee model we use fits a Lorentzian function, which assumes a flat exponent below the knee frequency. As this is not always true, we aimed to assess whether pre-knee exponents differ across stages. Thus, we fit a fixed model to the frequency range from 1 Hz up to

the knee frequency. The resulting exponents showed sleep-stage-dependent variations (see Table 4 and Supplementary Fig. 9).

To test whether the aperiodic parameters are good predictors of the difference between sleep stages, we employed an LDA classifier trained on the knee frequency, the exponent of the knee model, or the exponent of the fixed model, using brain regions as features (Fig. 4F). The classification results revealed significantly above chance-level classification accuracy using the knee frequency ($t(15) = 10.57$, $p_{bonf} < 0.001$, $d = 3.73$), the exponent of the fixed model ($t(15) = 18.28$, $p_{bonf} < 0.001$, $d = 6.46$), as well as the exponent of the knee model ($t(15) = 5.61$, $p_{bonf} < 0.001$, $d = 1.99$). A comparison of the three parameters yielded significant differences ($\chi^2(2) = 24.13$, $p < 0.001$, $W = 0.75$). Dunn's post-hoc test with Bonferroni's correction revealed no significant difference in decoding accuracy between knee frequency and the exponent of the fixed model ($z(15) = 1.79$, $p_{bonf} = 0.22$, $d = 0.4$). However, decoding with the exponent of the knee model was significantly worse than with that of the fixed model ($z(15) = 5.03$, $p_{bonf} < 0.001$, $d = 1$) or with the knee frequency ($z(15) = 3.23$, $p_{bonf} = 0.004$, $d = 0.7$).

Given this pattern of multiple aperiodic parameters relating to sleep stages, we next sought to explore the potential relationship between these parameters. To do so, we computed Spearman correlations between the knee frequency and the exponent estimates while controlling for sleep stages. Comparing the knee frequency of the knee model and the exponent

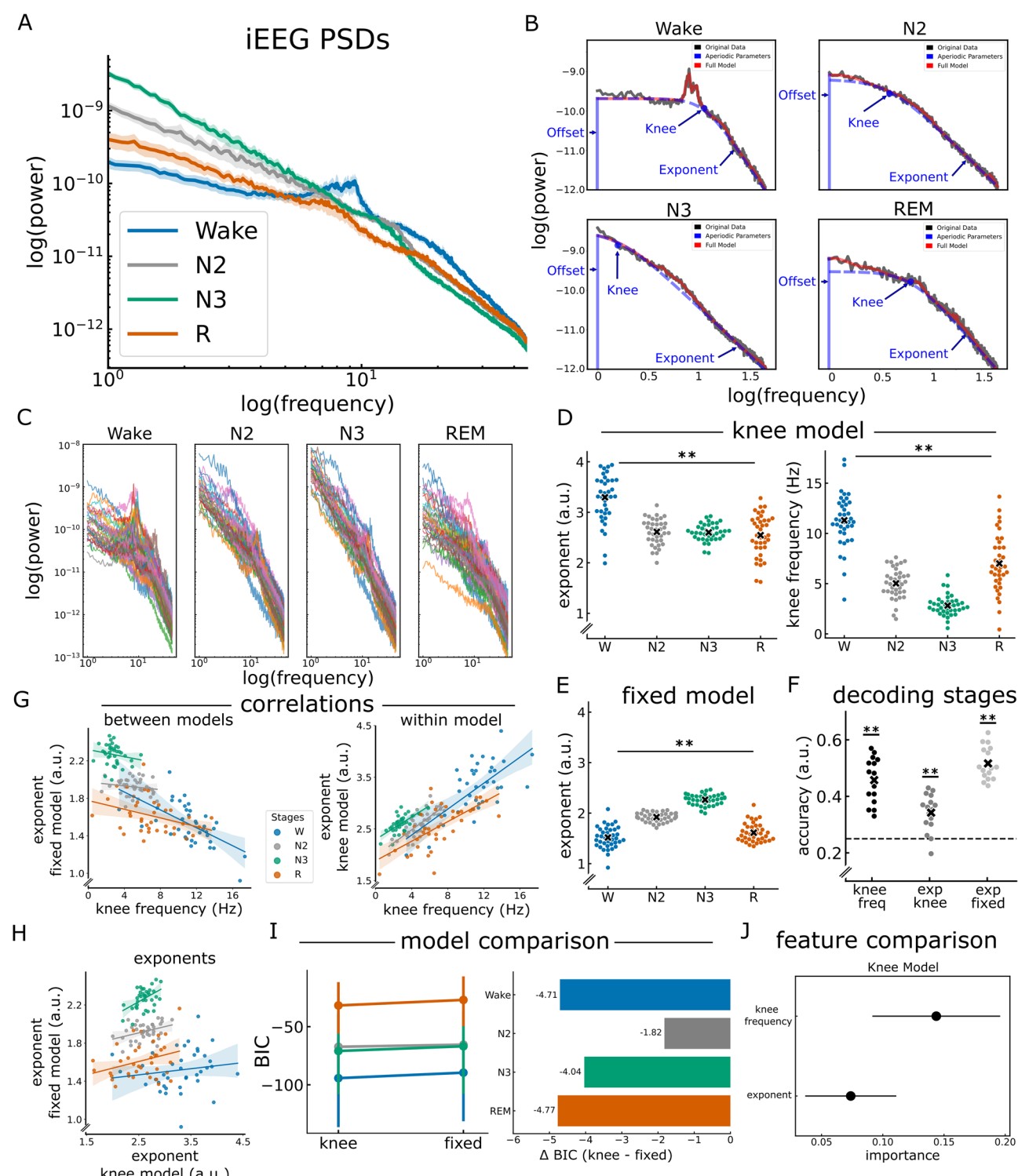

of the fixed model – there was a significant negative correlation (Fig. 4G left; rho(149) = −0.64, 95% CI [−0.73, −0.54], $p < 0.001$). This relationship is reversed when examining the exponent values resulting from the knee model. Specifically, the correlation between the knee frequency and the exponent of the knee model was positive and significant (within-model correlation: Fig. 4G right; rho(149) = 0.61, 95% CI [0.50,0.70], $p < 0.001$). Further we conducted a Spearman correlation to examine the relationship between the exponents from the two different models and found a weak and non-significant correlation (Fig. 4H; rho(149) = 0.08, 95% CI [−0.08,0.23], $p = 0.35$). These findings highlight a potential interdependence between the knee frequency and the spectral exponent, suggesting that in the presence of

a knee, exponent estimates derived from the fixed model may partially reflect underlying changes in the knee frequency. In the context of sleep data, this suggests that previously reported sleep-stage differences in the exponent may, at least in part, be driven by differences in the knee frequency between sleep stages, as seen here when explicitly measuring knee models.

To compare the performance of the two models in the iEEG data, we used the Bayesian Information Criterion (BIC) for Gaussian distributions (Fig. 4I), which is based on the likelihood function and evaluates model fit while penalizing complexity. BIC differences between the knee and fixed models favored the knee model across all sleep stages, as indicated by the

**Fig. 4 | Aperiodic activity in iEEG sleep data. A** Average PSDs of the different sleep stages of the iEEG data. **B** An example of the aperiodic fit from one brain region using the knee model across a broad frequency range in the different sleep stages of the iEEG data. The model was fit to the average PSD of each sleep stage. **C** The PSDs of the different sleep stages, where each color represents a different region. **D** (Left) The exponent values from the knee model across sleep stages (fit range: 1–45 Hz). Note that the exponent values are different between wakefulness and sleep but show no differences across sleep stages. (Right) The knee frequency values across sleep stages. Unlike the exponent, the knee frequency differed significantly between sleep stages. **E** The exponent of the fixed model (fit range: 1–45 Hz) was significantly different between sleep stages. **F** The classification accuracy between sleep stages using an LDA classifier that uses either the knee frequency, the exponent of the fixed model, or the exponent of the knee model. All three parameters demonstrated above performance in differentiating between sleep stages. **G** Correlations between the knee frequency and the exponents of both the knee and the fixed models. The change in

knee frequency correlates negatively with the exponent of the fixed model and positively with the exponent of the knee model. **H** Correlation between exponents of both models, showing a weak, non-significant positive correlation. **I, J** Comparison of aperiodic models and features in iEEG data. I) BIC comparison between the knee model and the fixed model. (Left) absolute BIC values for both models across sleep stages, with lower values indicating better fit. (Right) BIC difference (knee - fixed) across sleep stages. Negative values indicate that the knee model outperforms the fixed model. Dots represent mean values, and lines indicate 95% confidence intervals. **J** Random forest feature-importance analysis comparing the knee frequency and the exponent of the knee model, indicating greater importance of the knee frequency. Dots represent mean values, and lines show the standard error of the mean (SEM). iEEG data were obtained from 38 brain regions across 106 subjects, with each region including a different subset of subjects. Each dot represents one brain region in the iEEG data. Horizontal lines represent chance level (0.25). **$p < 0.001$.

**Table 1 | The statistical results of the post-hoc Dunn's test to the Friedman test of the difference in exponent values of the knee model between sleep stages in iEEG**

| Pairwise | z | p | cd [95% CI] |
|---|---|---|---|
| Wake vs N2 | 5.45 | <0.001 | 0.79 [0.76 0.82] |
| Wake vs N3 | 5.74 | <0.001 | 0.82 [0.79 0.85] |
| Wake vs REM | 5.82 | <0.001 | 0.77 [0.74, 0.80] |
| N2 vs N3 | 0 | 1 | 0.04 [−0.01 0.10] |
| N2 vs REM | 0 | 1 | 0.07 [0.02 0.12] |
| N3 vs REM | 0 | 1 | 0.03 [−0.02 0.08] |

*z* z-value, *p* Bonferroni-corrected *p*-values, *cd* Cliff's delta effect size.

**Table 2 | The statistical results of the post-hoc Dunn's test to the Friedman test of the difference in knee frequencies between sleep stages in iEEG**

| Pairwise | z | p | cd [95% CI] |
|---|---|---|---|
| Wake vs N2 | 6.18 | <0.001 | 0.94 [0.92, 0.97] |
| Wake vs N3 | 8.21 | <0.001 | 0.99 [0.98, 1.00] |
| Wake vs REM | 3.87 | 0.001 | 0.73 [0.68, 0.78] |
| N2 vs N3 | 3.76 | 0.001 | 0.78 [0.73, 0.83] |
| N2 vs REM | 2.31 | 0.13 | 0.47 [0.39, 0.55] |
| N3 vs REM | 6.07 | <0.001 | 0.86 [0.82, 0.90] |

Upper bound of CI clipped at 1.00 because Cliff's δ cannot exceed 1.
*z* z-value, *p* Bonferroni-corrected *p*-values, *cd* Cliff's delta effect size.

**Table 3 | The statistical results of the post-hoc Dunn's test to the Friedman test of the difference in exponent values of the fixed model between sleep stages in iEEG**

| Pairwise | z | p | cd [95% CI] |
|---|---|---|---|
| Wake vs N2 | 5.22 | <0.001 | 0.92 [0.88, 0.95] |
| Wake vs N3 | 8.21 | <0.001 | 1 [0.98, 1.00] |
| Wake vs REM | 0 | 1 | 0.26 [0.16, 0.36] |
| N2 vs N3 | 3.70 | <0.001 | 0.98 [0.96, 1.00] |
| N2 vs REM | 4.02 | <0.001 | 0.84 [0.79, 0.89] |
| N3 vs REM | 8.21 | <0.001 | 0.99 [0.97, 1.00] |

Upper bound of CI clipped at 1.00 because Cliff's δ cannot exceed 1.
*z* z-value, *p* Bonferroni-corrected *p*-values, *cd* Cliff's delta effect size.

**Table 4 | The statistical results of the post-hoc Dunn's test for sleep-stage differences in pre-knee exponent in iEEG data**

| Pairwise | z | p | cd |
|---|---|---|---|
| Wake vs N2 | 6.08 | <0.001 | 0.92 [0.90, 0.94] |
| Wake vs N3 | 8.21 | <0.001 | 0.99 [0.98, 1.00] |
| Wake vs REM | 1.94 | 0.05 | 0.56 [0.52, 0.60] |
| N2 vs N3 | 2.34 | 0.02 | 0.64 [0.60, 0.68] |
| N2 vs REM | 3.26 | 0.001 | 0.69 [0.66, 0.72] |
| N3 vs REM | 6.41 | <0.001 | 0.92 [0.90, 0.94] |

Upper bound of CI clipped at 1.00 because Cliff's δ cannot exceed 1.

negative difference (knee - Fixed) values (Wake: −4.71, N2: −1.82, N3: −4.04, REM: −4.77). Additionally, to identify the stronger predictor of sleep stages (knee frequency vs exponent) within the knee model, we conducted a random forest analysis followed by permutation importance analysis (Fig. 4J). Results showed that the knee frequency (permutation importance: $0.14 \pm 0.07$) is significantly more important than the spectral exponent (permutation importance: $0.07 \pm 0.05$) ($t(15) = −5.13$, $p < 0.001$, $d = 1.06$), suggesting that within the knee model the knee frequency serves as a more informative predictor of sleep stages than the exponent.

### Stage-dependent patterns in EEG aperiodic activity

Based on the results of the iEEG analysis, we next examined sleep-related aperiodic activity in a full night recording of EEG, as most previous results are in extracranial data. Starting again with a visual inspection, we saw that the PSDs across distinct sleep stages revealed differences in the exponent across stages, as well as a stage-dependent knee (Fig. 5A). Specifically, we observed a prominent knee during REM sleep that appears to be attenuated in wakefulness, N1 and N2, and looks to be absent during N3. Figure 5B illustrates PSDs for different sleep stages, across participants, emphasizing the consistency of stage differences across all regions, and highlighting the consistency of the knee parameter in REM sleep.

We next fit spectral models to all 183 EEG electrodes, with the goal of replicating the analyses from the iEEG data, in order to evaluate the effectiveness of the exponent and knee parameters in differentiating between sleep stages in EEG data (Fig. 5C, D). Using the knee model, we observed significant differences in the exponent across sleep stages, similar to the findings from the iEEG data (Fig. 5C left; $X^2 = 58.21$, $p < 0.001$, $W = 0.86$). Specifically, we observed an increase in the exponent from wakefulness to sleep, with further variations observed between different sleep stages. The detailed outcomes of the post-hoc tests can be found in Table 5. Similarly, the knee frequency differed significantly across sleep stages (Fig. 5C, right; $\chi^2 = 16.48$, $p < 0.001$, Kendall's $W = 0.41$). Post-hoc comparisons via Dunn's test revealed the most significant differences between N3 and Wake ($p = 0.02$), N3 and N2 ($p = 0.004$), and N3 and REM ($p = 0.03$), highlighting

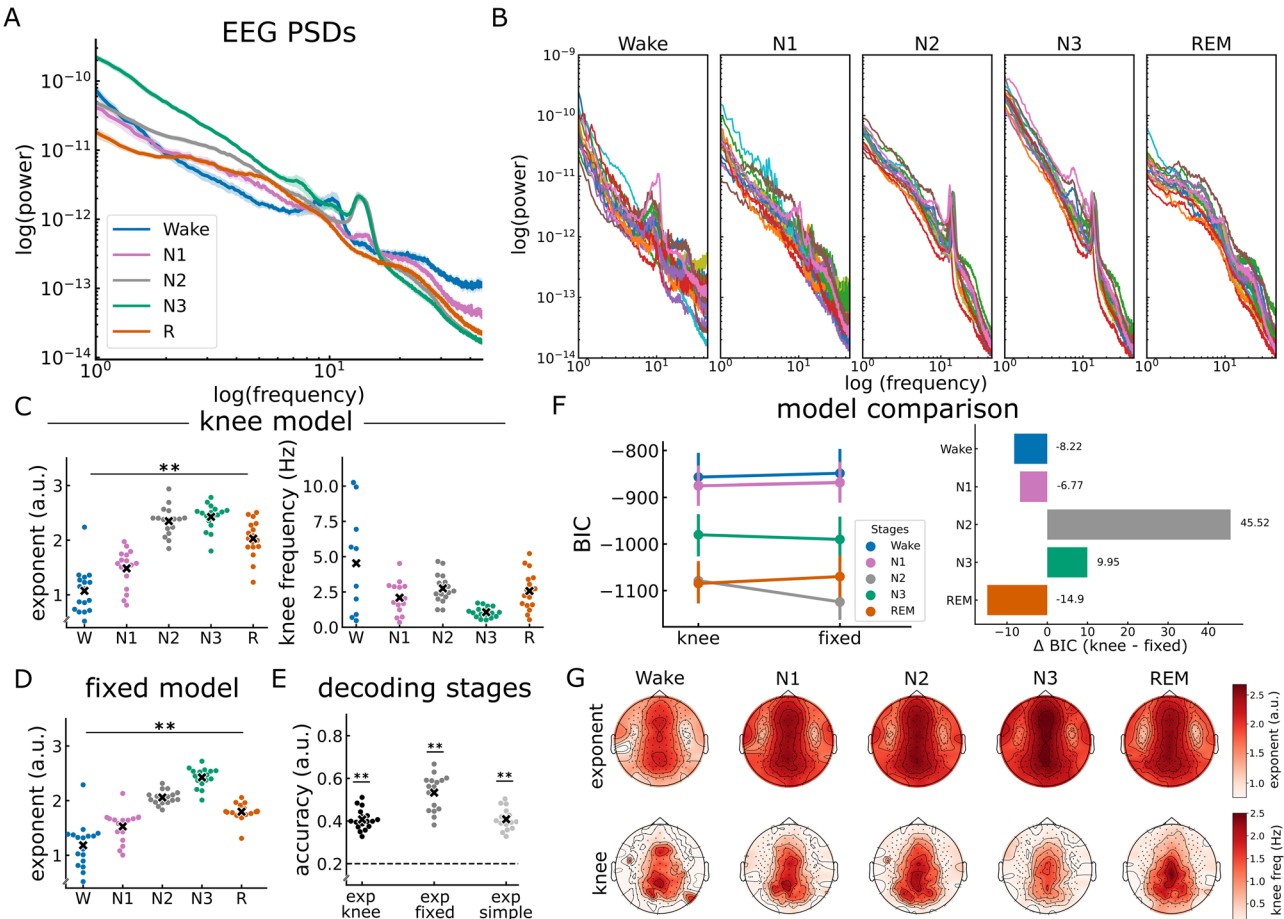

**Fig. 5 | Aperiodic activity in EEG sleep data. A** PSDs of the different sleep stages calculated at electrode Cz. **B** The PSDs of the different stages per subject measured at electrode Cz. Note the stage-dependent observation of the 'knee' in the PSD. **C** Exponent values (left) and knee frequency values (right) of the knee model (fit range: 1–45 Hz) across sleep stages. **D** Exponent values of the fixed model across sleep stages (fit range: 1–45 Hz). The exponents of both the knee model and the fixed model significantly differed between sleep stages, while the knee frequency did not. **E** Classification accuracy of sleep stages based on the EEG exponent of the different sleep stages using an LDA classifier trained on the fixed model exponent, the knee model exponent, or the exponent of a fixed model with a narrow band (30–45 Hz). Regardless of the frequency range and the model used, the classifier was able to use the exponent to classify between the stages with an above chance (0.2) accuracy.

**F** BIC comparison between the knee model and the fixed model suggesting that the knee model provided a better fit than the fixed model in a stage-dependent manner. The left panel presents absolute BIC values for both models across sleep stages, while the right panel shows their difference (knee - fixed). Negative values indicate that the knee model outperforms the fixed model. Note the differences of the preferred model across stages. Dots represent mean values, and lines indicate 95% confidence intervals. **G** Topographical maps depicting the spatial distribution of the exponent of the knee model (top) and the knee frequency (bottom) across the scalp. Note the consistency of the central topography of the exponent and the knee frequency across stages. We used EEG data from 17 healthy participants during overnight sleep. Each dot represents one participant of the EEG dataset. Horizontal lines represent chance level (0.2). *$p < 0.05$, **$p < 0.001$.

the significantly low knee frequency in N3. For detailed post-hoc statistics, see Table 6. When applying a fixed model (fit range 1–45 Hz), the exponent also showed a significant difference between sleep stages (Fig. 5D; Friedman chi-square test: $X^2 = 61.93$, $p < 0.001$, $W = 0.91$). The results of the post-hocs test are detailed in Table 7. The $R^2$ of the various models for different sleep stages are depicted in Supplementary Fig. 10A, B.

To investigate how aperiodic parameters vary across sleep stages, we examined the classification accuracies using the different exponent estimates with EEG electrodes as features, including the exponent of the a) knee model, b) fixed model broad range, and c) fixed model narrow range (30–45 Hz), which was added to compare to previous reports. The results showed that the exponents of all models performed significantly higher than chance levels (Fig. 5E; exp_knee: $t(16) = 17.47$, $p_{bonf} < 0.001$, $d = 5.99$, exp_fix_broad: $t(16) = 16.87$, $p_{bonf} < 0.001$, $d = 5.79$, exp_fix_narrow: $t(16) = 17.65$, $p_{bonf} < 0.001$, $d = 6.05$). Comparing the accuracies of these three parameters demonstrated a significant difference between the accuracy values from the different models (Friedman chi-square test: $X^2 = 25.53$, $p < 0.001$, $W = 0.75$). Post-hoc comparisons via Dunn's test revealed that the exponent of the fixed model (broad range) performed significantly higher

than that of the knee model ($z(16) = 4.03$, $p_{bonf} < 0.001$, $d = 0.81$), and the exponent fit over the 30–45 (narrow) range ($z(16) = 3.93$, $p_{bonf} < 0.001$, $d = 0.76$). There was no statistically significant difference between the exponent of the knee model and that of the narrow range ($z(16) = 0.1$, $p_{bonf} = 1$, $d = 0.02$).

Similar to our analysis of iEEG data, we computed the BIC difference between the knee model and the fixed model in EEG data (Fig. 5F). The results revealed a sleep stage-dependent performance for these models. Specifically, we calculated the BIC differences between the knee and fixed models (Knee - Fixed) in EEG data. The results indicate strong evidence for a better fit of the knee model during Wakefulness (−8.22), N1 (−6.77), and REM sleep (−14.9). In contrast, the fixed model provided a better fit during N2 (45.52) and N3 (9.95), as reflected by lower BIC values. Further, Spearman correlation analysis while controlling for sleep stages (Supplementary Fig. 10C, D) revealed a significant negative correlation between the knee frequency and the exponent of the fixed model (between models; rho(73) = −0.31, $p = 0.006$, 95% CI = [−0.5, −0.09]), consistent with findings from iEEG data. However, we found no significant correlation between the knee frequency and the exponent of the knee model (within-model;

**Table 5 | The statistical results of the post-hoc Dunn's test to the Friedman test of the difference in exponent values of the knee model between sleep stages in EEG**

| Pairwise | z | p | cd [95% CI] |
|---|---|---|---|
| Wake vs N1 | <0.001 | 1 | 0.62 [0.47, 0.77] |
| Wake vs N2 | 5.23 | <0.001 | 0.96 [0.89, 1.00] |
| Wake vs N3 | 6.07 | <0.001 | 0.98 [0.92, 1.00] |
| Wake vs REM | 3.30 | 0.001 | 0.89 [0.79, 0.99] |
| N1 vs N2 | 3.77 | <0.001 | 0.99 [0.93, 1.00] |
| N1 vs N3 | 4.64 | <0.001 | 0.99 [0.93, 1.00] |
| N1 vs REM | 1.63 | 0.10 | 0.77 [0.63, 0.91] |
| N2 vs N3 | <0.001 | 1 | 0.29 [0.08, 0.50] |
| N2 vs REM | 0.24 | 0.81 | 0.52 [0.33, 0.71] |
| N3 vs REM | 1.58 | 0.11 | 0.70 [0.53, 0.87] |

Upper bound clipped of CI at 1.00 because Cliff's δ cannot exceed 1.
z z-value, p Bonferroni-corrected p-values, cd Cliff's delta effect size.

**Table 6 | The statistical results of the post-hoc Dunn's test to the Friedman test of the difference in knee frequency between sleep stages in EEG**

| Pairwise | z | p | Cd [95% CI] |
|---|---|---|---|
| Wake vs N1 | <0.001 | 1 | 0.30 [0.08, 0.52] |
| Wake vs N2 | <0.001 | 1 | 1 [0.92, 1.00] |
| Wake vs N3 | 2.40 | 0.02 | 1 [0.92, 1.00] |
| Wake vs REM | <0.001 | 1 | 1 [0.92, 1.00] |
| N1 vs N2 | <0.001 | 1 | 0.1 [−0.12, 0.32] |
| N1 vs N3 | 1.14 | 0.25 | 0.63 [0.43, 0.83] |
| N1 vs REM | <0.001 | 1 | 0.02 [−0.20, 0.24] |
| N2 vs N3 | 3.10 | 0.002 | 0.92 [0.77, 1.00] |
| N2 vs REM | <0.001 | 1 | 0.13 [−0.09, 0.35] |
| N3 vs REM | 2.34 | 0.02 | 0.73 [0.53, 0.93] |

Upper bound of CI clipped at 1.00 because Cliff's δ cannot exceed 1.
z z-value, p Bonferroni-corrected p-values, cd Cliff's delta effect size.

**Table 7 | The statistical results of the post-hoc Dunn's test to the Friedman test of the difference in exponent values of the fixed model between sleep stages in EEG**

| Pairwise | z | p | cd |
|---|---|---|---|
| Wake vs N1 | <0.001 | 1 | 0.62 [0.57, 0.67] |
| Wake vs N2 | 4.68 | <0.001 | 0.89 [0.85, 0.92] |
| Wake vs N3 | 6.86 | <0.001 | 0.98 [0.96, 0.99] |
| Wake vs REM | 2.29 | 0.02 | 0.83 [0.79, 0.87] |
| N1 vs N2 | 3.28 | 0.001 | 0.91 [0.87, 0.94] |
| N1 vs N3 | 5.54 | <0.001 | 0.99 [0.97, 1.00] |
| N1 vs REM | 0.37 | 0.71 | 0.78 [0.73, 0.82] |
| N2 vs N3 | 0.84 | 0.40 | 0.88 [0.84, 0.92] |
| N2 vs REM | 0.87 | 0.38 | 0.88 [0.84, 0.92] |
| N3 vs REM | 3.56 | <0.001 | 0.99 [0.97, 1.00] |

Upper bound clipped of CI at 1.00 because Cliff's δ cannot exceed 1.
z z-value, p Bonferroni-corrected p-values, cd Cliff's delta effect size.

rho(73) = −0.0007, $p = 0.99$, 95% CI = [−0.23, 0.23]). Overall, these analyses of model forms and aperiodic parameter relationships to sleep stages suggest a difference in prominence of the aperiodic knee as compared to the intracranial data – while there are observable knees in the EEG data, with differences across stages, the variability of this parameter is such that the knee frequency is less predictive of different sleep stages.

We also examined the aperiodic parameters across the scalp, whereby the topographies for the exponent of the knee model and the knee frequency across sleep stages show a consistent spatial pattern of the largest exponent and knee frequencies around the central and posterior electrodes, with this topography being consistent across sleep stages (Fig. 5G; see Supplementary Fig. 10E for the topography of the fixed model).

## Time-resolved EEG exponent tracks sleep architecture

Building on the observed differences in aperiodic parameters across sleep stages, our next aim was to explore the temporal dynamics of aperiodic activity during sleep (Fig. 6A). Using the knee model, we examined changes of the exponent across time. To do so, we extracted the exponent at electrode Cz for each epoch within each sleep stage, perserving the chronological order of epochs across the entire night. We then analyzed these stage-specific, time-ordered exponents using separate regression models for each sleep stage (Fig. 6B). The findings revealed distinct temporal patterns in the exponent values across different sleep stages. During wakefulness, there was a notable increase in the exponent value as the night advanced. A significant portion of this increase could be attributed to the passage of time ($R^2 = 0.83$, $F(1,11) = 55.1$, $p < 0.001$). Conversely, we observed a significant effect of time on the decrease in the exponents during N3 and REM (N3: $R^2 = 0.26$, $F(1,129) = 46.66$, $p < 0.001$, REM: $R^2 = 0.07$, $F(1,90) = 7.24$, $p = 0.008$). For N1, although there was an upward trend in the exponent values, the change was not statistically significant ($R^2 = 0.13$, $F(1,12) = 1.76$, $p = 0.21$), while during N2 sleep, the exponent values showed no significant change ($R^2 = 0.002$, $F(1,190) = 0.3$, $p = 0.58$).

To further examine the fluctuations in aperiodic activity over the course of the night, we assessed the changes in the exponent by dividing epochs for each sleep stage into quartiles and comparing exponent values across these quartiles (Fig. 6C). In this analysis, the exponent during REM decreased significantly over quartiles of the night (Friedman chi-square test: $X^2 = 13.31$, $p = 0.004$, $W = 0.26$), while the exponent of N1 increased significantly across the night (Friedman chi-square test: $X^2 = 8.65$, $p = 0.001$, $W = 0.17$). There was no statistically significant difference for Wake (Friedman chi-square test: $X^2 = 3.21$, $p = 0.36$, $W = 0.06$), N2 (Friedman chi-square test: $X^2 = 2.29$, $p = 0.51$, $W = 0.04$), or N3 (Friedman chi-square test: $X^2 = 6.04$, $p = 0.11$, $W = 0.12$).

Next, we evaluated the temporal precision of the changes in aperiodic activity by analyzing time-resolved exponent values during transitions between sleep stages, as identified by our sleep scoring algorithm. To gain insights into how the exponent values change during these transitions, we compared them to a baseline condition of equal length where no change in sleep stage occurred. We specifically focused on the most prevalent transitions during sleep, including N1 to N2, N2 to N1, and N2 to N3, as indicated by the transition matrix (Supplementary Fig. 11). Comparing the transitions from N1 to N2 against a baseline of continuous N1, i.e. two consecutive N1 epochs (Fig. 6D), we found that the exponent increased shortly after the transition from N1 to N2 reaching a significant difference to that of the baseline starting at 24 s following the transition (28.71 ± 10.02 trials - 24–60 s: $\sum t(16) = 87.82$, $p < 0.001$, $d = 1.38$). Similarly, when examining transitions from N2 to either N3 or N1 (Fig. 6E), the results revealed a significant decrease in the exponent 18 s after transition from N2 to N1 (17.59 ± 7.1 trials - $\sum t(16) = 164.24$, $p < 0.001$, $d = 1.58$), and a significant increase in the exponent 14 s after the transition from N2 to N3 (22.18 ± 6.81 trials - 14–30 s: $\sum t(16) = 33.19$, $p = 0.003$, $d = 1.38$).

Furthermore, we sought to examine a pivotal transition in sleep that marks a significant change in brain activity: the shift from NREM sleep to

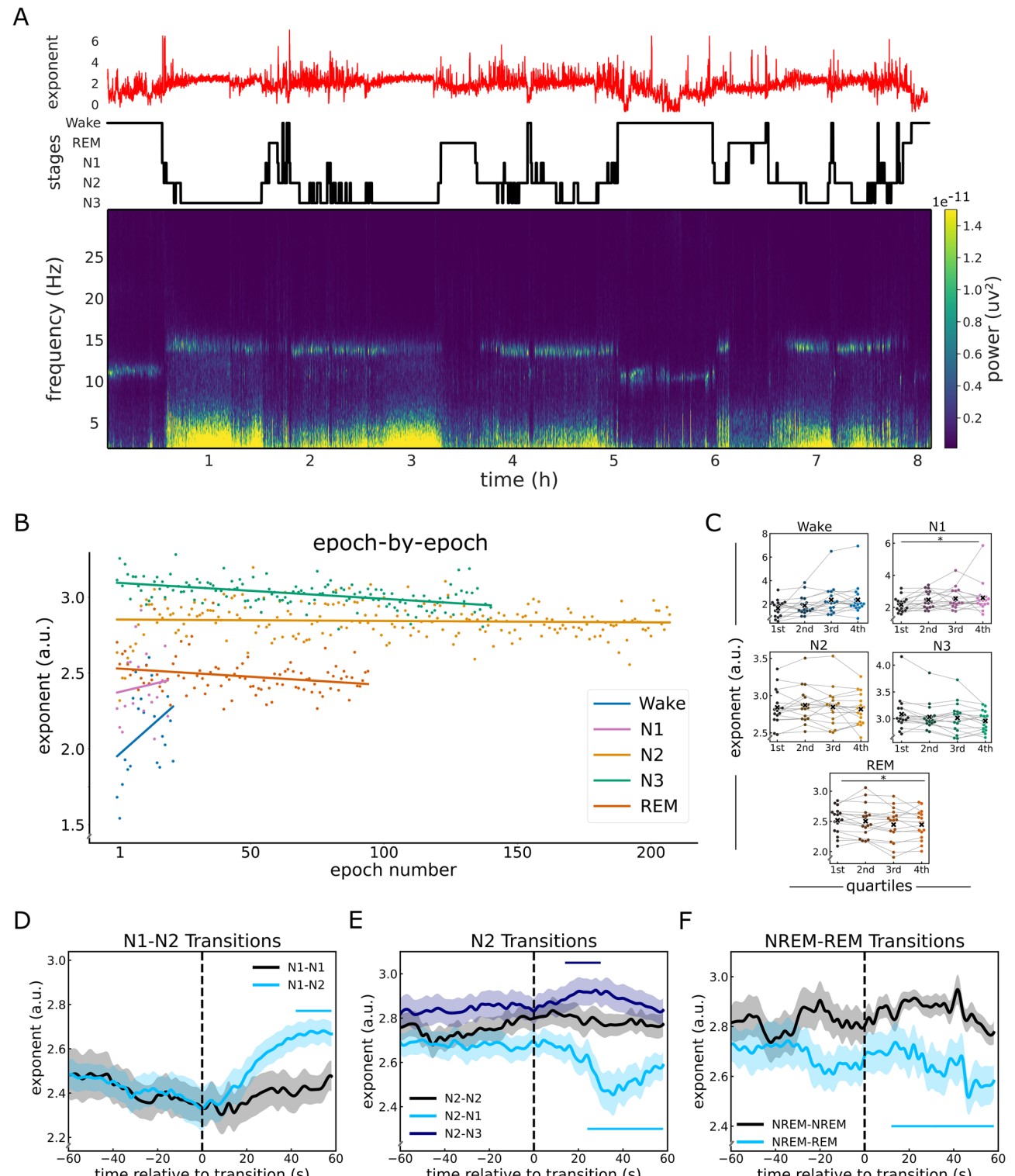

**Fig. 6 | EEG exponent tracks the dynamics of sleep architecture. A** Temporally resolved estimates of the spectral exponent, overlaid on top of the sleep staging and spectral plot for the entire night, taken from an example subject. **B** Epoch-by-epoch exponent values from across the night for each sleep stage. Notably, Wake and N1 show an increase in the exponent as the night progresses, while N3 and REM show a decrease, and N2 shows no significant change. **C** Stage-specific exponent values split across the quartiles of the night, where each dot represents a subject. The exponent shows no significant differences across the quartiles, except for N1 where it increases and REM sleep where it decreases significantly across the night. **D–F** Time-resolved estimates of the spectral exponent during sleep stage transitions. Each transition is compared to a control period of adjacent epochs when no change in sleep stage occurred. **D** Sleep stage

transition from N1, showing an increase in the exponent in the transition from N1 to N2 sleep stages, which is significant starting 24 s after transition. **E** Sleep stage transitions from N2 to either N1 or N3. Post-transition from N2 to N1, the exponent significantly decreased after 18 s. Conversely, transitioning from N2 to N3 led to an increase in the exponent, this change was significantly different from the period of uninterrupted N2 sleep during a short time window (14–30 s). **F** Transitions from NREM (N2 and N3) to REM sleep, showing a significant decrease in exponent starting 18 s after the transition. All exponent values in this analysis reflect the exponent of the knee model (fit range: 1–45 Hz). We used EEG data from 17 healthy participants during overnight sleep. Vertical dashed lines in (**D–F**) indicate the transitions between stages. Horizontal lines indicate significant clusters.

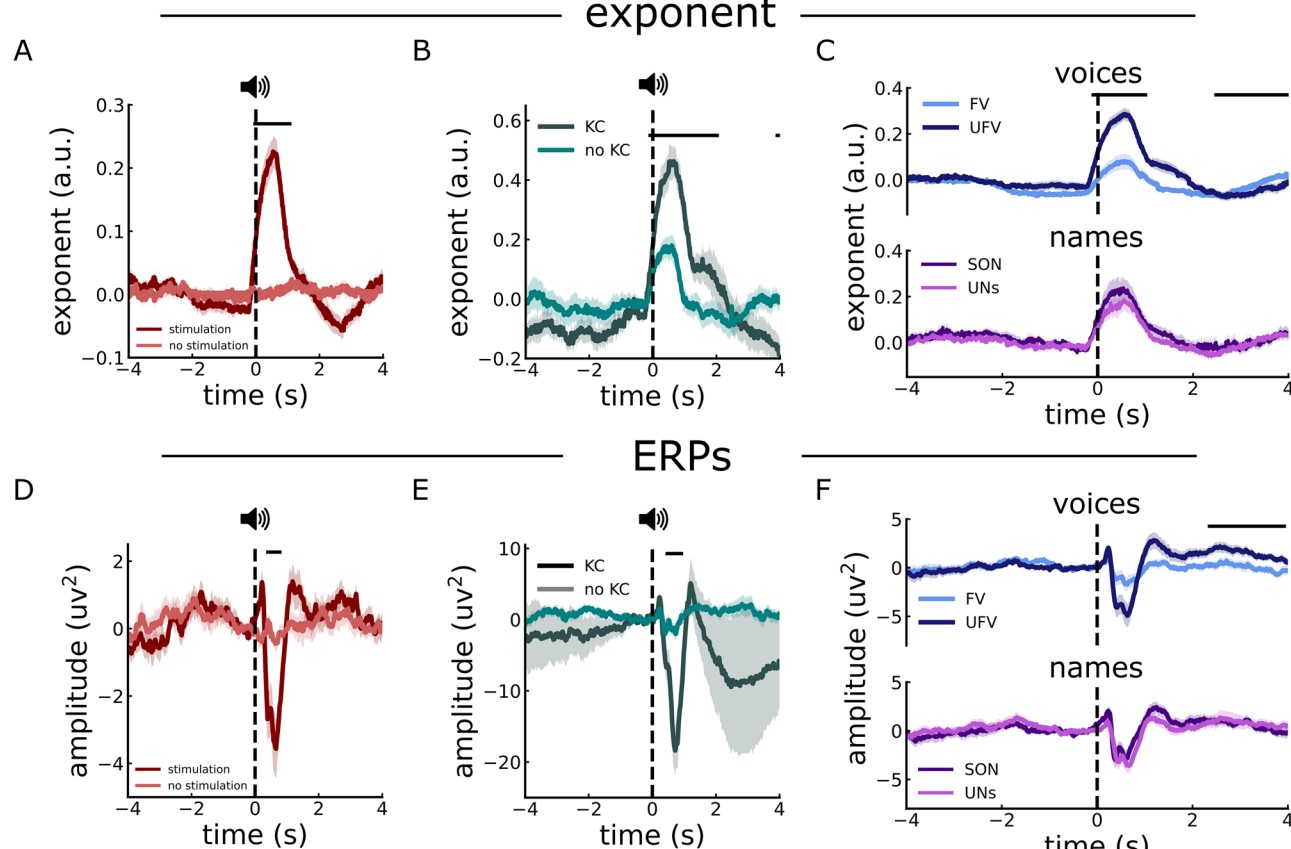

**Fig. 7 | Auditory-evoked exponent and ERP responses during sleep. A** Time-resolved estimates of aperiodic exponent in response to auditory stimuli (blue) compared to data segments with no stimuli (orange) during NREM sleep (N2 and N3), showing a transient, time-locked increase in exponent values during stimulus presentation. **B** Comparison of auditory-evoked exponent changes for trials with and without elicited K-complexes during N2. **C** Auditory-evoked exponent responses split by auditory stimuli, comparing unfamiliar voices (UFV) and familiar voices (FV), as well as one's own name (SON) and unfamiliar names (UNs). Aperiodic exponent responses differ by voice familiarity, but not by stimulus content. **D** Same as in (**A**) but for ERP responses, showing temporal responses to sounds during NREM sleep. **E** Same as in (**B**) but for ERP responses, comparing stimulus trials with and without evoked K-complexes. **F** Same as in (**C**) but for ERPs, comparing responses across stimulus conditions, showing ERP responses also differentiate the familiarity of voice. We used EEG data from 17 healthy participants during overnight sleep. Across all panels, the onset of the stimulus is marked by dashed vertical lines, and clusters representing significant differences between plotted measures are indicated by solid horizontal lines.

REM sleep (Fig. 6F). For this purpose, we analyzed the exponent during the transition from NREM (either N2 or N3) sleep to REM sleep, contrasting it with a baseline period of uninterrupted NREM sleep. The results demonstrate a significant decrease in the exponent during the NREM-to-REM transition as compared to the NREM baseline (4 ± 1.53 trials - 16–60 s: $\sum t(16) = 73.39$, $p = 0.002$, $d = 1.12$).

Overall, these results align with the expected changes in the aperiodic exponent observed in non-time-resolved analyses and highlight the temporal precision of these changes across sleep stages. Additionally, we conducted time-resolved analyses of sleep stage transitions using both the exponent of the fixed model and the knee frequency (see Supplementary Figs. 12 and 13). The results are broadly consistent, and we found no statistically significant difference in the model fits ($R^2$ values) between the knee and fixed models (Wilcoxon sign-rank test: $t(16) = 1.885$, $p = 0.108$, $d = 0.22$), suggesting that the two models may capture complementary aspects of stage-dependent spectral dynamics rather than differing in overall fit quality.

**Selective EEG exponent responses to auditory stimuli during sleep**

Next, we aimed to investigate how the exponent responds to external events by analyzing evoked responses to auditory stimuli. To do so, we first computed and analyzed time-resolved measures of aperiodic activity time locked to auditory stimuli, as compared to baseline periods of equal length

where no stimuli were presented. During NREM (N2 and N3) sleep, we observed a significant increase in the exponent following stimulus presentation (Fig. 7A, −0.08 s to 1.12 s: $\sum t(15) = 1029.54$, $p < 0.001$, $d = 1.94$). It's noteworthy that the average duration of these stimuli was 808 ms, suggesting that the dynamics of the transient exponent matched the time course of the stimulus presentation. In contrast, during REM sleep, there was no statistically significant change in the exponent following the presentation of stimuli (Supplementary Fig. 14A). To ensure the validity of the observed changes in the exponent values and to confirm that these changes were genuine and not a result of poor model performance, we assessed the $R^2$ values around the time of stimulus presentation. The $R^2$ values were found to be high (all $R^2 > 0.99$) and stable across time (Supplementary Fig. 14B, C).

We noted that the aperiodic responses we observed were similar to the typical auditory-evoked KC responses that have also been reported during NREM sleep[48]. To evaluate how these transient aperiodic responses relate to KC activity, we compared the changes in the exponent in trials where a KC was evoked with those where no KCs were detected (Fig. 7C). The results indicated that trials in which a KC was evoked showed a significantly larger change in the exponent compared to trials without a KC (−0.14 to 2.07; $\sum t(16) = 789.86$, $p < 0.001$, $d = 1.99$). This suggests a possible association between the aperiodic evoked response and the auditory-evoked KC. Indeed, when we conducted a comparison between the PSDs at the peak of the exponent responses when KCs were elicited and when no KCs were elicited, we observed a broadband power difference indicating an exponent

shift (Supplementary Fig. 15, 1–13 Hz: $\sum t(16) = 234.54$, $p < 0.001$, $d = 1.23$; 15–45 Hz: $\sum t(16) = 103.06$, $p = 0.003$, $d = 1.35$). Thus, an aperiodic shift in neural activity might go hand in hand with the appearance of a KC – although we still observed exponent responses in trials where no evoked KC was detected, suggesting that these two phenomena do not fully overlap.

Additionally, we examined whether these evoked responses of the aperiodic exponent are sensitive to different types of stimuli. Our investigation encompassed a range of stimulus categories, particularly contrasting familiar voices (FV) with unfamiliar voices (UFV), and the subject's own name (SON) against unfamiliar names (UNs) during both NREM and REM sleep. During NREM sleep, the analysis indicated a significantly larger response of the exponent following the presentation of UFVs as compared to FVs (Fig. 7D-Top, −0.14 s to 1.03 s: $\sum t(16) = 577.26$, $p = 0.003$, $d = 1.19$). Subsequently, the exponent for UFVs decreased to be significantly lower than FVs (2.44 s to 4 s: $\sum t(16) = 793.48$, $p = 0.001$, $d = 1.06$). No discernible difference was found when comparing the exponent's response to the SON and UNs (Fig. 7D-Bottom). During REM sleep, the responses did not significantly differ between either the different voices or the names (Supplementary Fig. 7–1E).

Finally, we also investigated time-domain responses by examining the auditory ERPs for all auditory stimuli, as well as for the different categories of stimuli. The analysis revealed negative peaks in response to auditory stimuli during NREM sleep (Fig. 7E, $\sum t(16) = -249.72$, $p = 0.003$, d = 0.9), but no difference to no-stimulation periods in REM sleep (Supplementary Fig. 7–1D). The ERPs exhibited a stronger negative peak in trials that triggered a KC (Fig. 7G, 0.41–0.96 s: $\sum t(16) = -249.72$, $p = 0.003$, d = −1.03). Further, while we observed a negative peak following UFV presentations compared to FVs between 0.45 s and 0.72 s, it lacked statistical significance (Fig. 7H-top; $\sum t(16) = -151.97$, $p = 0.12$, $d = 0.56$). However, a significant positive deflection was noted following UFVs in the duration from 2.23 s to 3.96 s ($\sum t(16) = 1293.46$, $p < 0.001$, $d = 0.89$). No significant variations were observed in the responses to different names (Fig. 7H-bottom). Last but not least, ERPs during REM showed no difference between either voices or names (Supplementary Fig. 7–1F). Taken together, our findings once more point to a potential overlap between measures of ERPs, KCs, and the aperiodic exponent in neural activity during sleep, though with some notable distinctions between them. In particular, we observe stimulus-related responses in the exponent when no KCs are detected, when there is also no clear ERP response, suggesting at least some independence between the different features.

## Discussion

Recent investigations of electrophysiological data have established that aperiodic neural activity offers information about brain function beyond what is available using conventional approaches that focus primarily on periodic, oscillatory aspects[27,58]. This includes research establishing that aperiodic activity varies systematically across the sleep cycle[7,9,12–14,16]. In this exploratory work, we sought to extend the investigation of aperiodic activity during sleep, by examining it across modalities (EEG and iEEG), frequency ranges, model forms, and across time. In doing so, we were able to replicate previous findings of variations of aperiodic activity during sleep, while extending these results to demonstrate how multiple aperiodic features show variations with temporal dynamics that track sleep stage transitions and neural responses to external stimulation. Overall, these findings suggest that by improving our approaches for measuring aperiodic activity, we can capture aperiodic fluctuations that reflect both macro- and micro-structures of sleep, offering a more comprehensive view of sleep dynamics.

In reviewing the previous literature on aperiodic activity during sleep, we noted that previous studies have typically used a variety of different frequency ranges to estimate aperiodic activity, with no clear consensus. We therefore investigated the influence of varying the frequency range for model fitting. Overall, we observed a high model fit ($R^2 > 0.86$) across all ranges, however, when using narrow frequency ranges, there was a trend towards i) lower model fit quality, ii) more dependency on the length of the

time window, and iii) higher variability in model fit quality and parameter results. This suggests that while different frequency ranges can be validly examined, choosing broader ranges may offer more stable estimates. Indeed, the use of broader frequency ranges led to reduced variance in the derived parameters, suggesting increased reliability. Thus, we suggest that future studies of aperiodic activity during sleep should explicitly indicate which fitting ranges are used, consider fitting broader ranges (>20 Hz bandwidth and encompassing lower frequencies) where appropriate, and, where possible, include sensitivity analysis examining the dependency of findings on the fitting range.

Another notable aspect of the previous literature is the use of 'fixed' exponent models, which estimate the spectral exponent by fitting the equivalent of a straight line in the log-log space within a narrow frequency range, typically between 30 and 50 Hz[14,22]. While this approach has proven effective in distinguishing between sleep stages[12–14], it does not capture the breadth of aperiodic activity, which by definition extends across all frequencies. Notably, fitting a single exponent to a narrow band range avoids detecting a 'knee' or bend in the PSD, after which the exponent changes[4,30], which can provide additional information beyond the single-exponent model[9]. By examining broader frequency ranges, such analyses can capture more variance in the data, including of slower frequencies which are particularly relevant during sleep, and can also explicitly measure the knee frequency and what it reflects. Accordingly, we sought to build on previous work that investigated the narrowband exponent to explore sleep dynamics and differentiate sleep stages by extending the frequency range and explicitly incorporating the aperiodic knee —an approach that may provide a more complete and physiologically grounded representation of the underlying neural dynamics. Notably, selection of frequency ranges and model forms should be considered together – for example, examining whether fitting a broader frequency range would require fitting a knee parameter – as qualitative differences in the structure of the data may impact model fitting, even if quantitative measures of goodness-of-fit such as $R^2$ are similar.

By applying a spectral parameterization approach that examines broad frequency ranges and fits an aperiodic model with a knee, we demonstrate differences in aperiodic activity between sleep stages that go beyond what can be captured by a single-exponent (fixed) model. For instance, in the intracranial data, our findings demonstrate that when fitting a knee model, the knee frequency is more effective than the exponent at distinguishing between sleep stages. This suggests that, at least in this dataset, a prominent change across sleep stages is primarily in the knee frequency, whereby this can look like an exponent change when using a single-exponent model. Further supporting this, we observed a negative correlation between the knee frequency (in the broad range knee model) and the exponent of the fixed model, both in iEEG and EEG data. Moreover, the comparable EEG topographies observed for both the knee frequency and the exponent may also imply overlapping underlying processes. Overall, while a single exponent fit within a narrow band frequency may suffice if the primary goal is to differentiate between sleep stages, the underlying changes in the data may be better captured by fitting more complex models over broader frequency ranges. Although our approach goes beyond a single exponent model, it is also important to acknowledge that the Lorentzian function we fit assumes a near-zero slope below the knee, potentially oversimplifying low-frequency dynamics that also appear to be sleep stage-specific. Future work should aim to incorporate models capable of estimating the knee, as well as both pre- and post-knee exponents to more accurately characterize the full aperiodic signal.

Notably, the presence of a knee in electrophysiological signals is not universal, particularly in EEG in which very little research has sought to measure a knee as it is usually assumed to be absent. In the EEG dataset examined here, a knee is most prominently visible during REM sleep, while being somewhat visible during Wake and N1, and seemingly absent during NREM stages N2 and N3. The seemingly variable presence of the knee in the EEG data introduces a model-selection challenge, whereby no single model definition is best across all sleep stages in EEG. This underscores that while both the knee and the fixed models might be suitable for a qualitative

differentiation between sleep stages, an informed model-selection decision is crucial for accurately and quantitatively investigating the specific neural activity patterns within each stage.

In terms of interpretations of aperiodic neural activity, previous investigations which have largely focused on changes in the aperiodic exponent, have typically interpreted changes in this parameter in terms of its putative relationship to the excitation-inhibition (E-I) balance, whereby a steeper slope signifies an increase in inhibition, and conversely, a flatter slope indicates an increase in excitation[21,22]. The pattern of changes in previous studies is consistent with a general shift towards more inhibition during the transition from wakefulness to sleep[25,26], with the exponent becoming steeper from wakefulness to NREM to REM sleep[14,15]. While our findings partially align with previous observations, demonstrating that the exponent differs significantly across the sleep-wake cycle, the direction of these differences varies depending on the model used. Specifically, when fitting a fixed exponent model to both iEEG and EEG sleep data, we observed a progressive steepening of the exponent from wakefulness to sleep. In contrast, iEEG data analyzed with the knee model showed a decrease in the exponent from wakefulness to sleep, with no further changes observed during sleep.

This discrepancy reflects distinct features of the data that are captured by the knee and fixed models. The relationships of the aperiodic features to each other, within and between the different models, and their differing patterns across stages emphasize that these different models and features are related, but not equivalent. This relates to the impact and importance of the knee – while a true 1/f signal has the same exponent value across all frequency ranges (which can also be called mono-fractal or scale-free), the knee reflects a frequency-specific transition in the power spectrum. In such a case, which can be called multi-fractal or multi-scale, the knee parameter reflects a transition between 1/f regions, and the exponent of the knee model reflects the frequency region beyond the knee frequency. Even when models (knee vs single-exponent) are fit across the same frequency range, their exponents show only weak non-significant correlation, further confirming that they capture different spectral properties and may lead to distinct interpretations. This suggests that aperiodic parameters cannot be interpreted in isolation, for example, the exponent from a knee model should be considered in relation to the measured knee and may differ from the exponent of a fixed model. This also implies that the association between the exponent and E/I balance is necessarily more nuanced than a one-to-one mapping, as different frequency ranges can have different measured exponent values, which may relate to E/I balance in different ways. While the results here support the use of the knee model, further work is needed to relate results and interpretations of the aperiodic exponent measures of these different models, as well as in comparison to previous work that typically examined narrower ranges.

In applying the knee model, this study highlights notable changes in the knee parameter across sleep stages, which provide distinct insights into neural dynamics. The knee frequency, reflecting the "timescale" of neural processing, maps directly to the decay time constant from the autocorrelation function[4,30]. N3 sleep exhibited the lowest knee frequency, indicating the longest processing timescale, while REM sleep and wakefulness showed higher knee frequencies, suggesting faster processing. These findings align with prior sleep-related autocorrelation analyses[59] and timescale estimates from spiking activity[60]. Our results suggest that sleep-related changes in aperiodic activity involve both E-I balance and neural timescales. While E-I dynamics inferred from the exponent reflect global changes between wake and sleep, knee frequency more effectively differentiates sleep stages and explains exponent changes when using a single-exponent model. This pattern is particularly clear in the intracranial data, in which measurements of neural timescale from local field potential (LFP) data have been established[30], though similar patterns are observed in EEG data, indicating the potential for extracranial recordings to measure neural timescales as well.

By analyzing sleep data from intracranial (iEEG) and extracranial (EEG) recordings, we identified both similarities and modality-specific

differences that warrant further investigation. The knee parameter was more pronounced in iEEG but less consistent in EEG with model selection comparisons suggesting that the knee model fits iEEG data better overall, whereas in EEG, no single model emerges as universally superior. However, EEG showed a clear knee during certain sleep stages, with a better knee model fit in Wakefulness, REM, and N1, indicating that the prominence of the knee parameter may be sleep-stage-dependent. These findings highlight the need for further research into stage-specific knee dynamics in EEG. While stage-dependent knee frequency has been observed in other contexts[31,61,62], its role in sleep spectral dynamics remains underexplored. This underscores the importance of data-driven model selection, prioritizing broadband frequency ranges, adequate time windows, and robust goodness-of-fit measures to improve reproducibility and model selection across neural states. Another potential contributor to the differences between iEEG and EEG recordings may be their respective referencing montages. While we re-referenced EEG data to an average reference, iEEG recordings used a bipolar montage, which may impact power spectral characteristics and, as a consequence, model fitting outcomes.

Interestingly, applying the knee model revealed differences between modalities that were not evident when using the fixed model. Specifically, while the pattern of exponent values across sleep stages was consistent across modalities in the fixed model—and remained similar in the EEG data when using the knee model—the iEEG data showed a distinct pattern of exponent changes across stages under the knee model. This suggests potential modality-specific effects, likely related to the influence of knee prominence on exponent estimates. These differences may also reflect variation in underlying signal sources between EEG and iEEG. Overall, this result highlights the importance of selecting model forms that align with the characteristics of the data and the goals of the analysis, and contributes to the broader research effort aimed at understanding how different parameterizations capture neural dynamics across modalities. Future studies should further examine knee occurrence in sleep EEG and explore how exponent changes relate to knee shifts across sleep stages, as observed in iEEG, and seek to further understand the differences between modalities.

Another key aspect of this investigation is the use of time-resolved measures of aperiodic activity, extending beyond and complementing more common analyses of temporal patterns of oscillatory components[63]. In this study, we showed that by tracking the temporal fluctuations of the spectral exponent, we can map the dynamics of transitions between distinct sleep stages, as well as event-related, transient, stimulus-specific responses of the sleeping brain to auditory perturbations. This is consistent with contemporary work emphasizing the dynamic nature of aperiodic activity, with rapid changes across brain states and in response to external stimuli[33,34]. Previous research has shown that the spectral exponent of NREM sleep flattens across successive sleep cycles[11], and that time-resolved aperiodic activity can distinguish between healthy and disordered sleep[10]. Building on these findings, our work refines measurement methods for aperiodic activity, capturing fluctuations at macro and micro levels of sleep structure to enhance our understanding of sleep dynamics. Future work can further probe the temporal dynamics of aperiodic activity during sleep, offering insights into its role in predicting spontaneous sleep-specific events (KCs, sleep spindles, slow waves, etc), as well as its broader functional significance during sleep[64].

These advancements may deepen our understanding of the neural processes underlying sleep and their relation to brain function. On a similar note, our findings also have implications for sleep scoring – including suggesting features that may assist in the detection of different sleep stages, and information about the transitions between sleep stages. While sleep scoring is defined primarily in terms of oscillatory features, this work supports previous research suggesting the aperiodic exponent may be a useful parameter for adjudicating between different sleep stages[13,14,39]. In addition, our findings extend previous results by showing that the aperiodic knee may also be especially useful in distinguishing between sleep stages, potentially contributing to improving the accuracy and reliability of the staging process. Of particular interest is the finding that time-resolved aperiodic parameters

show distinct deviations over time during sleep stage transitions, suggesting that these measures may be useful for empirically investigating the progression between sleep stages, and potentially offer increased time-resolution for detecting changes within and between sleep stages. The generalizability and reliability of these findings should be assessed through further research and validation across different sleep datasets.

Our analysis of evoked responses to external stimuli revealed transient, event-related steepening of the aperiodic exponent during NREM sleep, suggesting an increase in neural inhibition that may support sleep maintenance[36,48]. Notably, the responses were stimulus-specific, with greater steepening following unfamiliar voices compared to familiar ones, consistent with prior findings of similar trends during NREM sleep[48]. Crucially, we observed similarities between exponent responses, ERPs, and elicited KCs, which raises questions about their independence and overlap. While our analyses suggest an aperiodic component contributes to evoked KCs, exponent changes were also observed in trials without KCs and appear partially distinct from ERP responses. Additionally, our KC detection method, based on a strict amplitude criterion, may have missed weaker or atypical KCs, potentially influencing the observed spectral changes. Previous work has shown that event-related exponent changes differ from ERPs[65,66], suggesting aperiodic responses as a distinct measure of neural activity. These findings establish event-related aperiodic responses during sleep and highlight the need for further research to explore their role in stimulus-specific processing and their relationship to other evoked responses.

## Limitations

Limitations of this study include the relatively small number of subjects in the EEG dataset, as well as the relatively small amount of data per subject in the iEEG dataset, which precluded evaluating the analyses across larger datasets. Potential confounds, such as differences in referencing procedures between EEG and iEEG, may affect the comparisons between the different recording modalities. The complexity of model selection—particularly regarding the presence or absence of the knee parameter across sleep stages and modalities—also poses challenges for standardizing analyses across applications so as to be directly comparable. Relatedly, although the spectral parameterization approach using the Lorentzian function allows for examining the knee points in the spectrum, it does not always fully capture low-frequency activity. Future studies with larger, more diverse cohorts and further work advancing the modeling techniques are needed to validate and extend these findings in order to further clarify the physiological basis of aperiodic features and improve their applicability in both clinical and research contexts.

## Conclusion

In this investigation, we sought to explore aperiodic activity during sleep in both intra- and extracranial data. We have shown that by using a broader frequency range and fitting a model with a knee, we can characterize more of the data and obtain estimates of aperiodic activity that map onto sleep stages, sleep stage transitions, and transient responses to external events. A key aspect of this work is the analysis of the knee frequency, which effectively discriminates between sleep stages and provides potentially new insights into the timescale of neuronal processing during sleep. Overall, these findings highlight the importance of studying aperiodic neural activity during sleep. By adopting these expanded parameters, we can gain new insights and perspectives on sleep processes, improving our understanding and interpretation of both the temporal and spectral dynamics of neural activity during sleep.

## Data availability

iEEG open-access data are available on the Montreal Neurological Institute (MNI; https://mni-open-ieegatlas.research.mcgill.ca/). The EEG sleep dataset analyzed in the current study is available from the corresponding author upon request.

## Code availability

Code for this project was primarily written in the Python programming language (v3.10.6), except for the pre-processing and segmentation of EEG data which were primarily done using EEGLab[67] (v14.1.1b) functions in Matlab v.2019a. We analyzed iEEG and EEG data using MNE-Python[68], and performed spectral parameterization analyses using the 'specparam' toolbox (https://github.com/fooof-tools/fooof). We deposited the code of this project in the project repository and made it openly available and licensed for reuse at https://github.com/mohamedsameen/Aperiodic_sleep.

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

## Acknowledgements

We would like to thank Christine Blume and Renate del Giudice for data collection. MSA is funded by the Austrian Academy of Sciences (ÖAW) and the Austrian Science Funds (FWF; Doctoral College "Imaging the mind"). The funders had no role in study design, data collection and analysis, decision to publish or preparation of the manuscript. This research was funded in whole or in part by the Austrian Science Fund (FWF) [10.55776/W1233]. For open access purposes, the author has applied a CC BY public copyright license to any author accepted manuscript version arising from this submission.

## Author contributions

M.S.A. contributed to the analysis plan, performed data analysis, and wrote the manuscript. JJ contributed to the analysis plan. M.S. contributed to the experimental design. K.H. contributed to the analysis plan and manuscript drafting. T.D. contributed to the analysis plan, data analysis, and manuscript drafting.

## Competing interests

The authors declare no competing interests.
