## [Transparent Peer Review file · Communications Psychology]

Temporally Resolved Analyses of Aperiodic Features Track Neural Dynamics during Sleep

Corresponding Author: Dr Mohamed Ameen

Version 0:

Decision Letter:

Dear Dr Ameen,

Thank you for your patience during the peer-review process. Your manuscript titled "Temporally Resolved Analyses of Aperiodic Features Track Neural Dynamics during Sleep" has now been seen by 3 reviewers, whose comments are appended below. You will see that they find your work of some potential interest. However, they have raised quite substantial concerns that must be addressed. In light of these comments, we cannot accept the manuscript for publication, but would be interested in considering a revised version that fully addresses these serious concerns.

We hope you will find the Reviewers' comments useful as you decide how to proceed. Should additional work allow you to address these criticisms, we would be happy to look at a substantially revised manuscript. If you choose to take up this option, please highlight all changes in the manuscript text file, and provide a detailed point-by-point reply to the reviewers.

Editorially, there are three domains in which your manuscript requires substantive revisions.

First, please address all methodological concerns listed by the reviewers through additional data analysis. Second, Reviewer #1 notes disagreement with the literature that is included in the structured literature review (Figure 1); they also request citation/inclusion of specific articles. Editorially, we require systematic literature reviews to follow PRISMA guidelines. We therefore ask you to include a systematic PRISMA-guided approach (this will also require you to include a PRISMA flow-chart which may be in the SI). The range of search terms and publication years needs to be justified. Third, there are significant presentational issues that need to be addressed; these include a clear statement that your work is exploratory and not confirmatory (i.e. neither hypothesis-driven nor a replication); a balanced treatment of the literature that explains how your work confirms, extends, or contradicts previous publications; and the removal of all novelty claims.

I am attaching a checklist that details critical reporting requirements for the revised manuscript. Please attend to each item and ensure your manuscript is fully compliant. We are requesting that your manuscript aligns with these requirements as this facilitates the evaluation of your manuscript, reducing delays in re-review and potential future acceptance. If your revised manuscript is not aligned with these requests on major issues, such as those concerning statistics, it may be returned to you for further revisions without re-review. Additional information can be found in our style and formatting guide Communications Psychology formatting guide.

If the revision process takes significantly longer than five months, we will be happy to reconsider your paper at a later date, provided it still presents a significant contribution to the literature at that stage.

Please use the following link to submit your
- revised manuscript,
- point-by-point response to the referees' comments,
- cover letter (as a separate document),
- the Editorial Policy Checklist (see below),
- the Reporting Summary (see below), and
- the completed Editorial Request Table (attached):

Link Redacted

Thank you for the opportunity to review your work.

Best regards,

Marika Schiffer

Marika Schiffer, PhD
Chief Editor
Communications Psychology

REVIEWER EXPERTISE:

Reviewer #1 sleep, electrophysiology, time-frequency analysis

Reviewer #2 + #3 (these reviewers worked together on a single report) sleep, electrophysiology, time-frequency analysis

REVIEWER REPORTS:

Reviewer #1 (Remarks to the Author):

This paper focuses on an interesting issue, namely the sleep stage, time-related and stimulus evoked patterns in fractal spectra (aperiodic activity or spectral slope/exponent) of the wake-sleep (i)EEG of human subjects. Although interesting and promising, the manuscript suffers from major issues which imply major revision according to the points raised below.

I. Introduction

1. I understand that the focus of the authors was on the scientific literature published between 2009 and 2022, but still a biased overall picture of the field is not allowed. Authors overlook the literature

- on the very first case of fitting a 1/f-type function to human EEG (Pritchard, 1992)
- which relates the spectral exponent to the bistability of the local neural field potentials (Milstein et al., 2009), as well as to the steepness of the transitions between up and down states (Baranauskas et al., 2012) instead of the exclusive involvement of the frequently cited E:I ratio
- indicating the appropriateness of the term scale-free activity within the context of sleep-wake behaviour of neural field dynamics (Why the aperiodic activity is scale-free at the same time? This is an important theoretical construct for psychology emerging in several fields like the temporal integration of the self [Kolvoort et al., 2020] or the metastable brain-state switching driven by human cognition [Mora-Sánchez et al., 2019])
- proposing a model for the pivotal role of the spectral exponent in regulating wake-sleep dynamics, as well as the first descriptions of the time-resolved analysis of aperiodic activity during sleep in different age groups, as well as in healthy and insomnia subjects (Bódizs et al., 2024).

Besides being relevant in improving the Introduction of the manuscript, the above detailed shortcomings are relevant for the section Discussion as well.

<https://doi.org/10.3109/00207459208999796>

<https://doi.org/10.1002/hbm.25129>

<https://doi.org/10.1007/s11571-019-09533-0>

<https://doi.org/10.1016/j.pneurobio.2024.102589>

2. One additional reason for changing the focus of the current literature mostly involved in the practice of fitting the log-log spectra with a linear function in a narrow frequency range is evident from Figure 2C of the manuscript, but it is not explicitly claimed in the Introduction part of the paper. This panel of Figure 2 clearly indicates that narrow frequency ranges are more susceptible to the biases induced by fitting on incomplete spectral peaks (these biases are practically uncontrolled and non-visible when we consider time-resolved analyses). The latter issue was presented by Gerster et al (2022) under the subheading "Avoiding Oscillations Crossing Fitting Range Borders". Although this paper introduced the issue in the field, it provided a somewhat biased presentation of it, by emphasizing mainly the low frequency peaks instead of the very common high frequency ones, which could affect the narrow fits.

II. Methods

1. Please specify the three additional articles added to the list of automatically identified and manually verified ones, as well as the way you selected them. Moreover, authors should indicate if these articles would indeed imply the addition of new search terms by using the OR operators (i.e. by widening the literature search).

2. Authors mention that the non-cortical regions were excluded from further analyses of iEEG data and exemplify some areas which are indeed considered as cortical ones in the neuroanatomical literature (authors are asked to take a look at some brain atlases or review articles and search for the parahippocampal gyrus or the fusiform gyrus: both are clearly identified as cortical regions!). Please clarify the reason of excluding these regions which are of utmost interest in the neuropsychology of higher order cognitive functions!

3. iEEG data: concatenated, non-continuous records (even if filled with artificially inserted zeros), are clearly not suitable for time-resolved analyses of ongoing brain activity. As a consequence, authors should not indeed use that iEEG data for this type of analysis, which cancels a significant ambition of their work. Alternatively, authors should clearly state if the concatenated records were used and how they were used during their work.

4. EEG data

"Finally, we performed independent component analysis (ICA) in EEGLab and visually labeled and removed noise-, heart-, and eye-related components."

Did authors record the ECG activity by using a specifically dedicated (ECG) derivation? This would ensure that the ECG artefacts are appropriately identified. Also, the records of eye movements and muscle tonus are parts of the minimal criteria of staging sleep. Were these measures (EOG, EMG) available? If not, how were the sleep records staged?

5. PSD estimation is at the heart of the paper, but its description is unfortunately very vague. Authors claim the use of the Welch method, but miss the specification of some significant details (e.g. window type). Given this drawback, the reproducibility of these results is low.

a. The use of 15 s epochs (which are considerable longer than the ones frequently used in the sleep EEG field) do not always fit with the de-artefacted data, especially in the case of iEEG records, which were said to be concatenated. How were these concatenated and zero-filled segments PSD analysed?

b. PSD calculations of segments which are not conforming the length of the canonical power of 2 numbers (1024, 2048, etc): what kind of a procedure was used for handling this issue? Did authors always use Discrete Fourier Transformation or rather mixed-radix Fast Fourier Transformation? Another possibility is the zero padding procedure and the use of the common Fast Fourier Transformation routine. If this latter approach was used, how were the zeros added to the segments: before or after windowing (tapering)? What was the type of window used within the context of Welch method of PSD analysis? (See the same question above!)

c. The multitaper spectral analysis implemented for time-resolved parametrization of EEG activity is welcome, but should be specified during the description. The name of a script is far from an appropriate and accurate description. Scripts are changing, as do their baseline settings. As a consequence the technical term `mne.time_frequency.psd_multitaper` is only acceptable with the parallel use of an accurate scientific citation and specification of the settings. This would provide some control for the researcher degree of freedom in the field.

6. Spectral parametrization

"We calculated the knee frequency from the knee parameter, removing any values that were higher or lower than the mean \pm 2 standard deviations."

What kind of values were indeed removed? The term "any value" refers to what? Knees, spectral bins, etc? Is the mean the sample mean? Please clarify!

III. Results

1. My first and most important critique refers to the interdependence of the model parameters (the knee frequency and the spectral exponent) and its interpretation. A strong correlation between model parameters is usually considered as an ill-conditioned statistical scenario, characterized by a contorted error surface where the minimum is not easily identifiable. As a consequence, the best fitting parameter values cannot be easily found (see Li et al., 1996 for a more general discussion of the issue). In general such a strong interdependence between the model parameters is non-coherent with the overall aims of modeling EEG/LFP spectra, which was motivated by reducing the redundancy of bin-power values (Bódizs et al., 2021). Instead of acknowledging the strong redundancy of the model authors are arguing for the strength and relevance of this finding, stating that former, fixed (no-knee) models were indeed incorporating the differences in knee frequencies. However, the strong correlation between model parameters largely destabilize the basement of this argument, because we could derive a different (inverted) reasoning from these findings as well. Namely, knees could indeed reflect exponent values and not vice versa (correlation is non-directional). In this case, knees are reflecting the pressure to obtain a better fit, but they are indeed derived from the same underlying process which determines the spectral exponent (overfitting). Most probably the knee is best positioned to capture a white noise type of power distribution in the vicinity of zero frequency, but this distribution is not always of the white noise type (see also the fits provided by Colombo et al., 2019). My objection is also supported by the radical change in sleep stage-specific exponent values after introducing the knee parameter, resulting in a strongly atypical pattern of findings. Although these might be a real pattern, but could also emerge as a reflection of the ill-conditioned statistical setting characterized by the strong interdependence of the model parameters. Authors are requested to reflect on this issue and change their text from the current overenthusiasm type of reading to a more balanced and logical doubt in this respect.

2. Authors fail to provide parallel empirical evidence of sleep stage and time-related effects in gold standard, band-limited spectral indices, most notably Slow Wave Activity (SWA: 0.75-4.5 Hz). This would also help to clarify somehow the ambiguity concerning the knee vs exponent issue in many respects. Also SWA-differences might provide the readers with a gold standard type of knowledge in cases where effects like decreasing N3 sleep EEG exponents along the night imply a need for additional insight (e.g.: Is this effect paralleled by a decreasing sleep intensity?).

3. Sleep stage transitions in terms of fixed vs knee-model

"The results are broadly consistent, although less reliable and temporally precise as the exponent of the knee model, suggesting that the knee model may be well posed for detecting variations during the transitions between sleep stages." Such statements should be based on empirical and statistical evidence. How were the difference between reliability and temporal precision tested? Are the differences in d-values strong enough to support this claim? Please indicate!

IV. Discussion/Conclusion

1. Please consider the fact that sleep stage-related differences in aperiodic activity are not the only category of empirical findings which might be relevant for the interpretation of these findings. The formerly reported overnight changes in the spectral exponent during successive sleep cycles (Horváth et al., 2022) indicate that the time-related types of finding reported by the authors in the current paper are not without antecedents.

2. Equality/non-equality of the changes in stimulus-evoked aperiodic activity and K-complexes

The reasoning of the authors is largely considering their K-complex detecting procedure as a gold standard or even some ground truth. Evoked K-complexes could rather be considered as statistical phenomena, but not some all-or-nothing event. Authors might consider the fact that the EEG results from the admixture of multiple neural sources, thus the significant stimulus-induced changes in aperiodic activity in the absence of a detected K-complex should be considered within this context (as well).

3. The significant positive deflection following UFVs

Is this finding coherent with the literature? (E.g.: Perrin et al., 2000)

<https://doi.org/10.1038/s41598-022-23033-y>

https://journals.lww.com/neuroreport/abstract/2000/06050/functional_dissociation_of_the_early_and_late.8.aspx

Reviewer #2 (Remarks to the Author):

The work by Ameen and colleagues studies how the spectral exponent and knee of power spectra change during sleep. They present different models and frequency ranges to fit the exponent and time-resolved analyses using a variety of datasets. The manuscript deals with an important investigation of exploring how the exponent changes during sleep, with which model or frequency ranges to fit. All of these are valuable as the literature on the exponent has rapidly expanded. There are a few methodological and theoretical questions to address.

Major points

1. The introduction presents a broad overview of the literature which eventually lays out some objectives (expand previous research & broaden the investigation of aperiodic activity during sleep; evaluating frequency ranges and model forms, etc). However, there are no hypotheses on what is expected out of the presented analyses, which then makes the interpretation of findings challenging, as it is hard to evaluate what is surprising. This may be a methodological work, but still authors likely had some expectations and hypotheses when starting it out.

2. When analyzing the exponent with the knee model in EEG vs. iEEG data there seems to be a key difference: the iEEG exponent decreases from wake to N2/3 and REM (Fig 4D), while the EEG exponent increases (Fig 5C). This is currently not described in the results as such, as the results describing these figures mention that the exponent differs between stages, without highlighting the direction of the difference. Could authors describe this finding more explicitly in the results? Authors should also discuss why there is this difference in the change of the exponent depending on whether one is analyzing intracranial or scalp EEG. Moreover, this discrepancy is not found with the fixed model. Is this because the model is more reliable, or because it is less sensitive and thus does not uncover a key feature in the data?

3. For iEEG data, the direction of change in the exponent from wake to sleep differs depending on whether the knee or fixed model are used. With the knee there is a decrease from wake to sleep, while with the fixed model an increase (Fig 4D/E). This is also not really discussed. How can this be explained? Can authors add a more detailed description of this finding?

4. The authors report a strong positive correlation between the variance in R2 and the variance in the exponent (Fig 3E) and offer important insight into the possibility that fitting spectral models to broad frequency ranges offers increased reliability. For the fixed model in particular, but also for the knee model, the relative R2 and exponent between brain states appear similar (Supp Fig 4-2 and Fig 4 D-E). For the fixed model, when R2 increases, so does the exponent. Further, the confusion matrices in Fig 3B and Supp Fig 3-1B both appear to trend towards a positive correlation between the exponent and R2: higher R2 and higher exponent value with a shorter time window. Can the authors report the correlation between R2 and the exponent? If there is dependence between the two variables, the authors can consider including a point in the discussion on whether this dependence is an epiphenomenon of model fits being "better" for more rapidly decaying power spectra or if the possible dependence reflects "worse" fits being biased towards higher exponents.

5. In discussion, the theoretical implications of changes in the exponent are laid out. However these can be better linked to the present findings. It is mentioned for instance that "Our findings here are at least partially consistent with these findings, whereby across all the different model settings this general pattern of exponent differences between Wake, NREM, and REM stages is observed." What does partially consistent mean? Can authors elaborate which part of the results follows

previous work and which not, and possible interpretations for differences?

One critical parameter in the present work is the fitting range which differs compared to that of previous studies, which have established links between excitation-inhibition and exponent based among other things on simulated data. Given that the direction of change in the exponent is quite different depending on modality & model as well as the fitting range compared to previous studies, direct links between excitation-inhibition balance, exponent, and sleep dynamics seem more difficult to establish. Could authors (a) discuss more clearly which of their findings are partially consistent with previous work; (b) implications of such discrepancies on the interpretability and theoretical framing of the exponent for future studies?

6. When computing the knee model, authors discuss extensively how a knee may hard to identify on scalp EEG data but more pronounced in intracranial data. However, a knee may also be absent from some intracranial channels as well, or less reliable. How do authors recommend coping with that?

Minor points

1. It is unclear which iEEG data are used in section "3.1. Model selection for estimating aperiodic activity." Is this across all data segments or within a specific brain state?
2. In section 3.3. BIC differences between knee and fixed models are reported in parentheses (paragraph 4). It should be made more explicit that the values in parentheses are differences between models.
3. In section 3.4 it only becomes clear which of the two exponent models is used at the end of the section when authors refer to the supplemental analyses. Can you clarify earlier in the section?
4. Figure 6A: Can you reduce the line width of the exponent trace as it is currently not possible to clearly see the state-dependent differences in the exponent?
5. Figure 4C: I suppose that the individual colors in the line plots correspond to brain regions. As adding a legend seems difficult given the number of lines you could consider recoloring them to the same color to not overlap with sleep stages and create confusion.

EDITORIAL POLICIES

We ask that you ensure your manuscript complies with our editorial policies and reporting requirements.

To that end, we require revised manuscripts to be accompanied by two completed items: a reporting summary that collects information on study design and procedure, and an editorial policy checklist that verifies compliance with all required editorial policies

- <https://www.nature.com/documents/nr-reporting-summary.zip>>Nature Research Reporting Summary
- <https://www.nature.com/documents/nr-editorial-policy-checklist.pdf>>Editorial Policy Checklist

All points on the policy checklist must be addressed. Your revised manuscript can only be sent back to the referees if these checklists are completed and uploaded with the revision.

Notes: If you have submitted a Stage 1 Registered Report, Review, Primer, Comment, or Perspective you do not need to submit these forms. If you have already submitted these forms, you may disregard this request.

** Visit Nature Research's author and referees' website at <http://www.nature.com/authors>>www.nature.com/authors for information about policies, services and author benefits**

Version 1:

Decision Letter:

Dear Mohamed,

Thank you for your patience during the peer-review process. I am sincerely sorry for the delay in returning to you with a decision. As mentioned in my previous correspondence, two referees could not submit their reports. We therefore recruited another expert (Reviewer #4) to provide us with feedback on your revisions.

Your manuscript titled "Temporally Resolved Analyses of Aperiodic Features Track Neural Dynamics during Sleep" has now been seen by (old) Reviewer #1 and (new) Reviewer #4, and I include their comments at the end of this message.

Reviewer #1 finds your work much improved but has some remaining concerns. Reviewer #4 raises a number of methodological issues (some echoed in Reviewer #1's report).

We remain interested in the possibility of publishing your study in Communications Psychology, but would like to consider your responses to these concerns and assess a revised manuscript before we make a final decision on publication.

We therefore invite you to revise and resubmit your manuscript, along with a point-by-point response to the reviewers. Please highlight all changes in the manuscript text file.

I am attaching an Editorial Requests Table that details critical reporting requirements for the revised manuscript. Please attend to each item and ensure your manuscript is fully compliant. If your revised manuscript is not aligned with these requests on major issues, such as those concerning statistics, it may be returned to you for further revisions without re-review.

Please submit the following items:

- Revised manuscript
- Point-by-point response to the referees' comments
- Cover letter (as a separate document)
- <https://www.nature.com/documents/nr-reporting-summary.zip>>Nature Research Reporting Summary
- <https://www.nature.com/documents/nr-editorial-policy-checklist.pdf>>Editorial Policy Checklist
- Completed Editorial Request Table (attached).

via this link: Link Redacted .

Additional guidance is available in our style and formatting guide <https://www.nature.com/documents/commspsychol-style-formatting-guide-accept.pdf>>Communications Psychology formatting guide.

We hope to receive your revised paper within 8 weeks; please let us know if you aren't able to submit it within this time so that we can discuss how best to proceed. If we don't hear from you, and the revision process takes significantly longer, we may close your file. In this event, we will still be happy to reconsider your paper at a later date, provided it still presents a

significant contribution to the literature at that stage.

Best regards,

Marike

Marike Schiffer, PhD
Chief Editor
Communications Psychology

REVIEWER REPORTS:

Reviewer #1 (Remarks to the Author):

The clarity and scientific value of the paper increased considerably. Indeed, there are still some points which I think were not appropriately addressed during the process of the manuscript revision.

1. This point is about the Lorentzian function-based model of the (i)EEG power spectrum. In their rebuttal authors claim that timescales and auto-correlations are reflected by the knee-parameter derived from their model. We agree. But we do not agree in our views on the overall shape of power spectra below the knee frequency and the corresponding time series features. Authors implicitly and perhaps involuntarily opt for a flat spectrum-type model in the vicinity of zero frequency (Fig. 4B), and for the correlated model variables (knee frequency correlated with the spectral slope steepness in some sleep stages). Although, authors included a particularly large number of additional sentences and analyses in their revised manuscript, their claims are basically unchanged. Let me exemplify these points by the following remarks:

a. If the reader focuses on the reported data and considers panel A of Figure 4., which depicts the "Average PSDs of the different sleep stages of the iEEG data", (s)he can see slopes of varying steepness below the state-dependent means of knee frequencies (Fig. 4., panel D). But no flat lines (steepness equalling zero) are seen in these averages depicted in panel A. This implies that the expected shape of the spectra below the knee frequency is negatively sloped but heavily sleep state-dependent. Consequently, lower (i)EEG frequencies could indeed be characterized by their own, non-zero spectral exponents. In contrast to this empirical finding, the Lorentzian model is strongly determined to fit a straight line in the vicinity of zero frequency (see panel B of Figure 4 and the diverging model and data in the vicinity of the vertical axis!). In other words, frequency-independency is implicitly assumed below the knee, right in the range where large and spectacular differences in slopes are evident by visual inspection. The paper published by Colombo and his colleagues in 2019 is evidently not an implicit support of the usefulness of the knee-model (fitted Lorentzian). In this latter paper non-zero slopes are reported in both the low and high frequency ranges of the EEG (albeit frequency ranges are defined on an ad-hoc basis). I understand that the neural model of Gao suggests a primordial relevance of higher frequency slopes as an indicator of the assumed E-I ratio, as well as the fact that the knee-model is coherent with these theoretical claims. Still, broken power laws, which might be operative in the realm of the (i)EEG are not necessarily well-captured by the knee model (the latter assuming a non-power law shape of the spectra in the lowest frequency range). Consequently, authors are requested to reflect shortly on the potential limitations caused by the knee-model (Lorentzian-shaped spectra), with a special reference to the negatively sloped, potentially (seemingly) sleep stage-specific and non-zero low frequency power-law scaling in the iEEG data. The spectral parametrizations of the lowest frequency range (1-8 Hz) could be informative in this regard. Are these low frequency spectral slopes sleep stage-specific? If yes, are the results congruent with the outcomes of the knee-model-based parametrization (which prefer to emphasize the higher frequency slopes)? I was not able to find these types of data in the manuscript and its supplementary figures or tables.

b. The spectral knee frequency and the spectral slope are strongly correlated. This point is not acknowledged appropriately, and the caption of Figure 5-1 is even contradicting the information presented in the paper. "D) No correlation between the change of the knee frequency and that of the exponent of the knee model (Spearman correlation: $\rho = -0.006$, $p = 0.96$)."

Indeed, there is a particularly high sleep stage-specific correlation between these measures, depicted right in this panel! I do not think that this strong relationship can be considered as non-existing or non-significant (the W stage distorts the relationship but otherwise one can see a positive correlation, which is especially true in the case of REM sleep).

2. Authors implement a series of spectral parametrizations by relying on different frequency ranges derived from the literature. Indeed, the results of most of these parametrizations are only used to highlight the superiority of the broadband knee-model. Findings on sleep stage-specificity of these alternative spectral slopes are not included in the manuscript or the supplementary materials. Reasons for this lack of presentation seems to find its roots in the fact that the exponents are strongly dependent on the frequency range itself. Indeed, spectral slopes are not necessarily frequency-independent (see for example the broken power law models which are ubiquitous in nature, as well as the paper of Colombo and colleagues revealing pharmacological specificity of altering lower vs. higher frequency slopes). Authors should at least shortly include the sleep stage-specificity of these exponents and not just consider them as inferior to the highly preferred knee-model-derived ones. Based on the visual inspection of the iEEG spectra particularly large effects are expected in these analyses.

3. An additional minor, but significant point is the further potential source of the divergence between iEEG and EEG data. Besides other differences, scalp EEG is analyzed in average reference montage, whereas no indication of reference electrode (montage) is given in the case of iEEG records. The latter are presumably bipolar (the authors are invited to

consider this point and mention it in the section Methods [iEEG montage] and Discussion [divergent findings for divergent montages]).

Reviewer #4 (Remarks to the Author):

The article "Temporally Resolved Analysis of Aperiodic Features Track Neural Dynamics During Sleep" explores how different parameters in the analysis of aperiodic EEG and iEEG signals during sleep can affect model fit and overall results. The paper initially focuses frequency ranges and then on the less-used "knee" parameter, and then applies these models to time-resolved sleep data. The results indicate that broader frequency ranges and the knee parameter can improve model fit and can reliably discriminate between sleep stages and sleep microarchitecture events. They further demonstrate that aperiodic measures provide valuable insights into the temporal dynamics of sleep.

I found the work to be generally well done, the text very clear and understandable, and overall a valuable contribution to the literature. My main concern with the paper is that it does not always accurately describe its own results, and there are some pitfalls in the analysis choices that limit interpretability, therefore some statements throughout the text are not properly supported by their data. Below, I identify some critical changes to the paper that need to be addressed. I then suggest some minor points that should be addressed, and finally, a few larger but optional changes that could improve the manuscript.

Major

1) In results section 3.1, the clustering of frequency ranges into "broad" and "narrow" inappropriately penalizes the "narrow" category by combining non-overlapping frequency ranges. — The first analysis of this paper focuses on the effect of the frequency range on aperiodic model fit. The authors' choice to use ranges found in the literature grounds the paper to the parameters researchers are using in practice, but the downside is that they are now varying two factors at once: the width of the frequency range, and the range itself. Given that the authors wish to emphasize the importance of broad spectral ranges, they should only directly compare overlapping ranges of increasing width (1-8, 1-20, 1-30, 1-45, etc.) and not pool non-overlapping frequency ranges (1-8, 1-20 / 20-45, 30-45). In fact, the R^2 values within the category of "narrow range" appear inconsistent with the overall conclusion, since 1-8 Hz outperforms 30-45 Hz, and 1-20 Hz outperforms 20-45 Hz at both 15 and 20 s PSD windows. The authors could strengthen their analysis by either systematically varying range width in one figure (1-8, 1-20, 1-30, 1-45, etc) and frequency band locations in another (1-20, 10-30, 20-40, etc). At a minimum, the comparisons between 'broad' and 'narrow' frequency ranges (Figure 3C-E) should include only fully nested ranges for a more controlled comparison.

Related: The correlation analyses presented in Figure 3E and Supplementary Figure 3-3 appear inappropriate given the clustered distribution of the data points. In Figure 3E, we can see that for a nearly constant exponent variance value of approximately 0.55, R^2 values vary across the entire observed range, suggesting no correlation within this cluster. The reported significant correlation between exponent and R^2 variance appears to be driven by the relationship between clusters rather than a true continuous relationship within the data; a classic example of Simpson's paradox. The conclusion that "model fits influence the variability in exponent values" (page 15, line 435) is questionable, also since this relationship is not observed for any of the other models examined in Supplementary Figure 3-3, where correlations are non-significant. I recommend either removing this conclusion or conducting a more appropriate analysis that accounts for the clustered nature of the data.

2) Differences between the supplementary data and the main figures are often understated, sometimes in ways that are quite critical to the narrative of the paper. — The authors need to better emphasize the differences between the results with different models and different datasets.

a. On page 15 line 436, it says that the EEG data yielded similar results as iEEG data (Figure 3 vs. Suppl. Figure 3-2). This is inconsistent with the data presented in the Figure 3-2: when applied to EEG data, broader frequency ranges yielded worse models than narrower ranges. Models 1-20 and 1-45 Hz had identical R^2 values, and even 1-8 Hz outperformed 1-75 and 1-100 Hz. These results for EEG data are not surprising (high frequencies are known to be unreliable in the surface EEG), but it is important that the authors acknowledge this difference in the main text, especially since more readers will have access to EEG data rather than iEEG data.

b. On page 17 line 477, the text says that the exponent of the fixed model did not significantly differ between sleep stages, but the values in table 4 indicate that they did. This inconsistency needs to be resolved, as it appears to downplay the performance of the fixed model when the data suggests it may perform better than reported.

c. On page 24, line 661, the claim that fixed model exponents 'were less temporally related to the transitions between stages' is not supported by Supplementary Figure 6-2, where the dynamics appear equally or more pronounced than with the knee model. The main difference is that the fixed model shows larger baseline differences between stable and transition epochs, which may actually indicate greater sensitivity to pre-transition changes in brain state.

3) While the paper advocates for including a knee parameter in aperiodic analyses, the authors don't adequately address a concerning inconsistency: when comparing results between recording modalities, the exponents from the fixed model remain largely comparable between iEEG (Figure 4E) and EEG data (Figure 5D), whereas the knee parameters show dramatic differences (Figure 4D vs Figure 5C). Specifically, the exponent direction reverses across sleep depth, and knee frequencies are substantially lower in EEG compared to iEEG data. This inconsistency is problematic, as ideally, EEG analyses should correspond as closely as possible to iEEG measurements. The divergence in knee parameters between recording modalities raises questions about the reliability and interpretability of this parameter. The authors should directly address this limitation and explain why they still recommend including a knee parameter despite these cross-modal inconsistencies. This issue deserves thorough treatment in the discussion section, as it has significant implications for researchers deciding whether to implement knee models in their analyses of sleep EEG data.

Minor

Introduction

- page 3 line 71 "This relationship is consistent with observations that the spectral exponent of the electroencephalography (EEG) signal becomes progressively steeper from wakefulness to non-rapid eye movement (NREM) sleep to rapid eye movement (REM) sleep", whether REM is steeper than NREM is controversial; it depends on the frequency range, with lower-frequency broad ranges regularly finding REM exponents to be less steep than N3 and comparable to N2 (e.g. Schneider et al. 2022, this paper's figure 4E) and narrow band high frequency ranges showing steeper REM exponents compared to N3 (e.g. Lendner et al. 2020). I suggest the authors either directly address this issue, for example from the perspective of knees in the data, or rephrase this sentence to merely saying that EEG becomes steeper with deeper sleep.

- Page 4 line 81-82, it's a bit of a mischaracterization that there is a tendency to examine the 30-45 Hz range, considering the ranges provided in Table 1.

Methods

- Please specify how line noise was removed from the iEEG data (page 8). Was it filtered? How does the specparam algorithm handle the frequencies of line noise when the frequency range used to fit the aperiodic signal overlaps, as in the case of the iEEG data and the results in section 3.1?

- On page 8, the information regarding concatenating artefactual segments is a little hard to follow. Was this process done in the original dataset, including the 0-padding so all segments reached 68 seconds? How does zero-padding 2 s between segments minimize artefacts? Which artefacts? Typically, such an approach would lead to edge artifacts, especially problematic for the PSD, unless the start and end of each EEG segment was tapered to 0. Also, was there a minimum segment duration?

- Could the authors provide the number of trials used in the K-complex analyses?

- Page 11 line 291, I recommend indicating the figure being referred to, and what was averaged: "For the whole night time-frequency plot (Figure 6A) we plotted the channel-averaged value per epoch"

- In section 2.4.2, is there a reason 20 s windows were used for the PSD, but for other analyses it was 15 s? In general, is there a reason for 15 s windows for the PSD, especially considering that the results of figure 3 show better performance for shorter windows?

- In section 2.4.3, the KC data is baseline corrected in the 500 ms prestimulus window; based on the results of Figure 7, this may be somewhat problematic, as there are stimulus evoked effects even before stimulus onset. Ideally the authors should repeat the analysis with an earlier baseline correction. Alternatively, the authors may consider indicating somewhere in the text that future studies should use baselines further removed from stimulus onset, as there seems to be some temporal smearing of the time-resolved aperiodic signal.

- In section 2.6, the authors describe baseline correction which is a bit unconventional for ERPs; usually, like the authors did in section 2.4.3, a pre-stimulus baseline is subtracted from each trial before averaging. Why was a different approach for baseline correction used for these two analyses, considering that they are directly compared between each other?

- Can the authors provide a source for the interpretation of Cohen's interpretation of effect sizes for Kendall's W (page 14 line 378-380)?

Results

- Page 17 line 463; the authors could add Schneider et al. 2022 to the references, especially since those results are more in line with the ones presented in this paper.
- Page 17 line 485 is confusing, “as well as the exponent of the knee model performed at chance ($t(15) = 5.61$, $p_{\text{bonf}} < 0.001$, $d = 1.99$)”; the results show it performs significantly above chance.
- Page 19, line 516 “the knee frequency was a significantly better predictor of stages than the exponent, emphasizing the significance of fitting a knee model to the data”; this assertion could be made if there is a significant improvement between the fixed and knee model, but not between parameters of the same knee model. I would recommend removing it.
- Figure 4, panel G comes before E and F, was this a mistake or deliberate?
- In the caption of Figure 4F, it says “note that both the knee and the exponent of the fixed model had significantly above chance classification accuracies” this is a bit uncalled for, since all three classifiers are significant. I would remove.
- The last line of the caption of Figure 4 and Figure 5 “Each dot represents one brain region...” probably belongs to a different section of the legend, as it does not refer to the dots in Figure 4J or topographies of Figure 5G
- Page 20 line 556, the text says “in contrast to the iEEG data, the knee frequency did not exhibit significant differences between stages” but the p value provided in parentheses is significant ($p < .001$). Please check the statistics.
- In the caption of Figure 5F, BIC differences are provided in parentheses that do not correspond to those in the figure
- In the analyses of Figure 6F (+ suppl) NREM substages were all pooled. Were the control epochs matched by NREM substage to those of the NREM stages at the transitions, such that the same proportion of each substage of NREM was in both the control and transition data? The matrix in Suppl. Figure 6-1, indicates that the majority of transitions to REM came from N2, many came from N1, and almost none from N3. Given this, shouldn't N1-REM transitions also be included?
- In Table 1, Höhn et al. 2024 could be included?
- In the tables, please provide a consistent number of decimal places for the p-values (ideally 3 or 4)

Discussion

- Page 30, lines 793-795 “notably, fitting a single exponent to a narrow band range ignores the potential presence of a knee or bend in the PSD”; this sentence may mischaracterize the literature. The choice of narrower and higher frequency ranges (35-45 Hz) by the cited authors deliberately avoids any of the lower frequency knees described in this paper (generally lower than 10 Hz). They are not ignoring the knee so much as avoiding it entirely.
- The authors have an extremely counterintuitive result: aperiodic exponents with the knee model in the iEEG data are steeper in wake than in sleep (Figure 4D). This goes against the theory that steeper slopes reflect the increased inhibition during sleep, as the authors mention to explain the results in Figure 7 (page 33, line 917-919). How do the authors reconcile this result of Figure 4D with the excitation/inhibition hypothesis? How do they explain that the same knee model yields opposite results across wake and sleep in iEEG vs EEG data? The authors should avoid relying on the excitation/inhibition hypothesis to explain other parts of their data, unless they can explain this discrepancy. It may undermine the interpretability of the exponent in a knee model entirely.

Supplementary material

- On page 49 there's a figure with no caption
- Suppl. Figure 4-2A has “knee model” as a title and “R2-fixed” for a y-axis label. This is switched also for Figure 4-2B

Optional

- Considering the importance of the knee to this paper, I would recommend expanding a little more in detail what “the population timescale” means (page 4, line 101) and how it relates to sleep stages, and why it theoretically reflects different information from the exponent (since these values are then shown in this paper to be highly correlated)
- For how the methods section is currently structured, it was difficult when reading the results to keep track and trace back how data was processed. I would recommend mirroring the methods section to the final results section, such that for each section first describes how the data was processed, how power was calculated, how the aperiodic model was fit, and finally the statistics. With each subsequent section, if the data or method was the same as a previous section, this could simply be

indicated without the need for repetition.

- The first analysis varies both frequency ranges and PSD window sizes; the authors discuss at length the frequency ranges but mostly ignore the effect of PSD window size; considering that they find smaller windows provide greater R^2 values, it would have been appropriate to conduct the later analyses with these 5 s windows. Understandably this would be a major revision over a minor point, but maybe the authors could discuss this aspect of the parameter selection as well, and why one should choose either short or long windows for the PSD.

To conclude, it's a good paper, raising important and often overlooked issues regarding how to conduct analyses on aperiodic signals during sleep. While I am giving many comments, they are mostly related to the text and so should be fairly quick to address.

I wish the authors all the best,
Sophia Snipes

Version 2:

Decision Letter:

Dear Mohamed,

Your manuscript titled "Temporally Resolved Analyses of Aperiodic Features Track Neural Dynamics during Sleep" has now been seen by our reviewers, whose comments appear below. In light of their advice I am delighted to say that we are happy, in principle, to publish a suitably revised version in Communications Psychology.

We therefore invite you to revise your paper one last time to address the remaining concerns of our reviewers and a list of editorial requests. At the same time we ask that you edit your manuscript to comply with our format requirements and to maximise the accessibility and therefore the impact of your work.

EDITORIAL REQUESTS:

SUBMISSION INFORMATION:

OPEN ACCESS:

* DATA AVAILABILITY:

Link Redacted

Best regards,

Marike

Marike Schiffer, PhD
Chief Editor
Communications Psychology

REVIEWERS' COMMENTS:

Reviewer #1 (Remarks to the Author):

Authors of the revised paper claim that the estimated parameters of the aperiodic component of the wake-sleep state specific electrical brain activity largely depend on the frequency range and the inclusion of a knee-parameter inherent to the Lorentzian function of a modified power law model. Broader frequency ranges, inclusion of low frequency activities and the implementation of the knee-parameter-based modelling increased the reliability of the estimation. Moreover, the knee-parameter resulted in a radical change of the wake-sleep state-specific differences in aperiodic spectral slopes. Likewise, the typical value of the spectral knee parameter was revealed to be subject of wake-sleep state-specific changes and depending heavily on recording modalities (intracranial EEG with bipolar reference vs scalp-derived EEG of the average reference-type). Last, but not least the time-resolved analyses of wake-sleep state transitions and stimulus-driven electrophysiological changes revealed a significant methodological divergence of the fixed (no-knee) vs knee-based estimations of the aperiodic activity. The findings are quite divergent, and the methodological pluralism implemented by the authors primarily resulted in different sets of combinations of recording-, setting- and model-specific findings. The latter raise new questions in the field of the spectral parametrization of wake-sleep state-specific neurophysiological signals and could

open new perspectives in interpreting the state-specificity of different time scales and the integration of neural activity over time. I think the number of questions raised by the knee-model are still high, but more appropriately and explicitly considered in the current form of the manuscript. Some of the state specific differences in aperiodic slopes reported before are now translocated to the knee frequencies, at least when considering bipolar iEEG. I think the paper might influence thinking in the field, but will also imply the emergence of some critical thoughts on the appropriateness of the Lorentzian-model in this case (namely spectra of (i)EEG activity).

The methods are sufficiently described and need no further details in order to provide the reader with reproducibility. I consider the paper as being appropriate for publication in its present form.

Reviewer #4 (Remarks to the Author):

The authors have addressed all of my previous concerns.

A few minor points:

- the new text regarding Figure 3E no longer discusses "exponent variance" but rather exponents themselves, despite the figure being about exponent variances. The new conclusion is "better model fits were associated with steeper exponents"; is this only referring to the correlations described in the text? in which case, Figure 3E comparing variances no longer seems relevant.

- the tables continue to not have consistent decimal places for p-values, the new table 5 indicates p-values of 0, and the tables indicate $> .001$ repeatedly, but should likely be $< .001$. But I imagine this is all at the editor's discretion

- the caption of figure 4-3 appears incomplete, with W left empty and "table X".

We would like to begin by expressing our gratitude to the reviewers for their detailed and constructive feedback. We have carefully considered their suggestions and made the corresponding revisions to the manuscript. The point-by-point response is below:

Editorial

1. First, please address all methodological concerns listed by the reviewers through additional data analysis.

All points and comments were addressed and taken into consideration when updating the manuscript.

2. Second, Reviewer #1 notes disagreement with the literature that is included in the structured literature review (Figure 1); they also request citation/inclusion of specific articles. Editorially, we require systematic literature reviews to follow PRISMA guidelines. We therefore ask you to include a systematic PRISMA-guided approach (this will also require you to include a PRISMA flow-chart which may be in the SI). The range of search terms and publication years needs to be justified.

The literature review analysis has been updated to follow PRISMA guidelines and includes additional literature. The points raised by the reviewers were addressed and all the articles included in the analysis were mentioned. A flow diagram was added to the manuscript.

Key Changes:

- Section 2.1 has been extensively revised to conform to PRISMA guidelines, with the addition of two new subsections: one for the literature review and another for the frequency analysis.
- Citation added: Page 7, Line 162 now cites the PRISMA 2020 guidelines (Page et al., 2021).
- Flow charts have been included in the supplementary material (Suppl. Figure 1-1) to illustrate the literature selection process.

3. Third, there are significant presentational issues that need to be addressed; these include a clear statement that your work is exploratory and not confirmatory (i.e. neither hypothesis-driven nor a replication); a balanced treatment of the literature that explains how your work confirms, extends, or contradicts previous publications; and the removal of all novelty claims.

Exploratory Nature: We have explicitly stated that this study is exploratory in nature in both the introduction and discussion sections. This clarification ensures that the work is positioned as hypothesis-generating rather than confirmatory.

[Introduction] Page 6, Line 134: Specifically, in this exploratory, data-driven investigation, we evaluated different frequency ranges and model forms for examining aperiodic activity in sleep recordings.

[Introduction] Page 6, Line 137: In doing so, we sought to explore the temporal dynamics of aperiodic neural activity during sleep, hypothesizing that changes in such activity would track sleep stage transitions.

[Discussion] Page 29, Line 779: In this exploratory work, we sought to extend the investigation of aperiodic activity during sleep, by examining it across modalities (EEG and iEEG), frequency ranges, model forms, and across time.

Balanced Literature Treatment: We have revised the introduction and discussion to provide a balanced perspective, highlighting how our findings confirm, extend, or contradict existing literature. Additional examples of prior studies have been included to contextualize our results appropriately (see updates in the review responses below).

Removal of Novelty Claims: All claims of novelty have been removed, and the language has been adjusted to reflect the contribution of the study in a measured and objective manner.

4. Additional figure updates:

- a) Figure 1B has been updated for a better visualization of the frequency range findings from the literature
- b) Updated Model Comparison Analysis:

Figure 6F: The EEG model comparison (BIC analysis) has been revised to fix an issue with correctly accounting for variation in model parameters between subjects, leading to updates in both the figure panel and the corresponding text.

- c) Improved Classification Analyses:

Figure 4E, Figure 4J, Figure 6F: The LDA classification has been updated to improve the cross-validation procedure, and Random Forest analysis has been moved to the main manuscript (Figure 4J) to further highlight the predictive power of the aperiodic features.

- d) Adjusted Frequency Resolutions:

Figure 4C, Figure 6A-B: The frequency resolutions of the PSDs have been adjusted for improved accuracy and consistency.

Reviewer #1 (Remarks to the Author):

I. Introduction

1. I understand that the focus of the authors was on the scientific literature published between 2009 and 2022, but still a biased overall picture of the field is not allowed. Authors overlook the literature:

a. on the very first case of fitting a $1/f$ -type function to human EEG (Pritchard, 1992).

We appreciate the reviewer's observation regarding historical reports and the significance of reports such as Pritchard (1992). As noted in our introduction, we cited early examples, such as Feinberg (1982) who gave early insights into changes in aperiodic activity during sleep, a topic central to our manuscript. We have also incorporated Pritchard (1992) into the introduction to acknowledge its significance in fitting a $1/f$ -type function to human EEG.

Page 3, Line 55: The aperiodic signal decays with increasing frequency in a $1/f^x$ relationship, quantifiable by the aperiodic exponent (x), which corresponds to the slope of the log-log power spectrum (He et al., 2010; Miller et al., 2009; Pritchard, 1992; Voytek et al., 2015).

b. which relates the spectral exponent to the bistability of the local neural field potentials (Milstein et al., 2009), as well as to the steepness of the transitions between up and down states (Baranauskas et al., 2012) instead of the exclusive involvement of the frequently cited E:I ratio

We have added a sentence to the introduction acknowledging that the spectral exponent can be associated with various phenomena, including to transitions between up & down states, as explored in the noted papers.

Page 3, Line 65: Computational models, supported by empirical data from animal studies, have identified the spectral exponent as a non-invasive indicator of key neural processes including (i) transitions between up and down states (Baranauskas et al., 2012; Milstein et al., 2009), or (ii) the cortical excitation-inhibition (E-I) balance (Chini et al., 2022; Gao et al., 2017; Lendner et al., 2023; Lombardi et al., 2017; Trakoshis et al., 2020), where a steeper exponent suggests stronger inhibitory control, and a flatter exponent indicates higher excitation.

c. indicating the appropriateness of the term scale-free activity within the context of sleep-wake behavior of neural field dynamics (Why the aperiodic activity is scale-free at the same time? This is an important theoretical construct for psychology emerging in several fields like the temporal integration of the self [Kolvoort et al., 2020] or the metastable brain-state switching driven by human cognition [Mora-Sánchez et al., 2019])

We agree that there are multiple conceptual terms that have been applied across the literature, including the term "scale-free." We do not emphasize this term in our work, since the Lorentzian model, which includes the knee parameter, is not strictly-speaking scale-free, as the knee reflects a frequency-dependent change in dynamics (such a system could be considered to have multiple scales or to be 'multi-fractal'). While we do not use this term in our main description, we do cite related works that use this term and conceptualization in relation to LFP measures.

In relation to the points about cognition, while broader investigations into scale-free properties across cognitive science are interesting, we think they fall outside the scope of this analysis as they would require significant discussions of the relationships between different signals and properties (scale-free vs multi-scale and how this maps across different analyses and recording modalities). To acknowledge these points, we have added a statement in the discussion section addressing this remark:

[Discussion] Page 31, Line 859: "...while a true 1/f signal has the same exponent value across all frequency ranges (which can also be called mono-fractal or scale-free), the knee reflects a frequency-specific transition in the power spectrum. In such a case, which can be called multi-fractal or multi-scale, the knee parameter reflects a transition between 1/f regions and the exponent of the knee model reflects the frequency region beyond the knee frequency. "

d. proposing a model for the pivotal role of the spectral exponent in regulating wake-sleep dynamics, as well as the first descriptions of the time-resolved analysis of aperiodic activity during sleep in different age groups, as well as in healthy and insomnia subjects (Bódizs et al., 2024).

We thank the reviewer for referring us to this article. Bódizs et al. (2024) provides critical insights into the time-resolved analysis of aperiodic activity across age groups and clinical populations, which align with and extend our findings. We have integrated these insights into the Introduction and Discussion sections.

[Intro] Page 3, Line 59: recent studies have consistently found that the spectral exponent varies across different sleep stages (Bódizs et al., 2024; Horváth et al., 2022; Höhn et al., 2024; Kozhemiako et al., 2022; Lendner et al., 2020, 2023; Miskovic et al., 2019; Rosenblum et al., 2022; Schneider et al., 2022), contributing to a growing body of literature focusing on aperiodic brain activity during sleep (Figure 1A).

[Discussion] Page 33, Line 909: Previous research has shown that the spectral exponent of NREM sleep flattens across successive sleep cycles (Horváth et al., 2022) and that time-resolved aperiodic activity can distinguish between healthy and disordered sleep (Bódizs et al., 2024). Building on these findings, our work refines measurement methods for aperiodic activity, capturing fluctuations at macro and micro levels of sleep structure to enhance our understanding of sleep dynamics.

Besides being relevant in improving the Introduction of the manuscript, the above detailed shortcomings are relevant for the section Discussion as well.

<https://doi.org/10.3109/00207459208999796>

<https://doi.org/10.1002/hbm.25129>

<https://doi.org/10.1007/s11571-019-09533-0>

<https://doi.org/10.1016/j.pneurobio.2024.102589>

We thank the reviewer for guiding us to this important literature. We have accordingly added this literature that we missed to the introduction and discussion.

2. One additional reason for changing the focus of the current literature mostly involved in the practice of fitting the log-log spectra with a linear unction in a narrow frequency range is evident from Figure 2C of the manuscript, but it is not explicitly claimed in the Introduction part of the paper. This panel of Figure 2 clearly indicates that narrow frequency ranges are more susceptible to the biases induced by fitting on incomplete spectral peaks (these biases are practically uncontrolled an non-visible when we consider time-resolved analyses). The latter issue was presented by Gerster et al (2022) under the subheading "Avoiding Oscillations Crossing Fitting Range Borders". Although this paper introduced the issue in the field, it provided a somewhat biased presentation of it, by emphasizing mainly the low frequency peaks instead of the very common high frequency ones, which could affect the narrow fits.

<https://doi.org/10.1007/s12021-022-09581-8>

We agree with the reviewer on this point, and to better acknowledge this point we have added to the discussion of frequency ranges where we cite the Gerster paper in the Introduction:

Page 3, Line 78: A key decision when examining aperiodic neural activity includes selecting the frequency range to analyze, including considering if chosen ranges could be biased by overlapping spectral peaks (Gerster et al., 2022).

II. Methods

1. Please specify the three additional articles added to the list of automatically identified and manually verified ones, as well as the way you selected them. Moreover, authors should indicate if these articles would indeed imply the addition of new search terms by using the OR operators (i.e. by widening the literature search).

We have revised this whole section also according to the PRISMA guidelines as requested by the editor. The new analysis is now explained in detail in section 2.1 of the revised manuscript.

Page 7, Line 160: We conducted a systematic review of the literature to identify reports examining aperiodic neural activity during sleep and to investigate the frequency ranges used in

such studies. The literature search was performed following the PRISMA 2020 guidelines (Page et al., 2021) using the LISC Python toolbox (v0.3.0) (Donoghue, 2019), which finds publications based on specified search terms in the Pubmed database. Literature searches collected reports published between 1929—the year of the first published EEG paper—and the end of 2023.

2.1.1. Systematic Review of Aperiodic Activity: To examine the prominence of investigations into aperiodic activity in neural data and how this has evolved over time, we collected the number of publications mentioning predefined terms related to brain activity and spectral properties. The predefined search terms included general descriptors of brain and sleep activity ("brain," "sleep") as well as specific terms related to spectral properties ("aperiodic exponent," "spectral slope," "1/f," and "power-law exponent"). **Eligibility Criteria:** Studies were included if they i) reported on brain activity or sleep, ii) included aperiodic or 1/f features, and, iii) employed electrophysiological techniques (EEG, MEG, iEEG). We excluded reports using consumer-grade devices, due to potential data quality issues. **Data Collection and Analysis:** Search terms were defined to search for reports with a co-occurrence of terms related to recordings (e.g., "brain," "sleep") and measures of aperiodic activity (e.g., "aperiodic exponent", "spectral slope"), using modality-related inclusion terms ("EEG", "MEG", "iEEG") to restrict results to relevant recordings. For analyses across time, results were extracted over two-year intervals. For visualization purposes, the time range was set to start at the year with the first non-zero number of publications. We provide a flow diagram illustrating the search in Suppl. Figure 1-1.

A.2.1.2. Frequency range for investigating the Spectral Exponent in Sleep: To examine studies that investigated the aperiodic exponent in the context of sleep, we ran an additional literature search to collect reports and metadata on such studies. The search terms specifically targeted studies examining the spectral exponent in sleep. These included variations of the spectral exponent ('spectral slope', 'spectral exponent', '1/f exponent', '1/f slope', 'aperiodic slope', 'aperiodic exponent', 'power-law exponent', 'power-law slope') combined with the keyword "sleep." **Eligibility Criteria:** We included studies that, i) focused on aperiodic features of neural activity, and ii) examined these features in the context of sleep. **Data Collection and Analysis:** The search yielded 21 studies, and metadata (publication year, title, abstract) were reviewed to confirm relevance. Five studies were excluded for the following reasons: i) consumer grade recording device (1 study), ii) absence of neural data (3 studies), or iii) non-human data (1 study). An additional five studies were excluded due to the absence of clear mention of the frequency bands used to estimate the aperiodic exponent. In addition, we manually added five studies meeting the inclusion criteria that were not detected by the literature searches. These studies were: Bódizs et al. (2021), Feinberg (1984), Lendner (2023), Miskovic et al. (2018), and Pereda et al. (1998). In total, 16 studies were included in our analysis, detailed in Table 1 of the supplementary material. We provide a flow diagram illustrating the search in Suppl. Figure 1-1B.

2. Authors mention that the non-cortical regions were excluded from further analyses of iEEG data and exemplify some areas which are indeed considered as cortical ones in the neuroanatomical literature (authors are asked to take a look at some brain atlases or review articles and search for the parahippocampal gyrus or the fusiform gyrus: both are clearly identified as cortical regions!). Please clarify the reason of excluding these regions which are of utmost interest in the neuropsychology of higher order cognitive functions!

To simplify the iEEG analysis, in the revised manuscript we have opted to include all cortical and subcortical regions in the iEEG analysis. In doing so, we first evaluated that there is no change

in the outcome of the analyses with the addition of the previously removed structures. We therefore include all regions available in the dataset.

The following Figures and tables have been updated:

- a) Figures 3, 3-1, 4, 4-2.
- b) Tables 2, 3, 4.

3. iEEG data: concatenated, non-continuous records (even if filled with artificially inserted zeros), are clearly not suitable for time-resolved analyses of ongoing brain activity. As a consequence, authors should not indeed use that iEEG data for this type of analysis, which cancels a significant ambition of their work. Alternatively, authors should clearly state if the concatenated records were used and how they were used during their work.

We would like to clarify that the time-resolved analysis in this study was exclusively conducted on the full-night, continuous EEG data. As detailed in the methods section, the iEEG data was used specifically to investigate model parameters and fits, focusing on changes in the knee and spectral exponent across different sleep stages. Time-resolved analysis was not applied to the iEEG data due to the concern raised by the reviewer.

To enhance clarity, we have emphasized this distinction between EEG and iEEG data processing in the methods section of the revised manuscript.

Page 11, Line 283: 2.4.1. PSD estimation across entire sleep stages (EEG and iEEG data)

Page 11, Line 297: 2.4.2. PSD estimation during sleep stage transitions (EEG data only)

Page 11, Line 304: 2.4.3. Analysis of Temporal Dynamics During Specific Events (EEG data only):

Regarding the concatenation of the iEEG data, in some recordings, the 60 s segments were not always continuous and consisted of concatenated segments (done by the lab that collected and released this dataset, see Frauscher et al., 2018). To minimize artifacts during concatenation, 2-s buffers were occasionally added between segments. For any given channel, a maximum of five segments were concatenated, resulting in a maximum of four 2 s buffers. This brought the total duration to 68 s (60 s of data + 8 s of buffers). To ensure consistency and reduce variability, channels with a total duration of less than 68 s were zero-padded at the end until they reached a total duration of 68 s. This ensured that all channels contained 60 s of data and 8 s of buffer, maintaining uniformity across recordings. For our analysis, PSD estimations were performed over entire epochs. As such, this standardization ensured no differences between channels, sleep stages, or participants due to variability in segment durations or the percentage of actual data in each segment, preserving the integrity and comparability of the results. This part is explained in more details now in the revised manuscript:

Page 8, Line 213: In some patients, continuous clean signals for 60 s were unavailable, so multiple segments were concatenated. To minimize artifacts, a 2-s buffer of zero amplitude was added between concatenated segments. Since a maximum of five segments were concatenated

in any given condition, a maximum of four 2-s buffers were included, resulting in a total duration of 68 s (60 s of data and 8 s of buffers). To ensure uniformity across recordings, channels with durations less than 68 s were zero-padded at the end to achieve a uniform length of 68 s for all channels and across all stages and patients. This standardization ensured consistent data and buffer durations across all recordings, preserving the comparability of PSD estimates across channels, sleep stages, and participants.

4. EEG data: "Finally, we performed independent component analysis (ICA) in EEGLab and visually labeled and removed noise-, heart-, and eye-related components."

Did authors record the ECG activity by using a specifically dedicated (ECG) derivation? This would ensure that the ECG artefacts are appropriately identified. Also, the records of eye movements and muscle tonus are parts of the minimal criteria of staging sleep. Were these measures (EOG, EMG) available? If not, how were the sleep records staged?

1) ICA and Heart Artifact Removal: There was no ECG that was recorded during the EEG experiment. Heart artifacts were removed using the standard ICA procedure. Specifically, ICA components were visually inspected, and those identified as resembling heart artifacts were excluded from further analyses. This approach, to the best of our knowledge, aligns with established practices for artifact removal.

2) Sleep Scoring: EOG and EMG channels were derived from the high-density 256-channel EEG cap, as the extensive number of electrodes made the placement of additional EOG and EMG electrodes impractical. For the EOG, two channels were extracted: one positioned above the right eye and the other one below the left eye, both referenced to the right mastoid. EMG channels were EEG electrodes located above the cheek muscles, one on each side of the face. Furthermore, sleep was scored using the SIESTA algorithm (Anderer et al., 2005; 2010), which has been shown to achieve high classification agreement with human scorers (Ameen et al., 2019). Scoring adhered to the AASM criteria (Berry et al., 2015), which require the inclusion of at least one EOG channel and a bipolar EMG channel.

We have clarified and updated this part of the methods section in the revised manuscript for accuracy and completeness.

Page 10, Line 263: *Sleep staging was conducted using two frontal (F3, F4), two central (C3, C4), two parietal (P3, P4), and two occipital (O1, O2) EEG electrodes. Two EOG channels were extracted from the high-density EEG cap: one positioned above the right eye and the other below the left eye, both referenced to the right mastoid. Bipolar EMG channels were recorded from facial muscles using EEG electrodes placed above the cheek muscles, with one electrode on each side of the face.*

5. PSD estimation is at the heart of the paper, but its description is unfortunately very vague. Authors claim the use of the Welch method, but miss the specification of some significant details (e.g. window type). Given this drawback, the reproducibility of these results is low.

We have revised this part of the manuscript, including all details on the performed PSD estimation analyses with Welch's procedure being described in the first paragraph:

Page 11, Line 283: *2.4.1. PSD Estimation Across Entire Sleep Stages (EEG and iEEG data): To estimate power spectra across entire sleep stage, we used Welch's method (Welch, 1967) on both EEG and iEEG data, with a 15 s window length with 50% overlap and tapered with a Hamming window. We created average PSDs per stages by taking the mean over all segments per epoch then averaging over all epochs. To compare the performance of the model at different frequency ranges and time windows of PSD calculation, we employed different time windows and frequency ranges for Welch's PSD calculations, with all other settings left unchanged. We used four different time windows (5 s, 10 s, 15 s, 20 s) and examined different frequency ranges covering either broad (1-30 Hz, 1-45 Hz, 1-60 Hz, 1-75 Hz) or narrow ranges (30-45 Hz, 25-45 Hz, 1-20 Hz, 1-8 Hz). These ranges were informed by the literature search. We report the results averaged over all sleep stages of iEEG (Wake, N2, N3, and REM) or EEG (Wake, N1, N2, N3, and REM) data. For the whole night time-frequency plot, we plotted the average value per epoch. Furthermore, we explored the changes in spectral parameters associated with transitions between sleep stages.*

2.4.2. PSD estimation Across Sleep Stage Transitions (EEG data only): For analyses across sleep stage transitions, the EEG data were segmented into 20 s intervals with a moving window of 2 s, resulting in a 90% overlap between adjacent segments. We then applied Welch's method to calculate the PSD for each of these segments using the same parameters outlined in section 2.4.1. We focused on segments falling within a 120 s timeframe around the transitions, extending from 60 s before to 60 s after each transition, to capture the spectral dynamics occurring around these changes in sleep stages.

2.4.3. Analysis of Temporal Dynamics During Specific Events (EEG data only): For measuring aperiodic activity during auditory stimulation and K-complex (KC) events, we computed frequency representations using a multitaper approach within the 1–45 Hz frequency range, with 0.5 Hz steps. We used Discrete Prolate Spheroidal Sequences (DPSS) as tapers and performed FFT-based convolutions with the number of cycles for each frequency set to match the frequency, resulting in a consistent time window of 1 s across all frequencies (e.g., at 1 Hz: 1 cycle → 1-s window, at 45 Hz: 45 cycles → 1 s window). EEG data were segmented into 10-s epochs centered around the event of interest (stimulus onset or KC onset). Bad epochs (amplitude fluctuations >1000 μ V) were excluded from analysis. To avoid edge artifacts, the first and last second of each epoch were discarded. We then baseline-corrected the resulting time-resolved aperiodic estimates by subtracting the mean of the estimates in the 500 ms pre-stimulus-onset window. We used a sampling rate of 250 Hz for all time-resolved analyses except for KC vs no-KC analysis, where we downsampled the data to 128 Hz, as KCs were scored on downsampled data. For the comparison between the exponent during stimulation and no-stimulation periods, one subject was removed from NREM and another from REM analysis due to poor model fit (R^2 values lower than 2 standard deviations below the mean). Thus, these analyses were conducted on 16 subjects. Further, for NREM, we restricted the analysis to only N2 and N3 stages, since brain responses to external stimuli are well documented during these stages.

a. The use of 15 s epochs (which are considerable longer than the ones frequently used in the sleep EEG field) do not always fith with the de-artefacted data, especially in the case of iEEG records, which were said to be concatenated. How were these concatenated and zero-filled segments PSD analysed?

We apologize for any missing information. We have expanded our description of the methods to provide a clearer methodological framework for how concatenated and zero-filled segments were created and further analyzed, ensuring consistency in PSD estimation.

Page 8, Line 213: *In some patients, continuous clean signals for 60 s were unavailable, so multiple segments were concatenated. To minimize artifacts, a 2-s buffer of zero amplitude was added between concatenated segments. Since a maximum of five segments were concatenated in any given condition, a maximum of four 2-s buffers were included, resulting in a total duration of 68 s (60 s of data and 8 s of buffers). To ensure uniformity across recordings, channels with durations less than 68 s were zero-padded at the end to achieve a uniform length of 68 s for all channels and across all stages and patients. This standardization ensured consistent data and buffer durations across all recordings, preserving the comparability of PSD estimates across channels, sleep stages, and participants.*

Page 11, Line 283: *2.4.1. PSD Estimation Across Entire Sleep Stages (EEG and iEEG data): To estimate power spectra across entire sleep stage, we used Welch's method (Welch, 1967) on both EEG and iEEG data, with a 15 s window length with 50% overlap and tapered with a Hamming window. We created average PSDs per stages by taking the mean over all segments per epoch then averaging over all epochs. To compare the performance of the model at different frequency ranges and time windows of PSD calculation, we employed different time windows and frequency ranges for Welch's PSD calculations, with all other settings left unchanged. We used four different time windows (5 s, 10 s, 15 s, 20 s) and examined different frequency ranges covering either broad (1-30 Hz, 1-45 Hz, 1-60 Hz, 1-75 Hz) or and narrow ranges (30-45 Hz, 25-45 Hz, 1-20 Hz, 1-8 Hz). These ranges were informed by the literature search. We report the results averaged over all sleep stages of iEEG (Wake, N2, N3, and REM) or EEG (Wake, N1, N2, N3, and REM) data. For the whole night time-frequency plot, we plotted the average value per epoch. Furthermore, we explored the changes in spectral parameters associated with transitions between sleep stages.*

b. PSD calculations of segments which are not conforming the length of the canonical power of 2 numbers (1024, 2048, etc): what kind of a procedure was used for handling this issue? Did authors always use Discrete Fourier Transformation or rather mixed-radix Fast Fourier Transformation? Another possibility is the zero padding procedure and the use of the common Fast Fourier Transformation routine. If this latter approach was used, how were the zeros added to the segments: before or after windowing (tapering)? What was the type of window used within the context of Welch method of PSD analysis? (See the same question above!)

PSD calculations were done in Python, using the MNE package which itself uses the scipy library for spectral estimations. Zero-padding and windowing was applied as described above,

without additional padding to a length of 2^n , as we opted not to improve run-times by padding to such lengths. Scipy uses Bluestein's algorithm for non factorizable lengths, which still has favorable run time.

Further information on the implementation is available in the scipy documentation:

<https://docs.scipy.org/doc/scipy-1.15.0/reference/generated/scipy.fft.fft.html>

c. The multitaper spectral analysis implemented for time-resolved parametrization of EEG activity is welcome, but should be specified during the description. The name of a script is far from an appropriate and accurate description. Scripts are changing, as do their baseline settings. As a consequence the technical term `mne.time_frequency.psd_multitaper` is only acceptable with the parallel use of an accurate scientific citation and specification of the settings. This would provide some control for the researcher degree of freedom in the field.

We would like to clarify that the report of the implementation used is not a reference to a script, but a reference to a stable and version-controlled implementation of the method used, that is uniquely accessible using the function name and version number in the version controlled MNE documentation. To add additional information, we have also revised the respective sub-section of the methods to include all relevant details regarding the time-resolved PSD estimation:

Page 11, Line 304: *For measuring aperiodic activity during auditory stimulation and K-complex (KC) events, we computed frequency representations using a multitaper approach within the 1–45 Hz frequency range, with 0.5 Hz steps. We used Discrete Prolate Spheroidal Sequences (DPSS) as tapers and performed FFT-based convolutions with the number of cycles for each frequency set to match the frequency, resulting in a consistent time window of 1 s across all frequencies (e.g., at 1 Hz: 1 cycle → 1 s window, at 45 Hz: 45 cycles → 1 s window). EEG data were segmented into 10-s epochs centered around the event of interest (stimulus onset or KC onset). Bad epochs (amplitude fluctuations >1000 μ V) were excluded from analysis. To avoid edge artifacts, the first and last second of each epoch were discarded. We then baseline-corrected the resulting time-resolved aperiodic estimates by subtracting the mean of the estimates in the 500 ms pre-stimulus-onset window.*

6. Spectral parametrization

"We calculated the knee frequency from the knee parameter, removing any values that were higher or lower than the mean \pm 2 standard deviations." What kind of values were indeed removed? The term "any value" refers to what? Knees, spectral bins, etc? Is the mean the sample mean? Please clarify!

The knee model was fit to the power spectral densities (PSDs) calculated for 30 s epochs of sleep stages. To ensure the robustness of the data and exclude outliers, any knee frequency value that was more than ± 2 standard deviations from the subject-specific mean was rejected.

This thresholding approach minimizes the influence of atypical or noisy data on subsequent analyses. This part is better explained in the revised methods section:

Page 12, Line 336: *We excluded knee frequency values that were more than ± 2 standard deviations from the mean knee frequency for each subject and each sleep stage. Further, we checked the goodness-of-fit (R^2) for all the spectral models. To examine potential changes in aperiodic parameters across the course of the night, we split the total number of epochs, per sleep stage, into quartiles, and examined average exponent values per quartiles across participants.*

III. Results

1. My first and most important critique refers to the interdependence of the model parameters (the knee frequency and the spectral exponent) and its interpretation. A strong correlation between model parameters is usually considered as an ill-conditioned statistical scenario, characterized by a contorted error surface where the minimum is not easily identifiable. As a consequence, the best fitting parameter values cannot be easily found (see Li et al., 1996 for a more general discussion of the issue). In general such a strong interdependence between the model parameters is non-coherent with the overall aims of modeling EEG/LFP spectra, which was motivated by reducing the redundancy of bin-power values (Bódizs et al., 2021). Instead of acknowledging the strong redundancy of the model authors are arguing for the strength and relevance of this finding, stating that former, fixed (no-knee) models were indeed incorporating the differences in knee frequencies. However, the Strong correlation between model parameters largely destabilize the basement of this argument, because we could derive a different (inverted) reasoning from these findings as well. Namely, knees could indeed reflect exponent values and not vice versa (correlation is non-directional). In this case, knees are reflecting the pressure to obtain a better fit, but they are indeed derived from the same underlying process which determines the spectral exponent (overfitting). Most probably the knee is best positioned to capture a white noise type of power distribution in the vicinity of zero frequency, but this distribution is not always of the white noise type (see also the fits provided by Colombo et al., 2019). My objection is also supported by the radical change in sleep stage-specific exponent values after introducing the knee parameter, resulting in a strongly atypical pattern of findings. Although these might be a real pattern, but could also emerge as a reflection of the ill-conditioned statistical setting characterized by the strong interdependence of the model parameters. Authors are requested to reflect on this issue and change their text from the current overenthusiasm type of readig to a more balanced and logical doubt in this respect.

<https://doi.org/10.1038/s41598-021-81230-7>

<https://psycnet.apa.org/doi/10.1037/0096-3445.125.4.360>

<https://doi.org/10.1016/j.neuroimage.2019.01.024>

To our best understanding, this comment relates to the motivation of the knee model as a model selection process and the correlation of the model fit parameters given the choice of this model, which we agree are key points our manuscript needs to motivate and address. We would like to take the opportunity to expand on these points, and have sought to clarify them in the manuscript.

We would first like to emphasize that the knee model is not a new proposal in this paper, but one much examined and discussed in previous work including (but not limited to) the following reports which motivate that it is well-posed and productive model:

- Miller, K. J., Sorensen, L. B., Ojemann, J. G., & den Nijs, M. (2009). Power-Law Scaling in the Brain Surface Electric Potential. *PLoS Computational Biology*, 5(12), e1000609. <https://doi.org/10.1371/journal.pcbi.1000609>
- Gao, R., van den Brink, R. L., Pfeffer, T., & Voytek, B. (2020). Neuronal timescales are functionally dynamic and shaped by cortical microarchitecture. *eLife*, 9, e61277. <https://doi.org/10.7554/eLife.61277>
- Lendner, J. D., Lin, J. J., Larsson, P. G., & Helfrich, R. F. (2024). Multiple Intrinsic Timescales Govern Distinct Brain States in Human Sleep. *The Journal of Neuroscience*, 44(42), e0171242024. <https://doi.org/10.1523/JNEUROSCI.0171-24.2024>
- Zeraati, R., Levina, A., Macke, J. H., & Gao, R. (2024). Neural timescales from a computational perspective (No. arXiv:2409.02684). arXiv. <http://arxiv.org/abs/2409.02684>

Notably these reports all support the applicability of the knee model to electrophysiological recordings and provide computational and physiological context for interpreting it (linking it to post-synaptic potential activity and auto-correlation / timescales). This prior work does not support the knee as reflecting white noise in the recorded signal – we are unaware of work that supports this interpretation, and note that adding a white noise term would generally lead to a flattening at the high-frequency end of the spectrum, but not at the low end of the spectrum as we observe and model here.

- **Model Selection:** Based on this prior work, we consider that the inclusion of the knee parameter is an established and a valid approach for capturing physiological phenomena in EEG data that fixed (no-knee) models cannot fully address. Theoretical and empirical studies have highlighted distinct underlying mechanisms driving changes in the knee frequency and the spectral exponent. For instance, Gao et al. (2017, 2020) demonstrated that the knee in the power spectrum is an expected property of power spectra of signals reflecting summed post-synaptic potentials, and that the knee relates to the auto-correlation of the data. Empirically, a study mentioned by the reviewer, i.e., Colombo et al. (2019), incorporated a knee, though indirectly, by measuring two different exponent ranges. We consider using the knee model to be consistent with this report, with the added benefit of explicitly modeling the knee frequency separate from the spectral exponent, as these likely reflect different aspects of the signal (see cited literature above). Moreover, both classic and contemporary research has emphasized the importance of the knee frequency or the bend in the PSD in characterizing various levels of consciousness. Early work by Nakata et al. 1993 identified the knee as a key indicator of physiological shifts in brain activity, while more recent findings by Lendner et al.

2024 illustrated how neural timescales—manifesting as knee frequencies—differentiate between distinct 1/f decay processes that correspond to varying brain states. These neural processes are governed by specific time constants, with knees acting as systematic inflection points that modulate the overall spectral slope.

Empirically, our approach is also motivated by the observation of a visible knee in the iEEG data, and to some extent in the EEG data. Note that if attempting to fit a power spectrum with a knee with a fixed model, the model is typically not only much less accurate, but will tend to overfit peaks to attempt to capture variance (see **Figure 2** of the manuscript and, for example, this tutorial), leading to a misspecified model that is dispreferred in explicit model comparison. We would also like to draw the reviewer’s attention to the added BIC comparison between the knee and fixed models in the iEEG data, which clearly indicates that the knee model outperforms the fixed one (**Figure 4I** of the revised manuscript - shown below).

Page 18, Line 515: *To compare the performance of the two models in the iEEG data, we used the Bayesian Information Criterion (BIC) for Gaussian distributions (Figure 4I), which is based on the likelihood function and evaluates model fit while penalizing complexity. BIC differences between the knee and fixed models favored the knee model across all sleep stages, as indicated by the negative difference (Knee - Fixed) values (Wake: -4.71, N2: -1.82, N3: -4.04, REM: -4.77). Additionally, to identify the stronger predictor of sleep stages (Knee frequency vs Exponent) within the knee model, we conducted a random forest analysis followed by permutation importance analysis (Fig. 4J). Results showed that the knee frequency (0.14 ± 0.07) was a significantly better predictor of stages than the exponent (0.07 ± 0.05) ($t(15) = -5.13$, $p < 0.001$, $d = 1.06$), emphasizing the significance of fitting a knee model to the data.*

Panels H, I and J of Figure 4 of the revised manuscript. (H) Correlation between exponents of both models, showing a weak, non-significant positive correlation. (I-J) Comparison of aperiodic models and features in iEEG data. I) BIC comparison between the knee model and the fixed model indicating that the knee model provides a better overall fit. The left panel presents absolute BIC values for both models across sleep stages, while the right panel shows their difference (knee - fixed). Dots represent mean values, and lines indicate 95% confidence intervals. (J) Random forest feature-importance analysis comparing the knee frequency and the exponent of the knee model, indicating greater importance of the knee frequency. Dots represent mean values, and lines show the standard error of the mean (SEM).

- **Addressing parameter correlations:** based on the above point, we do think the knee model is well motivated, and that fitting the knee parameter enhances the model's ability to

capture spectral dynamics. That said, the correlations between the parameters are notable and do pose conceptual and analytical challenges. We want to emphasize that we do not find that this correlation suggests these features are interchangeable or redundant in that removing the knee parameter leads to worse model fits and the overfitting of peaks, which is worse in terms of using modeling for a parsimonious description of the power spectrum.

Beyond parsimonious descriptions, we are interested in modelling to investigate putative interpretations of the power spectrum, which the knee model also allows for. As noted above, a key motivation for the knee model is a model of LFP as derived from post-synaptic potentials. To use this framework to further address the issue of the correlation between parameters, we simulated data using the ***sim_synaptic_current*** function from **NeuroDSP** toolbox (Cole et al., 2019), which implements the model from the Gao papers mentioned above. This function generates simulated synaptic current signals that mimic neuronal activity by modeling the interaction of excitatory and inhibitory postsynaptic potentials. This approach allowed us to independently examine the relationship between the knee frequency and the spectral exponent. The results, visualized in the correlation plot (Figure R1 of this document), show that by manipulating the parameters of the postsynaptic potentials that generate the simulated current, the model generates varying knee and exponent values, which are found to have a moderate positive correlation ($r = 0.39$, $P < 0.001$). These simulated results suggest that the observed relationship can be related to the assumed generative model of the data, rather than necessarily reflecting issues with fitting the model.

Figure R1. Correlation between the knee frequency and spectral exponent in simulated data. The data was generated using the function sim_synaptic_current function from NeuroDSP. Specifically, we generated signals with a sampling frequency of 250 Hz and a duration of 30 s from a combination using

varying model parameters. The function uses two key parameters, tau_rise and tau_decay, to define the rise and decay time constants of simulated post-synaptic potentials. The tau_decay parameter was sampled uniformly from between 0.01 and 0.1 s and tau_rise from between 0.001 and 0.009 s, reflecting physiologically plausible ranges (see Gao et al., 2017). This model creates power spectra with a Lorentzian shape and is the motivation for the knee model we used. For each sampled simulation, we computed PSD and fit the spectrum with the knee model between 1 and 45 Hz as in the manuscript. The Spearman correlation between the knee frequency and the exponent revealed a significant correlation ($r = 0.39$, $p < 0.001$).

In the above described simulations, the input parameters reflect the properties of the simulated post-synaptic potentials, and varying these parameters affects both the measured exponent and knee frequency. This model also reflects the putative interpretations of these features, with the balance between the number of excitatory and inhibitory potentials relating directly to the exponent (beyond the knee) and the lengths of the potentials relating to the knee (and therefore the auto-correlation or ‘timescale’ of the signal). The key point here is that we find that some degree of correlation between the measured parameters is consistent with the assumed generative model of the data, and need not relate to a miss-specified or miss-estimated model. More broadly, the correlation between the exponent and the knee parameter can be taken as a suggestion that there is some relationship between the putative sources of their variation, namely, that the E/I balance and auto-correlation / time scale of a circuit are not fully independent. Overall, while we fully agree that future work on understanding the relationships between these features is important, we hope these points emphasize that we consider the modelling choices taken in this manuscript to be motivated and productive.

- Difference between our reported stage-specific exponent changes and previous accounts: The difference in stage-specific changes in spectral exponent values observed after introducing the knee parameter could actually reflect the opposite of an ill-conditioned statistical condition. In the iEEG data, a simple visual inspection of the PSDs—whether at the single region, single subject level, or even at the average level—strongly supports the use of the two-exponent (knee) model, as one can clearly observe a shift in the exponent. In such cases, employing the knee model appears to be the most appropriate approach, while the use of a fixed exponent would require explicit justification, either to simplify computations or to prioritize simplicity. Here, we aimed to move beyond these simplifications and focus on approximating the ground truth. In fact, previous work, either used a narrow band (though removing the possibility of knee measurements) such as that by Lendner et al. (2020), or often overlooked the broader shape of the PSD. In contrast, by explicitly modeling the knee parameter, we can uncover previously unaccounted stage-specific dynamics, offering deeper insights into the underlying neural processes. This approach aligns with recent empirical evidence from Lendner et al. (2024), who demonstrated that the aperiodic (intrinsic) timescales of iEEG PSDs vary across stages, thus motivating a similar approach to ours.

In light of the reviewer’s critique, we have revised the discussion to adopt a more balanced perspective. While we remain confident in the utility of the knee model, and the important insight of the sleep-stage-induced changes of the knee frequency, we have tempered our language to acknowledge potential concerns about parameter interdependence and its implications for

model robustness. Additionally, we have included a discussion on the potential for overfitting and the need for further validation of the knee parameter across diverse datasets.

Page 31, Line 856: *The relationships of the aperiodic features to each other, within and between the different models, and their differing patterns across stages emphasize that these different models and features are related, but not equivalent. This relates to the impact and importance of the knee – while a true $1/f$ signal has the same exponent value across all frequency ranges (which can also be called mono-fractal or scale-free), the knee reflects a frequency-specific transition in the power spectrum. In such a case, which can be called multi-fractal or multi-scale, the knee parameter reflects a transition between $1/f$ regions and the exponent of the knee model reflects the frequency region beyond the knee frequency. Even when models are fit across the same frequency range, their exponents show only weak non-significant correlation, further confirming that they capture different spectral properties and may lead to distinct interpretations. This suggests that aperiodic parameters cannot be interpreted in isolation, for example, the exponent from a knee model should be considered in relation to the measured knee and may differ from the exponent of a fixed model. While the results here support the use of the knee model, further work is needed to relate results and interpretations of the aperiodic exponent measures of these different models, as well as in comparison to previous work that typically examined narrower ranges.*

Page 32, Line 879: *Our results suggest that sleep-related changes in aperiodic activity involve both E-I balance and neural timescales. While E-I dynamics inferred from the exponent reflect global changes between wake and sleep, knee frequency more effectively differentiates sleep stages and explains exponent changes when using a single-exponent model. This pattern is particularly clear in the intracranial data, in which measurements of neural timescale from local field potential (LFP) data have been established (Gao et al., 2020), though similar patterns are observed in EEG data, indicating the potential for extracranial recordings to measure neural timescales as well.*

2. Authors fail to provide parallel empirical evidence of sleep stage and time-related effects in gold standard, band-limited spectral indices, most notably Slow Wave Activity (SWA: 0.75-4.5 Hz). This would also help to clarify somehow the ambiguity concerning the knee vs exponent issue in many respects. Also SWA-differences might provide the readers with a gold standard type of knowledge in cases where effects like decreasing N3 sleep EEG exponents along the night imply a need for additional insight (e.g.: Is this effect paralleled by a decreasing sleep intensity?).

We appreciate the reviewer's insightful suggestion regarding the inclusion of slow wave activity (SWA) as a gold-standard measure of sleep intensity. We fully acknowledge the importance of SWA in characterizing sleep-stage dynamics and its potential role in providing additional context for our findings.

However, our analysis is specifically focused on electrophysiological signal components that extend beyond predefined frequency bands, such as the aperiodic exponent and knee frequency, to offer a broader understanding of spectral dynamics across the night. While we recognize that SWA differences could contribute valuable context, we do not believe that SWA

measures would directly relate to or enhance the explanatory power of the aperiodic parameters examined in our study. Our methodological approach is validated for distinguishing aperiodic activity from periodic components like slow waves, and the parameters we assess are thought to reflect distinct neurophysiological processes compared to those captured by SWA.

Furthermore, we would like to emphasize that our analyses are grounded in gold-standard sleep scoring methods, ensuring a robust framework for interpreting stage-specific changes in our data. Given the already extensive scope of our manuscript, we believe it is important to maintain a clear focus on aperiodic parameters. That said, we agree that future research should explore the integration of periodic and aperiodic measures to further elucidate their interplay during sleep.

We appreciate the reviewer's thoughtful feedback and hope that our clarification adequately addresses this point.

3. Sleep stage transitions in terms of fixed vs knee-model

"The results are broadly consistent, although less reliable and temporally precise as the exponent of the knee model, suggesting that the knee model may be well posed for detecting variations during the transitions between sleep stages." Such statements should be based on empirical and statistical evidence. How were the difference between reliability and temporal precision tested? Are the differences in d-values strong enough to support this claim? Please indicate!

We have amended this part of the results to clarify these points the difference in the time-resolved results using the fixed model:

***Page 24, Line 665:** Overall, these results align with the expected changes in the aperiodic exponent observed in non-time-resolved analyses and highlight the temporal precision of these changes across sleep stages. Additionally, we conducted time-resolved analyses of sleep stage transitions using both the exponent of the fixed model and the knee frequency (see Suppl. Figures 6-2 and 6-3). The results are broadly consistent, and we found no statistically significant difference in the model fits (R^2 values) between the knee and fixed models (Paired t-test: $t(16) = 1.885$, $p = 0.108$, $d = 0.22$). Although the exponent derived from the fixed model showed differences across time, these were less temporally related to the transitions between stages.*

IV. Discussion/Conclusion

1. Please consider the fact that sleep stage-related differences in aperiodic activity are not the only category of empirical findings which might be relevant for the interpretation of these findings. The formerly reported overnight changes in the spectral exponent during successive sleep cycles (Horváth et al., 2022) indicate that the time-related types of finding reported by the authors in the current paper are not without antecedents.

We thank the reviewer for highlighting the relevance of previously reported overnight changes in the spectral exponent during successive sleep cycles. To address this point, we have revised the discussion to incorporate previous work, including Horváth et al. (2022), to provide a more comprehensive framework for understanding the overnight dynamics of the spectral exponent and their relationship to sleep stages and cycles.

Page 33, Line 909: *Previous research has shown that the spectral exponent of NREM sleep flattens across successive sleep cycles (Horváth et al., 2022) and that time-resolved aperiodic activity can distinguish between healthy and disordered sleep (Bódizs et al., 2024). Building on these findings, our work refines measurement methods for aperiodic activity, capturing fluctuations at macro and micro levels of sleep structure to enhance our understanding of sleep dynamics.*

2. Equality/non-equality of the changes in stimulus-evoked aperiodic activity and K-complexes The reasoning of the authors is largely considering their K-complex detecting procedure as a gold standard or even some ground truth. Evoked K-complexes could rather be considered as statistical phenomena, but not some all-or-nothing event. Authors might consider the fact that the EEG results from the admixture of multiple neural sources, thus the significant stimulus-induced changes in aperiodic activity in the absence of a detected K-complex should be considered within this context (as well).

We would like to assure the reviewer that we did not present our method for the detection of K-complexes as a gold standard or definitive ground truth. We acknowledge that, due to the strict amplitude criterion used in our procedure, some K-complexes might not have been detected. This limitation means that the observed changes in the spectral exponent in the absence of K-complexes, could potentially be attributed to undetected "K-complexes". Furthermore, we fully agree with the reviewer that EEG results reflect the admixture of multiple neural sources, and significant stimulus-induced changes in aperiodic activity, even in the absence of a detected K-complex, should be interpreted within this broader context. In response to this valuable feedback, we have revised the corresponding section of the discussion to provide a more comprehensive interpretation of the results.

Page 34, Line 934: *Our analysis of evoked responses to external stimuli revealed transient, event-related steepening of the aperiodic exponent during NREM sleep, suggesting an increase in neural inhibition that may support sleep maintenance (Ameen et al., 2022; Blume et al., 2018). Notably, the responses were stimulus-specific, with greater steepening following unfamiliar voices compared to familiar ones, consistent with prior findings of similar trends during NREM sleep (Ameen et al., 2022). Crucially, we observed similarities between exponent responses, ERPs, and elicited KCs, which raises questions about their independence and overlap. While our analyses suggest an aperiodic component contributes to evoked KCs, exponent changes were also observed in trials without KCs and appear partially distinct from ERP responses. Additionally, our KC detection method, based on a strict amplitude criterion, may have missed weaker or atypical KCs, potentially influencing the observed spectral changes.*

3. The significant positive deflection following UFVs
Is this finding coherent with the literature? (E.g.: Perrin et al., 2000)

https://journals.lww.com/neuroreport/abstract/2000/06050/functional_dissociation_of_the_early_and_late.8.aspx

Our ERP findings align with previous research, showing significant effects of voices but no differences between names, consistent with patterns documented in the literature (Ameen et al., 2022; Blume et al., 2018). However, this differs from Perrin et al. (2000), who observed a specific differential response to the subject's own name. This discrepancy may stem from methodological or contextual differences, such as stimulus presentation or preprocessing steps, which could influence name recognition effects. A detailed exploration of these factors has been addressed in prior work (Ameen et al., 2022).

Reviewer #2 (Remarks to the Author):

The work by Ameen and colleagues studies how the spectral exponent and knee of power spectra change during sleep. They present different models and frequency ranges to fit the exponent and time-resolved analyses using a variety of datasets. The manuscript deals with an important investigation of exploring how the exponent changes during sleep, with which model or frequency ranges to fit. All of these are valuable as the literature on the exponent has rapidly expanded. There are a few methodological and theoretical questions to address.

We thank the reviewer for their valuable feedback, we address the concerns in detail below.

Major points

1. The introduction presents a broad overview of the literature which eventually lays out some objectives (expand previous research & broaden the investigation of aperiodic activity during sleep; evaluating frequency ranges and model forms, etc). However, there are no hypotheses on what is expected out of the presented analyses, which then makes the interpretation of findings challenging, as it is hard to evaluate what is surprising. This may be a methodological work, but still authors likely had some expectations and hypotheses when starting it out.

In addition to addressing the editor's request to label the empirical aspects of this work as exploratory, we have amended the introduction to attempt to clarify the goals and hypothesized results of the analysis in this project. Specifically, our expectations centered around the improvement of aperiodic modeling (e.g., exploring the role of the knee parameter and frequency range adjustments) to better capture time-resolved changes in aperiodic activity during sleep. We have revised the introduction to articulate these objectives, including in the following edits:

Page 6, Line 134: Specifically, in this exploratory, data-driven investigation, we evaluated different frequency ranges and model forms for examining aperiodic activity in sleep recordings. We then applied these measures in a time-resolved manner to examine dynamics across multiple timescales throughout the night. In doing so, we sought to explore the temporal dynamics of aperiodic neural activity during sleep, hypothesizing that changes in such activity would track sleep stage transitions.

Page 15, Line 419: A primary aim for this study was to broaden the scope of analyzing aperiodic activity during sleep. Our initial step involved a systematic investigation into how the selection of frequency ranges and model forms influences the results.

2. When analyzing the exponent with the knee model in EEG vs. iEEG data there

seems to be a key difference: the iEEG exponent decreases from wake to N2/3 and REM (Fig 4D), while the EEG exponent increases (Fig 5C). This is currently not described in the results as such, as the results describing these figures mention that the exponent differs between stages, without highlighting the direction of the difference. Could authors describe this finding more explicitly in the results? Authors should also discuss why there is this difference in the change of the exponent depending on whether one is analyzing intracranial or scalp EEG. Moreover, this discrepancy is not found with the fixed model. Is this because the model is more reliable, or because it is less sensitive and thus does not uncover a key feature in the data?

We thank the reviewer for this remark on clarifying the differences in the parameters between models and modalities. Indeed, the direction of change in the spectral exponent differs between EEG and iEEG data for the knee model but not for the fixed model.

These findings are now more explicitly mentioned in the results section:

- **[iEEG] Page 17, Line 471:** *Post-hoc tests revealed that the exponent decreased significantly from wakefulness to N2 ($p_{\text{bonf}} < 0.001$), to N3 ($p_{\text{bonf}} < 0.001$), and to REM ($p_{\text{bonf}} < 0.001$).*
- **[EEG] Page 20, Line 559:** *Using the knee model, we observed significant differences in the exponent across sleep stages, similar to the findings from the iEEG data (Figure 5C left; $X^2 = 58.21$, $p < 0.001$, $W = 0.86$). However, unlike the iEEG results, the knee model showed an increase in the exponent from wakefulness to sleep, with further variations observed between different sleep stages.*

We acknowledge the notable observation that the direction of change in the exponent within the knee model varies in iEEG depending on the chosen model. However, we do not believe this variability reflects issues with model reliability, as our model selection analysis supports the appropriateness of the Knee Model for iEEG data (see Figure 4I of the revised manuscript)

The reviewer raises a similar concern, along with other points, in the next comment. Therefore, we will provide a more comprehensive response addressing this issue alongside the other points in our reply to the next comment.

3. For iEEG data, the direction of change in the exponent from wake to sleep differs depending on whether the knee or fixed model are used. With the knee there is a decrease from wake to sleep, while with the fixed model an increase (Fig 4D/E). This is also not really discussed. How can this be explained? Can authors add a more detailed description of this finding?

We agree these findings warrant more discussion, following up from the previous point (point 2 of the reviewer) in which we added a clearer description of the iEEG results to the manuscript. These results – in which the knee model shows a decrease in the exponent from wake to sleep, while the fixed model shows an increase in the iEEG data (Figure 4D/E) – is intriguing.

We have also added additional text to the discussion section to further note the differences between the two models in terms of their results and interpretations, emphasizing that the models are different in the sense that the exponent estimates from the two are not equivalent or interchangeable. While we currently lack a definitive explanation for the differences between models, and how they relate to previous work we think this highlights these important differences between the two models. This reinforces the importance of carefully selecting model parameters, as the choice significantly influences the interpretation of spectral data. The knee model accounts for an additional parameter, capturing spectral changes that are not accommodated by the fixed model, and in doing so changes the outcome of the measured exponent. If and when a knee model is well motivated (as we believe is the case here, based on our model selection analyses), these differences in the modeled parameter space can lead to differences in the results and putative conclusions about neural activity. We now further discuss this in the manuscript and suggest further work is needed to further qualify the relationships between model forms, frequency ranges, and putative interpretations.

In response to this remark, we have expanded the corresponding discussion accordingly:

Page 31, Line 848: *While our findings partially align with previous observations, demonstrating that the exponent differs significantly across the sleep-wake cycle, the direction of these differences varies depending on the model used. Specifically, when fitting a fixed exponent model to both iEEG and EEG sleep data, we observed a progressive steepening of the exponent from wakefulness to sleep. In contrast, iEEG data analyzed with the knee model showed a decrease in the exponent from wakefulness to sleep, with no further changes observed during sleep. This discrepancy reflects distinct features of the data that are captured by the knee and fixed models. The relationships of the aperiodic features to each other, within and between the different models, and their differing patterns across stages emphasize that these different models and features are related, but not equivalent. This relates to the impact and importance of the knee – while a true 1/f signal has the same exponent value across all frequency ranges (which can also be called mono-fractal or scale-free), the knee reflects a frequency-specific transition in the power spectrum. In such a case, which can be called multi-fractal or multi-scale, the knee parameter reflects a transition between 1/f regions and the exponent of the knee model reflects the frequency region beyond the knee frequency. Even when models are fit across the same frequency range, their exponents show only weak non-significant correlation, further confirming that they capture different spectral properties and may lead to distinct interpretations. This suggests that aperiodic parameters cannot be interpreted in isolation, for example, the exponent from a knee model should be considered in relation to the measured knee and may differ from the exponent of a fixed model. While the results here support the use of the knee model, further work is needed to relate results and interpretations of the aperiodic exponent measures of these different models, as well as in comparison to previous work that typically examined narrower ranges.*

4. The authors report a strong positive correlation between the variance in R2 and the variance in the exponent (Fig 3E) and offer important insight into the possibility that fitting spectral models to broad frequency ranges offers increased reliability. For the fixed model in particular, but also for the knee model, the relative R2 and exponent between brain states appear similar (Supp Fig 4-2 and Fig 4 D-E). For the fixed model,

when R^2 increases, so does the exponent. Further, the confusion matrices in Fig 3B and Supp Fig 3-1B both appear to trend towards a positive correlation between the exponent and R^2 : higher R^2 and higher exponent value with a shorter time window. Can the authors report the correlation between R^2 and the exponent? If there is dependence between the two variables, the authors can consider including a point in the discussion on whether this dependence is an epiphenomenon of model fits being “better” for more rapidly decaying power spectra or if the possible dependence reflects “worse” fits being biased towards higher exponents.

We thank the reviewer for this insightful comment. At first glance, the confusion matrices may suggest a potential relationship between the spectral exponent and the R^2 values. Motivated by this observation, we analyzed the correlation between the exponent and R^2 across the different fits used in our models (refer to **Figure R2** below). However, our analysis revealed no significant correlation in any of the models, indicating no systematic relationship between the model R^2 and the measured exponent values.

We have included these correlation analysis in the revised manuscript (Suppl. Figure 3-3) and expanded the results section to emphasize the absence of bias.

Page 16, Line 442: *Further, we found no correlation between the R^2 and exponent values across all models, indicating that the exponent values are not systematically biased by model fit quality (Suppl. Figure 3-3).*

Figure R2 (Figure 3-3 of the revised manuscript). Correlation between spectral exponent and R² values across different model fits. Scatter plots illustrate the relationship between the spectral exponent and R² for four different model configurations: the Broadband Knee Model (top-left), the Broadband Fixed Model (top-right), the Narrow-range Knee Model (bottom-left), and the Narrow Fixed Model (bottom-right). Each plot includes a regression line with a 95% confidence interval. Correlation coefficients (r) and p-values are displayed for each model. No significant correlation was found across any model, indicating the absence of a systematic bias or fitting artifact.

5. In discussion, the theoretical implications of changes in the exponent are laid out. However these can be better linked to the present findings. It is mentioned for instance that “Our findings here are at least partially consistent with these findings, whereby across all the different model settings this general pattern of exponent differences between Wake, NREM, and REM stages is observed.” What does partially consistent mean? Can authors elaborate which part of the results follows previous work and which not, and possible interpretations for differences? One critical parameter in the present work is the fitting range which differs compared to that of previous studies, which have

established links between excitation-inhibition and exponent based among other things on simulated data. Given that the direction of change in the exponent is quite different depending on modality & model as well as the fitting range compared to previous studies, direct links between excitation-inhibition balance, exponent, and sleep dynamics seem more difficult to establish. Could authors (a) discuss more clearly which of their findings are partially consistent with previous work; (b) implications of such discrepancies on the interpretability and theoretical framing of the exponent for future studies?

The reviewer raises two crucial points that are central to the methodological aspects of this study.

1) Consistency with previous literature:

Our findings partially align with prior work by showing consistent differences in the exponent between wakefulness and sleep across models. However, the direction of these differences varies depending on the model used. This partial consistency arises from our use of the knee model and broad frequency bands, which were chosen to provide a holistic, data-driven reflection of the underlying neural dynamics. We note that visual inspection suggests that the knee model more closely represents the underlying PSDs, especially in iEEG and, in some cases, EEG. This added flexibility of the knee model enables it to capture subtle variations in aperiodic activity with greater specificity, which fixed models may overlook. However, these methodological differences, particularly the broader frequency fitting range, make it challenging to directly compare our results with prior studies that used narrower ranges and single-exponent models.

Page 31, Line 848: While our findings partially align with previous observations, demonstrating that the exponent differs significantly across the sleep-wake cycle, the direction of these differences varies depending on the model used. Specifically, when fitting a fixed exponent model to both iEEG and EEG sleep data, we observed a progressive steepening of the exponent from wakefulness to sleep. In contrast, iEEG data analyzed with the knee model showed a decrease in the exponent from wakefulness to sleep, with no further changes observed during sleep.

2) Interpreting the results:

We agree that using a different model and frequency range complicates the contextualization of our findings within the framework of previous literature and support treating these two models as independent yet overlapping. This challenge is particularly evident when comparing the exponents derived from the fixed model and the knee model, as these parameters are measured differently, are largely distinct, and may reflect different neurophysiological processes. Moreover, the knee model accounts for bi-dimensional variations in aperiodic activity—capturing both the exponent and the knee frequency—thereby reflecting stage-dependent dynamics, including processing speed and E-I balance. In contrast, the fixed model overlooks changes in knee frequency or neural timescales, potentially oversimplifying the data. Consequently, direct comparisons between these models must consider their methodological and conceptual differences, which stem from their distinct underlying assumptions and sensitivity to neural

processes. In this regard, the knee model provides valuable insights by offering a different perspective on the data. While it may complicate the narrative, this complexity is justified as it enables a more accurate reflection of the underlying neural processes.

As highlighted in our previous comment, we have addressed this issue in the revised manuscript:

A) Interpreting the differences between model parameters

Page 31, Line 855: *This discrepancy reflects distinct features of the data that are captured by the knee and fixed models. The relationships of the aperiodic features to each other, within and between the different models, and their differing patterns across stages emphasize that these different models and features are related, but not equivalent. This relates to the impact and importance of the knee – while a true $1/f$ signal has the same exponent value across all frequency ranges (which can also be called mono-fractal or scale-free), the knee reflects a frequency-specific transition in the power spectrum. In such a case, which can be called multi-fractal or multi-scale, the knee parameter reflects a transition between $1/f$ regions and the exponent of the knee model reflects the frequency region beyond the knee frequency. Even when models are fit across the same frequency range, their exponents show only weak non-significant correlation, further confirming that they capture different spectral properties and may lead to distinct interpretations. This suggests that aperiodic parameters cannot be interpreted in isolation, for example, the exponent from a knee model should be considered in relation to the measured knee and may differ from the exponent of a fixed model. While the results here support the use of the knee model, further work is needed to relate results and interpretations of the aperiodic exponent measures of these different models, as well as in comparison to previous work that typically examined narrower ranges.*

B) Interpreting the difference between modalities

Page 32, Line 887: *By analyzing sleep data from intracranial (iEEG) and EEG recordings, we identified both similarities and modality-specific differences that warrant further investigation. The knee parameter was more pronounced in iEEG but less consistent in EEG with model selection comparisons suggesting that the knee model fits iEEG data better overall, whereas in EEG, no single model emerges as universally superior. However, EEG showed a clear knee during certain sleep stages, with a better knee model fit in Wakefulness, REM, and N1, indicating that the prominence of the knee parameter may be sleep-stage-dependent. These findings highlight the need for further research into stage-specific knee dynamics in EEG. While stage-dependent knee frequency has been observed in other contexts (Lendner et al., 2024; Nakata et al., 1993), its role in sleep spectral dynamics remains underexplored. This underscores the importance of data-driven model selection, prioritizing broadband frequency ranges and robust goodness-of-fit measures to improve reproducibility and model selection across neural states. Future studies should further examine knee occurrence in sleep EEG and explore how exponent changes relate to knee shifts across sleep stages, as observed in iEEG, and seek to further understand the differences between modalities.*

C) Interpreting the difference in the knee frequency between sleep stages

Page 32, Line 872: *In applying the knee model, this study highlights notable changes in the knee parameter across sleep stages, which provide distinct insights into neural dynamics. The knee frequency, reflecting the "timescale" of neural processing, maps directly to the decay time constant from the autocorrelation function (Miller et al., 2009; Gao et al., 2020). N3 sleep*

exhibited the lowest knee frequency, indicating the longest processing timescale, while REM sleep and wakefulness showed higher knee frequencies, suggesting faster processing. These findings align with prior sleep-related autocorrelation analyses (Zilio et al., 2021) and timescale estimates from spiking activity (Hagemann et al., 2022). Our results suggest that sleep-related changes in aperiodic activity involve both E-I balance and neural timescales. While E-I dynamics inferred from the exponent reflect global changes between wake and sleep, knee frequency more effectively differentiates sleep stages and explains exponent changes when using a single-exponent model. This pattern is particularly clear in the intracranial data, in which measurements of neural timescale from local field potential (LFP) data have been established (Gao et al., 2020), though similar patterns are observed in EEG data, indicating the potential for extracranial recordings to measure neural timescales as well.

Minor points

1. It is unclear which iEEG data are used in section “3.1. Model selection for estimating aperiodic activity.” Is this across all data segments or within a specific brain state?

We apologize for this clarity issue. For the model selection procedure, we used the entire iEEG dataset, including all channels, all stages, and all regions. This clarification has been incorporated into the revised manuscript:

Page 15, Line 424: For this analysis, we used the iEEG dataset, including all patients, regions, and sleep stages, to fit a fixed exponent model to either broad (Figure 3A) or narrow frequency ranges (Figure 3B). This approach allowed us to evaluate the stability of the models across varying conditions.

2. In section 3.3. BIC differences between knee and fixed models are reported in parentheses (paragraph 4). It should be made more explicit that the values in parentheses are differences between models.

We thank the reviewer for this remark and have therefore improved the articulation of this part:

Page 18, Line 517: BIC differences between the knee and fixed models favored the knee model across all sleep stages, as indicated by the negative difference (Knee - Fixed) values (Wake: -4.71, N2: -1.82, N3: -4.04, REM: -4.77).

Page 21, Line 586: Specifically, we calculated the BIC differences between the knee and fixed models (Knee - Fixed) in EEG data. The results indicate strong evidence for a better fit of the knee model during Wakefulness (-8.22), N1 (-6.77), and REM sleep (-14.9). In contrast, the fixed model provided a better fit during N2 (45.52) and N3 (9.95), as reflected by lower BIC values for the fixed model.

3. In section 3.4 it only becomes clear which of the two exponent models is used at the

end of the section when authors refer to the supplemental analyses. Can you clarify earlier in the section?

The choice of the model is now mentioned earlier in the text:

Page 23, Line 623: *Building on the observed differences in aperiodic parameters across sleep stages, our next aim was to explore the temporal dynamics of aperiodic activity during sleep (Figure 7A). Using the knee model, we examined changes of the exponent across time.*

4. Figure 6A: Can you reduce the line width of the exponent trace as it is currently not possible to clearly see the state-dependent differences in the exponent?

The lines have been adjusted following the reviewers' suggestion.

5. Figure 4C: I suppose that the individual colors in the line plots correspond to brain regions. As adding a legend seems difficult given the number of lines you could consider recoloring them to the same color to not overlap with sleep stages and create confusion.

We initially explored using a monochromatic palette to reduce this overlap. However, we found that this significantly reduced the clarity of individual lines, as the high number of PSDs and their similarity led to a smeared appearance, making it difficult to distinguish between different regions. To improve visualization while maintaining differentiation, we opted to retain the current color scheme. However, we have now decreased the widths and increased the transparency of the individual lines to enhance clarity.

First of all, we would like to thank the reviewers and the editor for their help improving the manuscript. We have carefully considered all suggestions and made the corresponding revisions throughout the manuscript. A point-by-point response is provided below:

REVIEWER 1

The clarity and scientific value of the paper increased considerably. Indeed, there are still some points which I think were not appropriately addressed during the process of the manuscript revision.

We thank the reviewer for their helpful feedback. We apologize if we did not properly address any of the reviewer's points. Our detailed response will follow in this document.

1. This point is about the Lorentzian function-based model of the (i)EEG power spectrum. In their rebuttal authors claim that timescales and auto-correlations are reflected by the knee-parameter derived from their model. We agree. But we do not agree in our views on the overall shape of power spectra below the knee frequency and the corresponding time series features. Authors implicitly and perhaps involuntarily opt for a flat spectrum-type model in the vicinity of zero frequency (Fig. 4B), and for the correlated model variables (knee frequency correlated with the spectral slope steepness in some sleep stages). Although authors included a particularly large number of additional sentences and analyses in their revised manuscript, their claims are basically unchanged. Let me exemplify these points by the following remarks:

Thank you for this important point. We would like to clarify that we agree with the reviewer that the spectrum below the knee likely exhibits a non-zero slope, and that this characteristic might be relevant for capturing certain aspects of the underlying time series dynamics. In our work, we sought to motivate and emphasize the goal of exploring spectral models of sleep data that allowed for exploring different forms, while examining different frequency ranges. To do so, we employed a model that has precedence and that is available to do so - the Lorentzian model. We do acknowledge this model choice has limitations, particularly that its assumptions of a flat pre-knee slope may not always fully capture the shape of neural power spectra near 0 Hz. That is, our goal was simply to move beyond the more commonly used single-exponent models, by introducing an additional parameter (the knee) in a way that is currently available in existing methods. We sought to motivate the measurement of the knee as it reflects an important signal feature which is the transition between low- and high-frequency scaling regimes as well as a physiological phenomenon that is processing timescales.

We do acknowledge that this modeling choice does not always fully capture the low frequency regions. We view this as a compromise within current modeling constraints, rather than an endorsement of flatness in neural data per se. Importantly, however, we do not believe this modeling choice affected the central claims of our study, which address systematic variations in the knee frequency and post-knee slope across conditions. We will elaborate on this further in

our detailed responses to the reviewer's subsequent remarks, and have sought to add some additional notes on this in the manuscript.

a. If the reader focuses on the reported data and considers panel A of Figure 4., which depicts the "Average PSDs of the different sleep stages of the iEEG data", (s)he can see slopes of varying steepness below the state-dependent means of knee frequencies (Fig. 4., panel D). But no flat lines (steepness equalling zero) are seen in these averages depicted in panel A. This implies that the expected shape of the spectra below the knee frequency is negatively sloped but heavily sleep state-dependent. Consequently, lower (i)EEG frequencies could indeed be characterized by their own, non-zero spectral exponents. In contrast to this empirical finding, the Lorentzian model is strongly determined to fit a straight line in the vicinity of zero frequency (see panel B of Figure 4 and the diverging model and data in the vicinity of the vertical axis!). In other words, frequency-independency is implicitly assumed below the knee, right in the range where large and spectacular differences in slopes are evident by visual inspection. The paper published by Colombo and his colleagues in 2019 is evidently not an implicit support of the usefulness of the knee-model (fitted Lorentzian). In this latter paper non-zero slopes are reported in both the low and high frequency ranges of the EEG (albeit frequency ranges are defined on an ad-hoc basis). I understand that the neural model of Gao suggests a primordial relevance of higher frequency slopes as an indicator of the assumed E-I ratio, as well as the fact that the knee-model is coherent with these theoretical claims. Still, broken power laws, which might be operative in the realm of the (i)EEG are not necessarily well-captured by the knee model (the latter assuming a non-power law shape of the spectra in the lowest frequency range). Consequently, authors are requested to reflect shortly on the potential limitations caused by the knee-model (Lorentzian-shaped spectra), with a special reference to the negatively sloped, potentially (seemingly) sleep stage-specific and non-zero low frequency power-law scaling in the iEEG data. The spectral parametrizations of the lowest frequency range (1-8 Hz) could be informative in this regard. Are these low frequency spectral slopes sleep stage-specific? If yes, are the results congruent with the outcomes of the knee-model-based parametrization (which prefer to emphasize the higher frequency slopes)? I was not able to find these types of data in the manuscript and its supplementary figures or tables.

We agree with the broader point that the power spectrum below the knee, in most cases, does not exhibit a truly flat exponent in log-log space, and that a non-zero exponent may be/is more physiologically plausible. To reiterate, we believe the Lorentzian model is accurate and appropriate in measuring the knee and post-knee exponent – the key measurement aims of our study – while acknowledging it does not fully capture everything about the data. As such, we see our work as part of the evolution of analyses & methods to measure and understand a more nuanced reflection of the signal's complexity than a (narrow-range) single exponent model – though by no means a final destination.

To our knowledge, no standard approach currently allows for robust estimation of two distinct spectral exponents AND a knee frequency in a combined model fit (though we are working on this in other methods-related work). Indeed, while the study referred to by the reviewer (Colombo et al., 2019) measures two spectral exponents (motivated by the knee), it does not explicitly characterize changes in the knee frequency. Thus, we believe the approach we use

here is a reasonable step forward in highlighting the importance of the knee parameter. Our focus is to utilize the knee model as implemented in the `specparam` (formerly `FOOOF`) toolbox to better inform us on the dynamics of sleep and the changes in neural activity during sleep. In doing so, we focus on highlighting the changes in the knee frequency and the underlying physiological implications in addition to the post-knee spectral slope across conditions. We see no reason to believe that the Lorentzian model's assumption of a near-zero slope below the knee materially alters the interpretation of these parameters or the conclusions drawn from them. Nevertheless, we agree that future modeling efforts should continue to refine how both knees and spectral exponents are represented (a direction we fully support and are actively exploring, e.g., in `specparam 2.0`).

To go one step further in line with the reviewer's suggestion:

a. We have now included supplementary analyses of the slopes below the knee across sleep stages (see Supplementary Figure 4-3 of the revised manuscript). These analyses confirm that pre-knee exponents are indeed sleep stage-specific and systematically vary across conditions.

Page 18, Line 509: *It is also important to note that the knee model we use fits a Lorentzian function, which assumes a flat exponent below the knee frequency. To formally test whether pre-knee slopes differ across stages, we fit and compared the exponents below the knee frequency, which showed sleep-stage-dependent variations (see Suppl. Figure 4-3).*

Figure R1. Sleep-stage differences in pre-knee exponent in iEEG data. We estimated the knee frequency per region per stage, then measured the slope of the PSD in the range from 1 Hz till the frequency of the knee. Trials where no knee was detected or knee frequency was below 2 Hz were excluded. The analysis is therefore done on 37 regions (excluding Amygdala as no trials met these conditions in at least one of the stages). The Friedman test revealed a significant difference between exponents of the different stages ($X^2 = 96.02$, $p < 0.001$, $W = 0.87$). Post-hoc tests are reported in Table 5.

b. We extended our discussion acknowledging the limitations of the Lorentzian model in representing low-frequency dynamics and emphasize that the goal is not to rigidly advocate for a specific functional form, but rather to encourage the broader use of knee-fitting as a tool for characterizing timescales in neural data.

Page 31, Line 864: *Although our approach goes beyond a single exponent model, it is also important to acknowledge that the Lorentzian function we fit assumes a near-zero slope*

below the knee, potentially oversimplifying low-frequency dynamics that also appear to be sleep stage-specific. Future work should aim to incorporate models capable of estimating the knee, as well as both pre- and post-knee exponents to more accurately characterize the full aperiodic signal.

b. The spectral knee frequency and the spectral slope are strongly correlated. This point is not acknowledged appropriately, and the caption of Figure 5-1 is even contradicting the information presented in the paper. “D) No correlation between the change of the knee frequency and that of the exponent of the knee model (Spearman correlation: $\rho = -0.006$, $p = 0.96$).” Indeed, there is a particularly high sleep stage-specific correlation between these measures, depicted right in this panel! I do not think that this strong relationship can be considered as non-existing or non-significant (the W stage distorts the relationship but otherwise one can see a positive correlation, which is especially true in the case of REM sleep).

We would like to clarify that we did not dismiss the correlation between the two parameters. Our results from the partial correlation analysis—controlling for sleep stage as a covariate—revealed a significant relationship between knee frequency and the exponent *across* models (i.e., the knee frequency of the Knee model vs. the exponent of the Fixed model), but not *within* the Knee model (i.e., the knee frequency vs. exponent of the same model). Importantly, the partial correlation specifically addresses whether a consistent relationship between knee frequency and exponent exists *across* all stages. In this case, the answer is negative, as illustrated by the figure 5-1: the direction and strength of the correlation vary markedly across stages, potentially cancelling each other out in the global estimate.

So, while we have updated Figure 5-1 and edited the caption in the Figure for clarity, we want to point out that the reported statistic accurately reflected the particular test, whereby we otherwise otherwise do emphasize the stage specific correlations in many occasions:

a. In the results section, we wrote:

Page 19, Line 526: *Given this pattern of multiple aperiodic parameters relating to sleep stages, we next sought to explore the potential relationship between these parameters. To do so, we computed Spearman correlations between the knee frequency and the exponent estimates while considering sleep stages. Comparing the knee frequency of the knee model and the exponent of the fixed model – there was a significant negative correlation (Figure 4G left; $\rho(152)=-0.64$, 95% CI $[-0.73,-0.54]$, $p < 0.001$). This relationship is reversed when examining the exponent values resulting from the knee model. Specifically, the correlation between the knee frequency and the exponent of the knee model was positive and significant (within-model correlation: Figure 4G right; $\rho(152)=0.61$, 95% CI $[0.50,0.70]$, $p < 0.001$).*

b. Further, in the discussion we stated:

Page 32, Line 895: *The relationships of the aperiodic features to each other, within and between the different models, and their differing patterns across stages emphasize that these different models and features are related, but not equivalent. This relates to the impact and importance of the knee – while a true $1/f$ signal has the same exponent value across all*

frequency ranges (which can also be called mono-fractal or scale-free), the knee reflects a frequency-specific transition in the power spectrum. In such a case, which can be called multi-fractal or multi-scale, the knee parameter reflects a transition between $1/f$ regions and the exponent of the knee model reflects the frequency region beyond the knee frequency. Even when models are fit across the same frequency range, their exponents show only weak non-significant correlation, further confirming that they capture different spectral properties and may lead to distinct interpretations.

c. Regarding Figure 5-1, which has re-run the figure and updated the statistical tests and the results remain unchanged.

Figure 5-1. Extended EEG results. A) R^2 for the knee model for the different sleep stages. B) R^2 of the fixed model for the different sleep stages. C-D) Partial correlations between knee frequency and exponents of knee and fixed models after correcting for sleep stages. C) Between-models correlation: The exponent of the fixed model correlated significantly negatively with the knee frequency derived from the knee model (Spearman correlation: $\rho(61) = -0.43$, $p = 0.0005$, 95% CI = [-0.62, -0.2]). D) Within-model correlation: The correlation between the knee frequency and the exponent of the knee model (Spearman correlation: $\rho(61) = -0.15$, $p = 0.26$, 95% CI = [-0.39, 0.11]). E) The topography of the exponent of the fixed model. Each dot in panels A, B, C, and D represents one subject.

2. Authors implement a series of spectral parametrizations by relying on different frequency ranges derived from the literature. Indeed, the results of most of these parametrizations are only used to highlight the superiority of the broadband knee-model. Findings on sleep stage-specificity of these alternative spectral slopes are not included in the manuscript or the supplementary materials. Reasons for this lack of presentation seems to find its roots in the fact that the exponents are strongly dependent on the frequency range itself. Indeed, spectral slopes are not necessarily frequency-independent (see for example the broken power law models which are ubiquitous in nature, as well as the paper of Colombo and colleagues revealing pharmacological specificity of altering lower vs. higher frequency slopes). Authors should at least shortly include the sleep stage-specificity of these exponents and not just consider them as inferior to the highly preferred knee-model-derived ones. Based on the visual inspection of the iEEG spectra particularly large effects are expected in these analyses.

We agree with the reviewer that spectral exponents vary depending on the frequency range used for fitting, and this observation is in fact central to the motivation for our work (Figure 3 and its supplements). Our goal in including multiple parametrizations based on fixed frequency bands from the literature was not to dismiss them as inferior, but rather to illustrate the challenges inherent in comparing slope estimates across different ranges, and to highlight the value of trying a different strategy which incorporates both the knee which and the (post/knee) exponent. Notably, the sleep stage specific variations of these different ranges are generally already reported by the papers that originally employed them (e.g. the papers reflected in Figure 1), whereas given the main goals of our study and the already large number of figures and supplements, we elected to not report the changes across sleep stages for every fit range we explored in Figure 3.

We have revised the relevant text in the discussion to clarify that our intention is not to present traditional slope estimates as necessarily inferior, but to motivate the use of knee-fitting as a complementary approach that can account for changes in scaling behavior across frequencies within a unified model.

Page 31, Line 842: *Accordingly, we sought to build on previous work that investigated the narrowband exponent to explore sleep dynamics and differentiate sleep stages by extending the frequency range and explicitly incorporating the aperiodic knee —an approach that may provide a more complete and physiologically grounded representation of the underlying neural dynamics. Notably, selection of frequency ranges and model forms should be considered together – for example, examining whether fitting a broader frequency range would require*

fitting a knee parameter – as qualitative differences in the structure of the data may impact model fitting, even if quantitative measures of goodness-of-fit such as R^2 are similar.

3. An additional minor, but significant point is the further potential source of the divergence between iEEG and EEG data. Besides other differences, scalp EEG is analyzed in average reference montage, whereas no indication of reference electrode (montage) is given in the case of iEEG records. The latter are presumably bipolar (the authors are invited to consider this point and mention it in the section Methods [iEEG montage] and Discussion [divergent findings for divergent montages]).

We thank the reviewer for this remark. Indeed as the reviewer suggested, the referencing procedure differs between modalities which is likely to introduce differences in the signal. Indeed, the iEEG data were analyzed in a bipolar configuration, where bipolar channels were formed from neighboring contacts on each electrode. We have included this information in the methods section:

Page 8, Line 211: *Already pre-processed data was accessed from the MNI database, which provides channels of bipolar data collected from neighbouring pairs of electrodes.*

Additionally, we have extended the Discussion where we had a paragraph devoted completely to this issue to clarify that some of the observed modality-specific differences may partly reflect differences in referencing schemes. We now state:

Page 34, Line 941: *Another potential contributor to the differences between iEEG and EEG recordings may be their respective referencing montages. While we re-referenced EEG data to an average reference, iEEG recordings used a bipolar montage, which may impact power spectral characteristics and model fitting outcomes. ”*

REVIEWER 2

Major

1) In results section 3.1, the clustering of frequency ranges into “broad” and “narrow” inappropriately penalizes the “narrow” category by combining non-overlapping frequency ranges. — The first analysis of this paper focuses on the effect of the frequency range on aperiodic model fit. The authors’ choice to use ranges found in the literature grounds the paper to the parameters researchers are using in practice, but the downside is that they are now varying two factors at once: the width of the frequency range, and the range itself. Given that the authors wish to emphasize the importance of broad spectral ranges, they should only directly compare overlapping ranges of increasing width (1-8, 1-20, 1-30, 1-45, etc.) and not pool non-overlapping frequency ranges (1-8, 1-20 / 20-45, 30-45). In fact, the R^2 values within the category of “narrow range” appear inconsistent with the overall conclusion, since 1-8 Hz outperforms 30-45 Hz, and 1-20 Hz outperforms 20-45 Hz at both 15 and 20 s PSD windows. The authors could strengthen their analysis by either systematically varying range width in one figure (1-8, 1-20, 1-30, 1-45, etc) and frequency band locations in another (1-20, 10-30, 20-40, etc). At a minimum, the comparisons between 'broad' and 'narrow' frequency ranges (Figure 3C-E) should include only fully nested ranges for a more controlled comparison.

The reviewer here provides a valid point that highlights the unfair comparison between “broad” and “narrow” frequency ranges in our analysis in section 3.1. We agree that the clustering of frequency ranges into “broad” and “narrow” in our original analysis conflates two factors: range width and range location. As the reviewer correctly points out, this makes it difficult to isolate the effect of bandwidth alone, and may unfairly penalize certain narrow ranges that happen to exclude low frequencies.

We would like to clarify that the primary aim of the analysis in Section 3.1 was not to draw direct statistical comparisons between narrow and broad frequency ranges, but rather to illustrate how narrow frequency ranges produce more variable exponent estimates, particularly when the fitting range excludes lower frequencies. In contrast, broadband ranges, although varying in width, yield more stable fits. Thus, the central message of Figure 3 is the sensitivity of model quality estimation to fitting range, especially in narrow bands.

That said, we acknowledge that “nested” frequency ranges would provide a more controlled test of model performance. In response, we have conducted an additional sensitivity analysis (now included as *Supplementary Figure 3–4*) that systematically:

- Varies the bandwidth of the fitting range while keeping the lower bound fixed (e.g., 1–8, 1–20, 1–30, 1–45 Hz).
- Varies the location of the fitting range while keeping the bandwidth fixed (e.g., 1–20, 10–30, 20–40 Hz).

These new analyses confirm our initial conclusions: model fit quality decreases when narrow frequency ranges are used, and when low frequencies are excluded, supporting our interpretation that broader fitting ranges, particularly those that include lower frequencies, result in more reliable estimates of aperiodic activity.

The discussion section already includes such statement:

Page 30, Line 827: *Thus, we suggest that future studies of aperiodic activity during sleep should explicitly indicate which fitting ranges are used, consider fitting broader ranges (> 20 Hz bandwidth and encompassing lower frequencies), where appropriate, and, where possible, include sensitivity analysis examining the dependency of findings on the fitting range.*

Furthermore, we have incorporated this clarification into the main text (see revised paragraph below) and now reference the new supplementary figure. However, we opted to retain the original structure of Figure 3 in the main manuscript, as it serves a conceptual purpose in aligning with the narrative. We believe that the supplementary analysis now provides additional methodological rigor to support the main claims.

Page 16, Line 460: *To validate our analysis and disentangle whether differences in model fits were driven by the bandwidth (narrow vs. broad) or the location (low vs. high frequencies) of the frequency range used, we conducted additional analyses (Suppl. Figure 3-4). Specifically, we varied either the width of the frequency band (Suppl. Fig. 3-4A) or its location along the spectrum (Suppl. Fig. 3-4B). These analyses revealed a decline in model fit (R^2) when narrower frequency bands were used or when lower frequencies were excluded. Collectively, while R^2 values do not in isolation adjudicate between good and bad models (see e.g. Figure 2), these results suggest that fitting spectral models over broader frequency ranges, particularly those that include lower frequencies, provides more reliable estimates.*

Figure 3–4. Effects of frequency range width and location on spectral model performance in iEEG data. A-B) Sensitivity matrices showing R^2 values for models fitted across varying

frequency ranges (x-axis) and time windows (y-axis). A) Increasing the width of the frequency band used for model fitting improved model performance, as indicated by higher R^2 values. B) Shifting the frequency band to higher frequency ranges reduced model fit quality, suggesting that excluding lower frequencies has a detrimental effect on model performance.

Related: The correlation analyses presented in Figure 3E and Supplementary Figure 3-3 appear inappropriate given the clustered distribution of the data points. In Figure 3E, we can see that for a nearly constant exponent variance value of approximately 0.55, R^2 values vary across the entire observed range, suggesting no correlation within this cluster. The reported significant correlation between exponent and R^2 variance appears to be driven by the relationship between clusters rather than a true continuous relationship within the data; a classic example of Simpson's paradox. The conclusion that "model fits influence the variability in exponent values" (page 15, line 435) is questionable, also since this relationship is not observed for any of the other models examined in Supplementary Figure 3-3, where correlations are non-significant. I recommend either removing this conclusion or conducting a more appropriate analysis that accounts for the clustered nature of the data.

We thank the reviewer for this fair point. We agree with the reviewer that the correlation might be driven by between-cluster differences rather than within-cluster relationships. To circumvent this issue we performed two separate partial Spearman correlations for each range (Narrow vs Broad). We have consequently adjusted this part of the results section and toned down our conclusion.

Page 16, Line 448: *Additionally, to examine the association between spectral exponent and R^2 , we conducted partial Spearman correlation analyses separately for broad and narrow frequency ranges, while controlling for sleep stage (Figure 3E). In the broad frequency range, the partial correlation revealed a moderate positive association between exponent and R^2 ($r=0.37$, $p=0.003$, 95% CI=[0.13, 0.56]). A similar effect was also observed in the narrow frequency range, with a partial correlation of $r=0.37$ ($p=0.003$, 95% CI=[0.14, 0.57]). These results indicate that better model fits were associated with steeper exponents.*

In Figure 4E. A significant positive correlation was observed between the variances in R^2 and exponent values of the narrow ($\rho = 0.37$) and broad ($\rho = 0.37$) frequency ranges.

2) Differences between the supplementary data and the main figures are often understated, sometimes in ways that are quite critical to the narrative of the paper. — The authors need to better emphasize the differences between the results with different models and different datasets.

We apologize for such discrepancy. We respond to the reviewer at their respective points

a. On page 15 line 436, it says that the EEG data yielded similar results as iEEG data (Figure 3 vs. Suppl. Figure 3-2). This is inconsistent with the data presented in the Figure 3-2: when applied to EEG data, broader frequency ranges yielded worse models than narrower ranges. Models 1-20 and 1-45 Hz had identical R^2 values, and even 1-8 Hz outperformed 1-75 and 1-100 Hz. These results for EEG data are not surprising (high frequencies are known to be unreliable in the surface EEG), but it is important that the authors acknowledge this difference in the main text, especially since more readers will have access to EEG data rather than iEEG data.

We agree with the reviewer that iEEG and EEG results do not always perfectly align, however, they broadly align in both modalities. To ease the comparability of both modalities, we have adjusted the frequencies of the EEG analysis to match those of iEEG and to avoid very high frequencies which, as the reviewer correctly stated, are unreliable in EEG. Consequently we have updated the respective figure (Suppl. Fig 3-2) and adjusted the text in the results section.

- Adjusted Text:

Page 16, Line 455: However, in the EEG data, model performance varied more notably with frequency range: broader ranges yielded lower model fits compared to narrower bands, with the 1–8 Hz range outperforming both 1–75 Hz and 1–100 Hz (Suppl. Figure 3-2).

- Adjusted Figure:

Figure 3-2. Model performance for different frequency ranges in EEG. A) The R^2 values for the fixed model using broadband frequency ranges. B) The R^2 values for the fixed model using narrow frequency ranges. C-D) The R^2 values of the knee model using broadband (C) and narrowband (D) frequency ranges.

b. On page 17 line 477, the text says that the exponent of the fixed model did not significantly differ between sleep stages, but the values in table 4 indicate that they did. This inconsistency needs to be resolved, as it appears to downplay the performance of the fixed model when the data suggests it may perform better than reported.

We apologize for the mistake in the wording. We have corrected the text and verified the statistical details. The results presented in Table 4 are accurate; only the wording required adjustment.

Page 18, Line 505: Specifically, the exponent decreased significantly from Wake to N2 ($p_{\text{bonf}} < 0.001$), Wake to N3 ($p_{\text{bonf}} < 0.001$), from N2 to N3 ($p_{\text{bonf}} = 0.001$), from N2 to REM ($p_{\text{bonf}} < 0.001$), and from N3 to REM ($p_{\text{bonf}} < 0.001$). However, there was no significant difference between Wake and REM ($p_{\text{bonf}} = 1$).

c. On page 24, line 661, the claim that fixed model exponents 'were less temporally related to the transitions between stages' is not supported by Supplementary Figure 6-2, where the dynamics appear equally or more pronounced than with the knee model. The main difference is that the fixed model shows larger baseline differences between stable and transition epochs, which may actually indicate greater sensitivity to pre-transition changes in brain state.

We thank the reviewer for this remark. We agree that the comparison between the fixed and knee models in relation to stage transitions is more nuanced than what we wrote in the manuscript. While it is possible that the larger baseline differences observed in the fixed model reflect greater sensitivity to pre-transition changes, an alternative explanation could be that these differences arise from methodological constraints of the fixed model, i.e., inability to account for knee frequency changes, which could lead to exaggerated (possibly artifactual) differences across conditions. Speculatively, it seems possible that the pre-transition differences in exponent of the fixed model actually capture some differences in the knee. Note, for example, that the purple N2-N3 pre-transition exponent difference in 6-2C seems to match the pre-transition knee difference in 6-3C and the same can be said for REM transitions in 6-2D vs 6-3D – whereas N1-N2 transitions which do not show a pre-transition difference in exponent also do not show a difference in knee (6-2A vs 6-3A). If so, this is consistent with the general argument that the exponent of the fixed model can conflate these two features that are differentiated in the knee model, and this would suggest that exponent captures changes are more time-locked to state transitions, where as knee changes are perhaps slower / more variable and gradually change without necessarily being locked to state transitions. It is with all this in mind that we originally made our comments about the differences between models (and think the knee model is useful in this case), however we do agree that it is too simplistic to say that the fixed model is less temporally related to transitions. With this in mind we have updated the phrasing of this in the paper:

Page 25, Line 699: *The results are broadly consistent, and we found no statistically significant difference in the model fits (R^2 values) between the knee and fixed models (Paired t -test: $t(16) = 1.885$, $p = 0.108$, $d = 0.22$), suggesting that the two models may capture complementary aspects of stage-dependent spectral dynamics rather than differing in overall fit quality.*

3) While the paper advocates for including a knee parameter in aperiodic analyses, the authors don't adequately address a concerning inconsistency: when comparing results between recording modalities, the exponents from the fixed model remain largely comparable between

iEEG (Figure 4E) and EEG data (Figure 5D), whereas the knee parameters show dramatic differences (Figure 4D vs Figure 5C). Specifically, the exponent direction reverses across sleep depth, and knee frequencies are substantially lower in EEG compared to iEEG data. This inconsistency is problematic, as ideally, EEG analyses should correspond as closely as possible to iEEG measurements. The divergence in knee parameters between recording modalities raises questions about the reliability and interpretability of this parameter. The authors should directly address this limitation and explain why they still recommend including a knee parameter despite these cross-modal inconsistencies. This issue deserves thorough treatment in the discussion section, as it has significant implications for researchers deciding whether to implement knee models in their analyses of sleep EEG data.

We acknowledge the reviewer's concern regarding the difference in knee parameter estimates between iEEG and EEG recordings. While the exponents derived from the fixed model appear relatively consistent across modalities, it is precisely the strength of the knee model that it can detect frequency-dependent changes in the exponent metric. That is not the case for fixed models. Rather than interpreting this divergence as a limitation, we view it as an illustration of the model's sensitivity to the specific signal properties.

This contrast rather points to the importance of adopting a flexible, data-driven approach to model selection, tailored to the characteristics of the data. As we have already noted in the discussion, our aim is not to advocate for a single "best" model, but rather to emphasize that model form should be evaluated explicitly. In doing so, we can uncover meaningful differences that may otherwise remain obscured, and refine theoretical interpretations accordingly.

We have already stated that in the discussion:

Page 33, Line 929: *By analyzing sleep data from intracranial (iEEG) and extracranial (EEG) recordings, we identified both similarities and modality-specific differences that warrant further investigation.*

Page 34, Line 935: *These findings highlight the need for further research into stage-specific knee dynamics in EEG. While stage-dependent knee frequency has been observed in other contexts (Lendner et al., 2024; Nakata et al., 1993), its role in sleep spectral dynamics remains underexplored. This underscores the importance of data-driven model selection, prioritizing broadband frequency ranges and robust goodness-of-fit measures to improve reproducibility and model selection across neural states.*

Moreover, related to the comment number 3 of Reviewer 1, we have also added a sentence in the discussion as to why knees might not be consistently present across modalities due to different preprocessing decisions:

Page 34, Line 941: *Another potential contributor to the differences between iEEG and EEG recordings may be their respective referencing montages. While we re-referenced EEG data to an average reference, iEEG recordings used a bipolar montage, which may impact power spectral characteristics and model fitting outcomes.*

We would like to reassure the reviewer that our goal is not to assert the superiority of the knee model, but to advocate for explicitly testing model forms that account for potential nonlinearities in the aperiodic spectrum. Such models can help reveal systematic modality-specific or state-specific spectral features that may be overlooked when using fixed models. In this context, the emergence of a spectral knee in iEEG data should not be viewed as a problematic inconsistency, but rather a feature of the method that is worthy of investigation as it may provide a more accurate representation of the underlying neural activity. While knees were also detected in EEG data, their appearance was less consistent, possibly due to differences in spatial resolution, signal-to-noise ratio, and sensitivity to deeper sources. Given the potential physiological relevance of the knee, modeling it remains valuable. Therefore, we see this work as a step toward optimizing such modeling approaches, addressing questions about the presence and behavior of knees and exponents across recording modalities and sleep stages. We hope our findings encourage future studies to explore a wider range of model structures and parameters across modalities and neural states, ultimately refining both methodological practices and theoretical interpretations. In this light, our study is intended as a starting point for more rigorous modeling of aperiodic activity in neural signals—particularly in sleep—rather than a conclusive framework.

That said, we have revised the respective discussion section and expanded on the points raised by the reviewer:

***Page 34, Line 945:** Interestingly, applying the knee model revealed differences between modalities that were not evident when using the fixed model. Specifically, while the pattern of exponent values across sleep stages was consistent across modalities in the fixed model—and remained similar in the EEG data when using the knee model, the iEEG data showed a distinct pattern of exponent changes across stages under the knee model. This suggests potential modality-specific effects, likely related to the influence of knee prominence on exponent estimates. These differences may also reflect variation in underlying signal sources between EEG and iEEG. Overall, this result highlights the importance of selecting model forms that align with the characteristics of the data and the goals of the analysis, and contributes to the broader research effort aimed at understanding how different parameterizations capture neural dynamics across modalities.*

Minor

Introduction

- page 3 line 71 “This relationship is consistent with observations that the spectral exponent of the electroencephalography (EEG) signal becomes progressively steeper from wakefulness to non-rapid eye movement (NREM) sleep to rapid eye movement (REM) sleep”, whether REM is steeper than NREM is controversial; it depends on the frequency range, with lower-frequency broad ranges regularly finding REM exponents to be less steep than N3 and comparable to N2

(e.g. Schneider et al. 2022, this paper's figure 4E) and narrow band high frequency ranges showing steeper REM exponents compared to N3 (e.g. Lendner et al. 2020). I suggest the authors either directly address this issue, for example from the perspective of knees in the data, or rephrase this sentence to merely saying that EEG becomes steeper with deeper sleep.

Adjusted:

Page 3, Line 73: *This relationship is consistent with observations that the spectral exponent of the electroencephalography (EEG) signal becomes progressively steeper with deeper sleep (Lendner et al., 2020; Höhn et al., 2024; Kozhemiako et al., 2022; Rosenblum et al., 2022), consistent with an increase in inhibitory processes during sleep (Niethard et al., 2016; Birdi et al., 2020).*

- Page 4 line 81-82, it's a bit of a mischaracterization that there is a tendency to examine the 30-45 Hz range, considering the ranges provided in Table 1.

Adjusted:

Page 3, Line 82: *Previous sleep studies have thus far examined a wide variety of frequency ranges (Figure 1B). Narrow frequency ranges such as the 30 to 45 Hz range (Demirel et al., 2021; Kozhemiako et al., 2022; Lendner et al., 2020) are often chosen due to their potential relationship to the E-I ratio (Gao et al., 2017).*

Methods

- Please specify how line noise was removed from the iEEG data (page 8). Was it filtered? How does the specparam algorithm handle the frequencies of line noise when the frequency range used to fit the aperiodic signal overlaps, as in the case of the iEEG data and the results in section 3.1?

Thanks for this remark, we have extended our description of the line noise removal procedure in the revised manuscript.

Page 8, Line 213: *Line noise was attenuated using an adaptive filtering approach that estimated the amplitude of the line noise frequency (50 or 60 Hz depending on recording site) and the first two harmonics, which were subsequently removed.*

- On page 8, the information regarding concatenating artefactual segments is a little hard to follow. Was this process done in the original dataset, including the 0-padding so all segments reached 68 seconds? How does zero-padding 2 s between segments minimize artefacts? Which artefacts? Typically, such an approach would lead to edge artifacts, especially

problematic for the PSD, unless the start and end of each EEG segment was tapered to 0. Also, was there a minimum segment duration?

To clarify, the process of concatenating artefact-free EEG segments was carried out by the authors of the open-access dataset, not by us. The full preprocessing pipeline is described in the dataset documentation.

Briefly, we want to clarify the following:

- The artefact-free segments were concatenated by the dataset authors with 2 seconds of zero-amplitude padding inserted between them. This buffer minimizes the risk of introducing transient artefacts that could occur at the junctions between segments due to abrupt signal transitions. While we agree that such concatenation can introduce edge artefacts, particularly for spectral analyses such as PSD, this risk is mitigated to some extent by inserting zero-padding as a temporal buffer.
- As the number of concatenated segments varied across patients, the number of inserted 2-second zero-paddings also differed, leading to variable total signal lengths. To standardize this, all signals were zero-padded at the end to reach a uniform length of 68 seconds (13,600 samples at 200 Hz), regardless of how many segments were used.
- No mention is made of a minimum signal duration nor any tapering procedure before concatenation, so we acknowledge that edge artefacts could still occur. However, any such artefacts would primarily affect a narrow window of time surrounding the junctions and are unlikely to bias the analyses that average spectral content over longer time scales.

This procedure has been made clearer in the manuscript

Page 9, Line 220: *To minimize potential artifacts at segment boundaries, a 2-second buffer of zero amplitude was inserted between concatenated artefact-free segments. Up to five such segments were concatenated per condition, resulting in a maximum of four 2-second buffers and a total duration of 68 seconds (60 seconds of EEG data plus 8 seconds of inter-segment buffers). As the number of clean segments varied across recordings, the number of inserted buffers also differed. Therefore, to ensure uniformity in signal length across all channels, sleep stages, and patients, signals shorter than 68 seconds were zero-padded at the end to reach a consistent length of 68 seconds.*

- Could the authors provide the number of trials used in the K-complex analyses?

We added this information to the manuscript.

Page 12, Line 325: *For KC analysis, we used 74.47 ± 45.04 trials on average per subject.*

- Page 11 line 291, I recommend indicating the figure being referred to, and what was averaged: “For the whole night time-frequency plot (Figure 6A) we plotted the channel-averaged value per epoch”

Adjusted

- In section 2.4.2, is there a reason 20 s windows were used for the PSD, but for other analyses it was 15 s? In general, is there a reason for 15 s windows for the PSD, especially considering that the results of figure 3 show better performance for shorter windows?

The choice of 15-second and 20-second window lengths reflects a compromise between computational efficiency and methodological performance. While shorter windows—as shown in Figure 3—may yield better model fits, particularly for dynamic phenomena such as KC analysis, they substantially increase the number of segments to process, resulting in higher memory usage and longer runtimes, especially when analyzing full-night recordings across multiple subjects. We therefore opted for relatively longer time windows (15 s for classification tasks, 20 s for PSD estimation) as a practical compromise: short enough to preserve relevant temporal dynamics, yet long enough to ensure computational tractability for large-scale analysis.

- In section 2.4.3, the KC data is baseline corrected in the 500 ms prestimulus window; based on the results of Figure 7, this may be somewhat problematic, as there are stimulus evoked effects even before stimulus onset. Ideally the authors should repeat the analysis with an earlier baseline correction. Alternatively, the authors may consider indicating somewhere in the text that future studies should use baselines further removed from stimulus onset, as there seems to be some temporal smearing of the time-resolved aperiodic signal.

We appreciate the reviewer’s observation regarding potential pre-stimulus effects in the time-resolved aperiodic signal (Figure 7). We chose the 500 ms pre-stimulus baseline to maintain consistency with prior analyses conducted on the same dataset (Ameen et al., 2022), which used the same baseline window. While we acknowledge that stimulus-evoked activity may extend into the baseline period, potentially due to temporal smoothing and the choice of window size for PSD estimation, we opted to retain this baseline to allow for direct comparisons with previous work.

- In section 2.6, the authors describe baseline correction which is a bit unconventional for ERPs; usually, like the authors did in section 2.4.3, a pre-stimulus baseline is subtracted from each trial before averaging. Why was a different approach for baseline correction used for these two analyses, considering that they are directly compared between each other?

We thank the reviewer for pointing this out. This is actually a mistake in the order during reporting. In all analyses we performed pre-stimulus baseline subtraction prior to averaging. We have revised the text in Section 2.6 to accurately reflect the correct order of processing steps.

Page 13, Line 352: We performed baseline correction using the formula (data – mean baseline values) / mean baseline values followed by calculating the grand average of all epochs per participant

- Can the authors provide a source for the interpretation of Cohen's interpretation of effect sizes for Kendall's W (page 14 line 378-380)?

We have adjusted this section:

Page 14, Line 388: Kendall's W ranges between 0, indicating no relationship, and 1, indicating a perfect relationship. Interpretation of W values followed commonly used benchmarks, where W values between 0.00 and 0.20 indicate slight agreement, 0.21 to 0.40 fair agreement, 0.41 to 0.60 moderate agreement, 0.61 to 0.80 substantial agreement, and values above 0.80 reflect almost perfect agreement (Landis and Koch, 1977).

Results

- Page 17 line 463; the authors could add Schneider et al. 2022 to the references, especially since those results are more in line with the ones presented in this paper.

Added.

Page 18, Line 490: Previous findings have reported that the spectral exponent differs between sleep stages (Höhn et al., 2024; Kozhemiako et al., 2022; Lendner et al., 2020; Schneider et al., 2022).

- Page 17 line 485 is confusing, “as well as the exponent of the knee model performed at chance ($t(15) = 5.61$, $p_{\text{bonf}} < 0.001$, $d = 1.99$)”; the results show it performs significantly above chance.

We apologise for the typing mistake. It has been corrected in the new version.

Page 18, Line 516: The classification results revealed significantly above chance-level classification accuracy using the knee frequency ($t(15) = 10.57$, $p_{\text{bonf}} < 0.001$, $d = 3.73$), the exponent of the fixed model ($t(15) = 18.28$, $p_{\text{bonf}} < 0.001$, $d = 6.46$), as well as the exponent of the knee model ($t(15) = 5.61$, $p_{\text{bonf}} < 0.001$, $d = 1.99$).

- Page 19, line 516 “the knee frequency was a significantly better predictor of stages than the exponent, emphasizing the significance of fitting a knee model to the data”; this assertion could be made if there is a significant improvement between the fixed and knee model, but not between parameters of the same knee model. I would recommend removing it.

Our intention was to highlight that, within the knee model, the knee frequency was assigned significantly higher feature importance than the exponent in the Random Forest

classifier. We have revised the sentence accordingly to reflect this more precisely and have removed the overstated claim regarding the model comparison.

Page 19, Line 550: Results showed that the knee frequency (permutation importance: 0.14 ± 0.07) is significantly more important than the spectral exponent (permutation importance: 0.07 ± 0.05) ($t(15) = -5.13$, $p < 0.001$, $d = 1.06$), suggesting that within the knee model the knee frequency may be a more informative predictor of sleep stages than the exponent.

- Figure 4, panel G comes before E and F, was this a mistake or deliberate?

The placement of panel G before panels E and F was a deliberate decision based on aesthetic and layout considerations to optimize the visual balance of the figure. We ensured that the narrative in the main text follows the correct logical order of the panels, regardless of their visual positioning.

- In the caption of Figure 4F, it says “note that both the knee and the exponent of the fixed model had significantly above chance classification accuracies” this is a bit uncalled for, since all three classifiers are significant. I would remove.

Removed

- The last line of the caption of Figure 4 and Figure 5 “Each dot represents one brain region...” probably belongs to a different section of the legend, as it does not refer to the dots in Figure 4J or topographies of Figure 5G

These lines in the captions of Figures 4 and 5 are intended as general footnotes to clarify the meaning of dots wherever they appear across the figures. We have now reformatted the captions to make this clearer and to avoid potential misinterpretation..

- Page 20 line 556, the text says “in contrast to the iEEG data, the knee frequency did not exhibit significant differences between stages” but the p value provided in parentheses is significant ($p < .001$). Please check the statistics.

Thanks for pointing out the inconsistency. We re-ran the statistical analyses and corrected the corresponding sentence accordingly.

Page 21, Line 593: Similarly, the knee frequency differed significantly across sleep stages (Figure 5C, right; $\chi^2 = 16.48$, $p < 0.001$, Kendall's $W = 0.41$). Post-hoc comparisons revealed the most significant differences between N3 and Wake ($p = 0.02$), N3 and N2 ($p = 0.004$), and N3 and REM ($p = 0.03$). For detailed post-hoc statistics, see Table 7.

- In the caption of Figure 5F, BIC differences are provided in parentheses that do not correspond to those in the figure

We apologize for this mistake. This sentence has been adjusted and the values have been removed.

Figure 5F) BIC comparison between the knee model and the fixed model suggesting that the knee model provided a better fit than the fixed model in a stage-dependent manner.

- In the analyses of Figure 6F (+ suppl) NREM substages were all pooled. Were the control epochs matched by NREM substage to those of the NREM stages at the transitions, such that the same proportion of each substage of NREM was in both the control and transition data? The matrix in Suppl. Figure 6-1, indicates that the majority of transitions to REM came from N2, many came from N1, and almost none from N3. Given this, shouldn't N1-REM transitions also be included?

Regarding the first question: Yes, the number of epochs from each sleep stage was matched between the control and transition conditions.

Regarding the second question: The difference in the number of usable trials when including N1 epochs (mean number of epochs over all subjects: 5.53 ± 2.1) versus using only N2 and N3 epochs (mean number of epochs over all subjects: 4 ± 1.53) was minimal. This relatively small difference is due to the use of long epochs spanning from -60 to $+60$ seconds relative to the transition point. Notably, the N1 duration preceding REM sleep is typically not very long (1 epoch usually). Given this limited impact, we opted to retain our original analysis.

- In Table 1, Höhn et al. 2024 could be included?

We appreciate the suggestion to include Höhn et al. (2024) in Table 1. However, our literature analysis was conducted and finalized at the end of 2023, prior to the original preprinted version of this study. As such, Höhn et al. (2024) falls outside the predefined temporal scope of our review. We have now clarified further this in the manuscript to avoid confusion.

- In the tables, please provide a consistent number of decimal places for the p-values (ideally 3 or 4)

Adjusted

Discussion

- Page 30, lines 793-795 “notably, fitting a single exponent to a narrow band range ignores the potential presence of a knee or bend in the PSD”; this sentence may mischaracterize the literature. The choice of narrower and higher frequency ranges (35-45 Hz) by the cited authors deliberately avoids any of the lower frequency knees described in this paper (generally lower than 10 Hz). They are not ignoring the knee so much as avoiding it entirely.

The reviewer is correct. The wording of this sentence has been improved.

Page 31, Line 836: *Notably, fitting a single exponent to a narrow band range avoids detecting a 'knee' or bend in the PSD, after which the exponent changes (Gao et al., 2020; Miller et al., 2009), which can provide additional information beyond the single-exponent model (Zempel et al., 2012).*

- The authors have an extremely counterintuitive result: aperiodic exponents with the knee model in the iEEG data are steeper in wake than in sleep (Figure 4D). This goes against the theory that steeper slopes reflect the increased inhibition during sleep, as the authors mention to explain the results in Figure 7 (page 33, line 917-919). How do the authors reconcile this result of Figure 4D with the excitation/inhibition hypothesis? How do they explain that the same knee model yields opposite results across wake and sleep in iEEG vs EEG data? The authors should avoid relying on the excitation/inhibition hypothesis to explain other parts of their data, unless they can explain this discrepancy. It may undermine the interpretability of the exponent in a knee model entirely.

We thank the reviewer for this valuable observation. We have dedicated an entire section of the discussion to thoroughly address this issue. For the very reason mentioned by the reviewer, we had decided not to place too much emphasis on the relationship between the exponent and EI ratio.

Take on the opposing results between iEEG and EEG and link results to previous work:

Page 32, Line 880: *In terms of interpretations of aperiodic neural activity, previous investigations which have largely focused on changes in the aperiodic exponent, have typically interpreted changes in this parameter in terms of its putative relationship to the excitation-inhibition (E-I) balance, whereby a steeper slope signifies an increase in inhibition, and conversely, a flatter slope indicates an increase in excitation (Gao et al., 2017; Chini et al., 2022). The pattern of changes in previous studies is consistent with a general shift towards more inhibition during the transition from wakefulness to sleep (Birdi et al., 2020; Niethard et al., 2016) with the exponent becoming steeper from wakefulness to NREM to REM sleep (Lendner et al., 2020; 2023). While our findings partially align with previous observations, demonstrating that the exponent differs significantly across the sleep-wake cycle, the direction of these differences varies depending on the model used. Specifically, when fitting a fixed exponent model to both iEEG and EEG sleep data, we observed a progressive steepening of the exponent from wakefulness to sleep. In contrast, iEEG data analyzed with the knee model showed a decrease in the exponent from wakefulness to sleep, with no further changes observed during sleep. This discrepancy reflects distinct features of the data that are captured by the knee and fixed models. The relationships of the aperiodic features to each other, within and between the different models, and their differing patterns across stages emphasize that these different models and features are related, but not equivalent. This relates to the impact and importance of the knee – while a true 1/f signal has the same exponent value across all frequency ranges (which can also be called mono-fractal or scale-free), the knee reflects a frequency-specific transition in the power spectrum. In such a case, which can be called multi-fractal or multi-scale, the knee parameter reflects a transition between 1/f regions and the*

exponent of the knee model reflects the frequency region beyond the knee frequency. Even when models are fit across the same frequency range, their exponents show only weak non-significant correlation, further confirming that they capture different spectral properties and may lead to distinct interpretations.

In response, we have added a sentence at the end of the paragraph to emphasize the model-dependent nature of the relationship between the aperiodic exponent and excitation–inhibition (E/I) balance, highlighting that different exponent estimates may capture distinct aspects of neural dynamics and relate to sleep in different ways.

Page 33, Line 907: *This also implies that the association between the exponent and E/I balance is necessarily more nuanced than a one-to-one mapping, as different frequency ranges can have different measured exponent values, which may relate to E/I balance in different ways.*

Supplementary material

- On page 49 there's a figure with no caption
- Suppl. Figure 4-2A has “knee model” as a title and “R2-fixed” for a y-axis label. This is switched also for Figure 4-2B

Thanks for the remark. This has been corrected in the revised version.

Optional

- Considering the importance of the knee to this paper, I would recommend expanding a little more in detail what “the population timescale” means (page 4, line 101) and how it relates to sleep stages, and why it theoretically reflects different information from the exponent (since these values are then shown in this paper to be highly correlated).

This has been expanded on in the introduction, not too much though for length considerations.

Page 4, Line 102: *The frequency at which this knee occurs, referred to as the knee frequency, has been proposed to reflect the population timescale, i.e. the characteristic duration over which a neural population integrates or processes information (Miller et al., 2009; Gao et al., 2020). Notably, this knee frequency has also been shown to vary systematically across sleep stages (Höhn et al., 2024; Kozhemiako et al., 2022; Lender et al., 2023).*

- For how the methods section is currently structured, it was difficult when reading the results to keep track and trace back how data was processed. I would recommend mirroring the methods section to the final results section, such that for each section first describes how the data was processed, how power was calculated, how the aperiodic model was fit, and finally the statistics. With each subsequent section, if the data or method was the same as a previous section, this could simply be indicated without the need for repetition.

We thank the reviewer for this helpful suggestion. We agree that aligning the structure of the Methods section more closely with the Results section would improve clarity and traceability. However, implementing this fully would considerably increase the length of the Methods section and introduce a high degree of redundancy, as many steps such as power estimation and aperiodic fitting that are repeated across analyses with minimal variation.

- The first analysis varies both frequency ranges and PSD window sizes; the authors discuss at length the frequency ranges but mostly ignore the effect of PSD window size; considering that they find smaller windows provide greater R^2 values, it would have been appropriate to conduct the later analyses with these 5 s windows. Understandably this would be a major revision over a minor point, but maybe the authors could discuss this aspect of the parameter selection as well, and why one should choose either short or long windows for the PSD.

As noted in a related comment on Section 2.4.2, the choice of PSD window size represents a trade-off between model accuracy and computational efficiency. While shorter windows (e.g., 5 s) produced higher R^2 values, applying such short windows across all subsequent analyses, particularly for long overnight recordings, would have substantially increased computational demands. Importantly, when using broad frequency ranges, the performance differences due to time windows were minimal, as R^2 values remained high across conditions. Therefore, we adopted slightly longer windows in later analyses as a practical compromise. Our overall approach remains data-driven, with parameter choices tailored to the specific demands of each analysis, as previously discussed in the manuscript.

Page 34, Line 938: *This underscores the importance of data-driven model selection, prioritizing broadband frequency ranges, adequate time windows, and robust goodness-of-fit measures to improve reproducibility and model selection across neural states.*

We thank the editor and the reviewers for their constructive feedback and valuable suggestions, which have helped us improve the clarity and overall quality of the manuscript. The reviewers' comments were minor, and we have addressed each of them in detail in the following responses.

Reviewer #1 (Remarks to the Author):

Authors of the revised paper claim that the estimated parameters of the aperiodic component of the wake-sleep state specific electrical brain activity largely depend on the frequency range and the inclusion of a knee-parameter inherent to the Lorentzian function of a modified power law model. Broader frequency ranges, inclusion of low frequency activities and the implementation of the knee-parameter-based modelling increased the reliability of the estimation. Moreover, the knee-parameter resulted in a radical change of the wake-sleep state-specific differences in aperiodic spectral slopes. Likewise, the typical value of the spectral knee parameter was revealed to be subject of wake-sleep state-specific changes and depending heavily on recording modalities (intracranial EEG with bipolar reference vs scalp-derived EEG of the average reference-type). Last, but not least the time-resolved analyses of wake-sleep state transitions and stimulus-driven electrophysiological changes revealed a significant methodological divergence of the fixed (no-knee) vs knee-based estimations of the aperiodic activity. The findings are quite divergent, and the methodological pluralism implemented by the authors primarily resulted in different sets of combinations of recording-, setting- and model-specific findings. The latter raise new questions in the field of the spectral parametrization of wake-sleep state-specific neurophysiological signals and could open new perspectives in interpreting the state-specificity of different time scales and the integration of neural activity over time. I think the number of questions raised by the knee-model are still high, but more appropriately and explicitly considered in the current form of the manuscript. Some of the state specific differences in aperiodic slopes reported before are now translocated to the knee frequencies, at least when considering bipolar iEEG. I think the paper might influence thinking in the field, but will also imply the emergence of some critical thoughts on the appropriateness of the Lorentzian-model in this case (namely spectra of , (i)EEG activity).

The methods are sufficiently described and need no further details in order to provide the reader with reproducibility.

I consider the paper as being appropriate for publication in its present form.

We thank the reviewer for their thoughtful and comprehensive feedback on our manuscript. We greatly appreciate the recognition of the significance of our findings, as well as the constructive criticism provided. We agree that our analyses introduce complexity and raise new questions, which we believe will contribute to a deeper understanding of neural signals and brain activity during sleep. We are pleased that the reviewer finds the current work complete and look forward to exploring these avenues further in future studies.

Reviewer #4 (Remarks to the Author):

The authors have addressed all of my previous concerns.

We thank the reviewer for the valuable remarks that have improved our manuscript significantly.

A few minor points:

- the new text regarding Figure 3E no longer discusses "exponent variance" but rather exponents themselves, despite the figure being about exponent variances. The new conclusion is "better model fits were associated with steeper exponents"; is this only referring to the correlations described in the text? in which case, Figure 3E comparing variances no longer seems relevant.

We thank the reviewer for pointing this out. The original interpretation did not accurately reflect what is shown in Figure 3E. Our intention with this figure was to demonstrate that higher variance in R^2 values is accompanied by greater variance in spectral exponent estimates, which has direct implications for replicability. To address this, we have revised the wording in the manuscript to clarify this interpretation.

Page 16, Line 448: *Additionally, to examine the association between model fit stability and spectral estimates, we compared the variance of spectral exponents with the variance of R^2 values using partial Spearman correlation analyses, conducted separately for broad and narrow frequency ranges while controlling for sleep stage (Figure 3E). In the broad frequency range, the partial correlation revealed a moderate positive association between exponent variance and R^2 variance ($\rho(13) = 0.37$, $p = 0.003$, 95% CI = [0.13, 0.56]). A similar effect was also observed in the narrow frequency range, with a partial correlation of $r = 0.37$ ($\rho(13) = 0.003$, 95% CI = [0.14, 0.57]). These results indicate that greater variability in model fits corresponds to greater variability in exponent estimates, suggesting that instability in model fitting can reduce the replicability of spectral exponent measures.*

- the tables continue to not have consistent decimal places for p-values, the new table 5 indicates p-values of 0, and the tables indicate > .001 repeatedly, but should likely be < .001. But I imagine this is all at the editor's discretion

We have again carefully revised the tables to ensure consistent decimal places across all entries. Instances where p-values were previously indicated as "0" have been corrected to "< .001," and formatting has been standardized throughout.

- the caption of figure 4-3 appears incomplete, with W left empty and "table X".

We apologize for the typo. This has now been corrected.

Page 54, Line 1395: *The Friedman test revealed a significant difference between exponents of the different stages ($\chi^2 = 96.02$, $p < 0.001$, $W = 0.87$). Post-hoc tests are reported in Table 5.*